# Distributionally Robust Reinforcement Learning with Interactive Data Collection: Fundamental Hardness and Near-Optimal Algorithms

**Miao Lu**[1]*      **Han Zhong**[2]*      **Tong Zhang**[3]      **Jose Blanchet**[1]

[1]Department of Management Science and Engineering, Stanford University
[2]Center for Data Science, Peking University
[3]Department of Computer Science, University of Illinois Urbana-Champaign

## Abstract

Distributionally robust reinforcement learning (DRRL), often framed as a robust Markov decision process (RMDP), seeks to find a robust policy that achieves good performance under the worst-case scenario among all environments within a pre-specified uncertainty set centered around the training environment. Unlike previous work, which relies on a generative model or a pre-collected offline dataset enjoying good coverage of the deployment environment, we tackle robust RL via interactive data collection, where the learner interacts with the training environment only and refines the policy through trial and error. In this robust RL paradigm, two main challenges emerge: managing the distributional robustness while striking a balance between exploration and exploitation during data collection. Initially, we establish that sample-efficient learning without additional assumptions is unattainable owing to the curse of support shift; i.e., the potential disjointedness of the distributional supports between training and testing environments. To circumvent such a hardness result, we introduce the vanishing minimal value assumption to RMDPs with a total-variation distance robust set, postulating that the minimal value of the optimal robust value function is zero. Such an assumption effectively eliminates the support shift issue for RMDPs with a TV distance robust set, and we present an algorithm with a provable sample complexity guarantee. Our work makes the initial step to uncovering the inherent difficulty of robust RL via interactive data collection and sufficient conditions for sample-efficient algorithms with sharp sample complexity.

## 1 Introduction

Reinforcement learning (RL) serves as a framework for addressing complex decision-making problems through iterative interactions with environments. The advancements in deep reinforcement learning have enabled the successful application of the general RL framework across various domains, including mastering strategic games, such as Go (Silver et al., 2017), robotics (Kober et al., 2013), and tuning large language models (LLMs; Ouyang et al. (2022)). The critical factors contributing to these successes encompass not only the potency of deep neural networks and modern deep RL algorithms but also the availability of substantial training data. However, there are scenarios, such as healthcare (Wang et al., 2018) and autonomous driving (Kiran et al., 2021), among others, where collecting data in the target domain is challenging, costly, or even unfeasible.

In such cases, the *sim-to-real transfer* (Kober et al., 2013; Sadeghi and Levine, 2016; Peng et al., 2018; Zhao et al., 2020) becomes a remedy – a process in which the RL agents are trained in some simulated environment and subsequently deployed in real-world settings. Nevertheless, the training

---

*Equal contribution. Email to `miaolu@stanford.edu`, `hanzhong@stu.pku.edu.cn`

38th Conference on Neural Information Processing Systems (NeurIPS 2024).

environment may differ from the real-world environment. Such a discrepancy, also known as the *sim-to-real gap*, will typically result in suboptimal performance of RL agents in real-world applications. One promising strategy to control the impact in performance degradation due to the sim-to-real gap is robust RL (Iyengar, 2005; Pinto et al., 2017; Hu et al., 2022), which aims to learn policies exhibiting strong (i.e. robust) performance under environmental deviations from the training environment, effectively hedging the epistemological uncertainty arising from the differences between the training environment and the unknown testing environments.

A robust RL problem is often formulated within a robust Markov decision process (RMDP) framework, with various types of robust sets characterizing different environmental perturbations. In this robust RL context, prior works have developed algorithms with provable sample complexity guarantees. However, these algorithms typically rely on either a generative model (Yang et al., 2022; Panaganti and Kalathil, 2022; Xu et al., 2023; Shi et al., 2023) or offline data with good coverage of deployment environments (Zhou et al., 2021b; Panaganti et al., 2022; Shi and Chi, 2022; Ma et al., 2022; Blanchet et al., 2023). Notably, the current literature does not explicitly address the *exploration* problem, which stands as one of the fundamental challenges in reinforcement learning through trial-and-error (Sutton and Barto, 2018). Meanwhile, the empirical success of robust RL methods (Pinto et al., 2017; Kuang et al., 2022; Moos et al., 2022) typically relies on reinforcement learning through *interactive data collection* in the training environment, where the agent iteratively and actively interacts with the environment, collecting data, optimizing and robustifying its policy. Given that all the existing literature on robust RL theory relies on a generative model or pre-collected data, it is natural to ask:

> *Can we design a provably sample-efficient robust RL algorithm that relies on*
> *interactive data collection in the training environment?*

Answering the above question faces a fundamental challenge, namely, that during the interactive data collection process, the learner no longer has the oracle control over the training data distributions that are induced by the policy learned through the interaction process. In particular, it could be the case that certain data patterns that are crucial for the policy to be robust across all testing environments are not accessible through interactive data collection, even through a sophisticated design of an exploration mechanism during the interaction process. For example, specific states may not be accessible within the training environment dynamics but could be reached in the testing environment dynamics.

In contrast, previous work has demonstrated that robust RL through a generative model or a pre-collected offline dataset with good coverage does not face such difficulty. In the generative model setup, fortunately, the learner can directly query any state-action pair and obtain the sampled next state from the generator. Intuitively, once the states that could appear in the testing environment trajectory are queried enough times, it is possible to guarantee the performance of the learned policy in testing environments. The situation is similar if one has a pre-collected offline dataset that possesses good coverage of the testing environment. This motivates us to take the initial steps towards answering the above questions regarding robust RL with interactive data collection.

## 1.1 Contributions

In this work, we study robust RL in a finite-horizon RMDP with an $\mathcal{S} \times \mathcal{A}$-rectangular total-variation distance (TV) robust set (see Assumption 2.1 and Definition 2.4) through *interactive data collection*. We give both a fundamental hardness result in the general case and a sample-efficient algorithm within tractable settings. More specifically, our contributions are three folds.

**Fundamental hardness.** We construct a class of hard-to-learn RMDPs (see Example 3.1) and demonstrate that *any* learning algorithm inevitably incurs an $\Omega(\rho \cdot HK)$-online regret (Theorem 3.2) under at least one RMDP instance. Here, $\rho$ signifies the radius of the TV robust uncertainty set, $H$ is the horizon, and $K$ is the number of interactive episodes. This linear regret lower bound underscores the impossibility of sample-efficient robust RL via interactive data collection in general.

**Identifying a tractable case.** Upon close examination of the challenging instance, we recognize that the primary obstacle to achieving sample-efficient learning lies in the *curse of support shift*, i.e., the disjointedness of distributional support between the training environment and the testing environments. In a broader sense, the curse of support shift also refers to the situation when the states appearing in testing environments are extremely hard to arrive in the training environment.

Table 1: Comparison between `OPROVI-TV` and prior results on RMDP with $\mathcal{S} \times \mathcal{A}$-rectangular TV robust sets under various settings (generative model/offline dataset/interactive data collection). For the infinite horizon $\gamma$-discounted RMDPs, we denote $H_\gamma := (1 - \gamma)^{-1}$ as the effective horizon length. In the offline setting, $\mathcal{C}^\star_{\mathrm{rob}}$ and $\mathcal{C}_{\mathrm{full}}$ represent the robust partial coverage coefficient and full coverage coefficient, respectively. In the general case, our lower bound reads intractable, meaning that there exist hard instances where it is impossible to learn the nearly optimal robust policy via a finite number of interactive samples.

| Model Assump. | Algorithm | Data oracle | Sample complexity $\rho \in [0, 1)$ |
|---|---|---|---|
| general case | RPVL (Xu et al., 2023) | generative model | $\widetilde{\mathcal{O}}\left(\frac{H^5 SA}{\varepsilon^2}\right)$ |
| | DRVI (Shi et al., 2023) | generative model | $\widetilde{\mathcal{O}}\left(\frac{\min\{H_\gamma, \rho^{-1}\} H_\gamma^2 SA}{\varepsilon^2}\right)$ |
| | lower bound (Shi et al., 2023) | generative model | $\Omega\left(\frac{\min\{H_\gamma, \rho^{-1}\} H_\gamma^2 SA}{\varepsilon^2}\right)$ |
| | P$^2$MPO (Blanchet et al., 2023) | offline dataset | $\widetilde{\mathcal{O}}\left(\frac{\mathcal{C}^\star_{\mathrm{rob}} H^4 S^2 A}{\varepsilon^2}\right)$ |
| | lower bound (this work) | interactive data collection | intractable |
| "fail-state" assumption | RFQI (Panaganti et al., 2022) | offline dataset | $\widetilde{\mathcal{O}}\left(\frac{\mathcal{C}_{\mathrm{full}} H_\gamma^4 SA}{\rho^2 \varepsilon^2}\right)$ |
| vanishing minimal value (Assumption 4.1) | OPROVI-TV (this work) | interactive data collection | $\widetilde{\mathcal{O}}\left(\frac{\min\{H, \rho^{-1}\} H^2 SA}{\varepsilon^2}\right)$ |

To rule out these pathological instances, we propose the *vanishing minimal value* assumption (Assumption 4.1), positing that the optimal robust value function reaches zero at a specific state. Such an assumption naturally applies to the sparse reward RL paradigm and offers a broader scope compared to the "fail-state" assumption utilized in prior studies on offline RMDP with function approximation (Panaganti et al., 2022). For a comprehensive discussion on this comparison, please see Remark B.3. On the theoretical front, we establish that the vanishing minimal value assumption effectively mitigates the support shift issues between the training and the testing environments (Proposition 4.2), rendering robust RL with interactive data collection feasible for RMDPs with TV robust sets.

**Efficient algorithm with sharp sample complexity.** Under the vanishing minimal value assumption, we develop an algorithm named OPtimistic RObust Value Iteration for TV Robust Set (OPROVI-TV, Algorithm 1), that is capable of finding an $\varepsilon$-optimal robust policy with a total number of

$$\widetilde{\mathcal{O}}\big( \min\{H, \rho^{-1}\} \cdot H^2 SA/\varepsilon^2 \big) \tag{1.1}$$

interactive samples (Theorem 4.3). Here $S$ and $A$ denote the number of states and actions, $\rho$ represents the radius of the TV robust set, and $H$ is the horizon length of each episode. To our best knowledge, this is the first provably sample-efficient algorithm for robust RL with interactive data collection.

According to (1.1), the sample complexity of finding an $\varepsilon$-optimal robust policy decreases as the radius $\rho$ of the robust set increases. When the radius $\rho = 0$, an RMDP reduces to a standard MDP, and the sample complexity (1.1) recovers the minimax-optimal sample complexity for online RL in standard MDPs up to logarithm factors, i.e., $\widetilde{\mathcal{O}}(H^3 SA/\varepsilon^2)$.

In the end, we further extend our algorithm and theory to a new type of RMDPs, $\mathcal{S} \times \mathcal{A}$-rectangular discounted RMDP equipped with robust sets consisting of transition probabilities with bounded ratio to the nominal kernel (See Appendix B.4.2). This newly identified class of RMDPs naturally does not suffer from the support shift issue. It is equivalent to the $\mathcal{S} \times \mathcal{A}$-rectangular RMDP with TV robust set and vanishing minimal value assumption in an appropriate sense due to Proposition 4.2. Consequently, by a clever usage of Algorithm 1, we can also solve this new model sample-efficiently, as shown in Corollary B.5. Such a result echoes our intuition on the curse of support shift.

**Comparison to related works.** Due to the space limit, we only compare with the most related works through Table 1. A detailed discussion of related works is in Appendix A.

## 2 Preliminaries

**Notations.** For a set $\mathcal{X}$, we denote $\Delta(\mathcal{X})$ as the set of probability distributions on $\mathcal{X}$. For a distribution $p \in \Delta(\mathcal{X})$, we define the shorthand for expectation and variance as $\mathbb{E}_{p(\cdot)}[f] := \mathbb{E}_{X \sim p(\cdot)}[f(X)]$ and $\mathbb{V}_{p(\cdot)}[f] = \mathbb{E}_{p(\cdot)}[f^2] - (\mathbb{E}_{p(\cdot)}[f])^2$. Given any set $\mathcal{Q} \subseteq \Delta(\mathcal{X})$, we define the robust expectation

operator as $\mathbb{E}_{\mathcal{Q}}[f] := \inf_{p(\cdot) \in \mathcal{Q}} \mathbb{E}_{X \sim p(\cdot)}[f(X)]$. For any $x, a \in \mathbb{R}$, we denote $(x)_+ = \max\{x, 0\}$ and $x \vee a = \max\{x, a\}$. We use $\mathcal{O}(\cdot)$ to hide absolute constant factors and use $\widetilde{\mathcal{O}}$ to further hide logarithmic factors. For a positive integer $H \in \mathbb{N}_+$, we denote the set $\{1, 2, \ldots, H\}$ by $[H]$.

## 2.1 Robust Markov Decision Processes

We first introduce our underlying model for doing robust RL, the episodic robust Markov decision process (RMDP), denoted by a tuple $(\mathcal{S}, \mathcal{A}, H, P^\star, R, \mathbf{\Phi})$. Here the set $\mathcal{S}$ is the state space and the set $\mathcal{A}$ is the action space, both with finite cardinality. The integer $H$ is the length of each episode. The set $P^\star = \{P_h^\star\}_{h=1}^H$ is the collection of *nominal* transition kernels where $P_h^\star : \mathcal{S} \times \mathcal{A} \mapsto \Delta(\mathcal{S})$. The set $R = \{R_h\}_{h=1}^H$ is the collection of reward functions where $R_h : \mathcal{S} \times \mathcal{A} \mapsto [0, 1]$. For simplicity, we denote $\mathcal{P} = \{P(\cdot|\cdot, \cdot) : \mathcal{S} \times \mathcal{A} \mapsto \Delta(\mathcal{S})\}$ as the space of all possible transition kernels, and we denote $S = |\mathcal{S}|$ and $A = |\mathcal{A}|$. Most importantly and different from standard MDPs, the RMDP is equipped with a mapping $\mathbf{\Phi} : \mathcal{P} \mapsto 2^{\mathcal{P}}$ that characterizes the *robust set* of any transition kernel in $\mathcal{P}$. Formally, for any transition kernel $P \in \mathcal{P}$, we call $\mathbf{\Phi}(P)$ the *robust set* of $P$. One could interpret the nominal transition kernel $P_h^\star$ as the transition of the training environment, while $\mathbf{\Phi}(P_h^\star)$ contains all possible transitions of the testing environments.

Given an RMDP $(\mathcal{S}, \mathcal{A}, H, P^\star, R, \mathbf{\Phi})$, we consider using a Markovian policy to make decisions. A Markovian decision policy (or simply, policy) is defined as $\pi = \{\pi_h\}_{h=1}^H$ with $\pi_h : \mathcal{S} \mapsto \Delta(\mathcal{A})$ for each step $h \in [H]$. To measure the performance of a policy $\pi$ in the RMDP, we introduce its *robust value function*, defined as for any $(s, a) \in \mathcal{S} \times \mathcal{A}$,

$$V_{h,P^\star,\mathbf{\Phi}}^\pi(s) := \inf_{\widetilde{P}_h \in \mathbf{\Phi}(P_h^\star), 1 \le h \le H} \mathbb{E}_{\{\widetilde{P}_h\}_{h=1}^H, \{\pi_h\}_{h=1}^H} \left[ \sum_{i=h}^H R_i(s_i, a_i) \,\middle|\, s_h = s \right],$$

$$Q_{h,P^\star,\mathbf{\Phi}}^\pi(s,a) := \inf_{\widetilde{P}_h \in \mathbf{\Phi}(P_h^\star), 1 \le h \le H} \mathbb{E}_{\{\widetilde{P}_h\}_{h=1}^H, \{\pi_h\}_{h=1}^H} \left[ \sum_{i=h}^H R_i(s_i, a_i) \,\middle|\, s_h = s, a_h = a \right].$$

Here the expectation is taken w.r.t. the state-action trajectories induced by policy $\pi$ under the transition $\widetilde{P}$. One can also extend the definition of the robust value functions in terms of any collection of transition kernel $P = \{P_h\}_{h=1}^H \subset \mathcal{P}$ as $V_{h,P,\mathbf{\Phi}}^\pi$ and $Q_{h,P,\mathbf{\Phi}}^\pi$, which we usually use in the sequel.

Among all the policies, we define the optimal robust policy $\pi^\star$ as the policy that can maximize the robust value function at the initial time step $h = 1$, i.e.,

$$\pi^\star \in \operatorname*{argmax}_{\pi = \{\pi_h\}_{h=1}^H} V_{1,P^\star,\mathbf{\Phi}}^\pi(s_1), \quad \forall s_1 \in \mathcal{S}. \tag{2.1}$$

In other words, the optimal robust policy $\pi^\star$ maximizes the worst case expected total rewards in all possible testing environments. For simplicity and without loss of generality, we assume in the sequel that the initial state $s_1 \in \mathcal{S}$ is fixed. Our results could be directly generalized to $s_1 \sim p_0(\cdot) \in \Delta(\mathcal{S})$. Similarly, we can also define the optimal robust policy associated with a given stochastic process defined through any collection of transition kernels $P = \{P_h\}_{h=1}^H \subset \mathcal{P}$ in the same way as (2.1). We denote the optimal robust value functions associated with $P$ as $V_{h,P,\mathbf{\Phi}}^\star$ and $Q_{h,P,\mathbf{\Phi}}^\star$ respectively.

$\mathcal{S} \times \mathcal{A}$-**rectangularity and robust Bellman equations.** We consider robust sets $\mathbf{\Phi}$ that have the $\mathcal{S} \times \mathcal{A}$-rectangular structure (Iyengar, 2005). which requires that the robust set is decoupled and independent across different $(s, a)$-pairs. This kind of structure results in a dynamic programming representation of the robust value functions (efficient planning), and is thus commonly adopted in the literature of distributionally robust RL. More specifically, we assume the following.

**Assumption 2.1** ($\mathcal{S} \times \mathcal{A}$-rectangularity). *We assume that the mapping $\mathbf{\Phi}$ satisfies for any transition kernel $P \in \mathcal{P}$, the robust set $\mathbf{\Phi}(P)$ is in the form of*

$$\mathbf{\Phi}(P) = \bigotimes_{(s,a) \in \mathcal{S} \times \mathcal{A}} \mathcal{P}(s, a; P), \quad where \quad \mathcal{P}(s, a; P) \subseteq \Delta(\mathcal{S}).$$

Under above Assumption 2.1, we have the so-called robust Bellman equation (Iyengar, 2005; Blanchet et al., 2023) which gives a dynamic programming representation of robust value functions.

**Proposition 2.2** (Robust Bellman equation). *Under Assumption 2.1, for any transition* $P = \{P_h\}_{h=1}^H \subseteq \mathcal{P}$ *and any policy* $\pi = \{\pi_h\}_{h=1}^H$ *with* $\pi_h : \mathcal{S} \mapsto \Delta(\mathcal{A})$, *it holds that*

$$V_{h,P,\Phi}^\pi(s) = \mathbb{E}_{\pi_h(\cdot|s)}\big[Q_{h,P,\Phi}^\pi(s,\cdot)\big], \quad Q_{h,P,\Phi}^\pi(s,a) = R_h(s,a) + \mathbb{E}_{\mathcal{P}(s,a;P_h)}\big[V_{h+1,P,\Phi}^\pi\big].$$

Regarding the robust value functions of the optimal robust policy, we also have the following dynamic programming solution which plays a key role in our algorithm design and theoretical analysis.

**Proposition 2.3** (Robust Bellman optimal equation). *Under Assumption 2.1, for any* $P = \{P_h\}_{h=1}^H \subseteq \mathcal{P}$, *the robust value functions of any optimal robust policy of* $P$ *satisfies that,*

$$V_{h,P,\Phi}^\star(s) = \max_{a\in\mathcal{A}} Q_{h,P,\Phi}^\star(s,a), \quad Q_{h,P,\Phi}^\star(s,a) = R_h(s,a) + \mathbb{E}_{\mathcal{P}(s,a;P_h)}\big[V_{h+1,P,\Phi}^\star\big].$$

*Taking* $\pi_h^\star(\cdot|s) = \mathrm{argmax}_{a\in\mathcal{A}}\, Q_{h,P,\Phi}^\star(s,a)$, *then* $\pi^\star = \{\pi_h^\star\}_{h=1}^H$ *is optimal robust policy under* $P$.

**Total-variation distance robust set.** In Assumption 2.1, the robust set $\mathcal{P}(s,a;P)$ is often modeled as a "distribution ball" centered at $P(\cdot|s,a)$. In this paper, we mainly consider this type of robust sets specified by a *total-variation distance* ball. We put it in the following definition.

**Definition 2.4** (Total-variation distance robust set). *Total-variation distance robust set is defined as*

$$\mathcal{P}_\rho(s,a;P) := \Big\{\widetilde{P}(\cdot) \in \Delta(\mathcal{S}) : D_{\mathrm{TV}}\big(\widetilde{P}(\cdot)\big\|P(\cdot|s,a)\big) \le \rho\Big\},$$

*for some* $\rho \in [0,1)$, *where* $D_{\mathrm{TV}}(\cdot\|\cdot)$ *denotes the total variation distance defined as*

$$D_{\mathrm{TV}}\big(p(\cdot)\|q(\cdot)\big) := \frac{1}{2}\sum_{s\in\mathcal{S}}\big|p(s) - q(s)\big|, \quad \forall p(\cdot), q(\cdot) \in \Delta(\mathcal{S}). \tag{2.2}$$

The TV robust set has recently been extensively studied by Yang et al. (2022); Panaganti and Kalathil (2022); Panaganti et al. (2022); Xu et al. (2023); Blanchet et al. (2023); Shi et al. (2023), which all focus on robust RL with a generative model or with a pre-collected offline dataset. More importantly, we emphasize that by (2.2) in Definition 2.4, we *do not* define the TV distance through the notion of $f$-divergence which requires that the distribution $p$ is absolute continuous w.r.t. $q$, as is generally adopted by the above previous works on learning RMDPs with TV robust sets. According to (2.2), we *allow $p$ to have a different support than $q$*. That is, there might exist an $s \in \mathcal{S}$ such that $p(s) > 0$ and $q(s) = 0$. Given that, the TV robust set in Definition 2.4 could contain transition probabilities that have different supports than the nominal transition probability $P^\star(\cdot|s,a)$.

An essential property of the TV robust set is that the robust expectation involved in the robust Bellman equations (Propositions 2.2 and 2.3) has a duality representation that only uses the expectation under the nominal transition kernel, as is shown in the following theorem and proved in Appendix C.1.

**Proposition 2.5** (Strong duality representation). *Under Definition 2.4, the following duality representation for the robust expectation holds, for any* $V : \mathcal{S} \mapsto [0, H]$ *and* $P_h : \mathcal{S} \times \mathcal{A} \mapsto \Delta(\mathcal{S})$,

$$\mathbb{E}_{\mathcal{P}_\rho(s,a;P_h)}\big[V\big] = \sup_{\eta\in[0,H]}\Big\{-\mathbb{E}_{P_h(\cdot|s,a)}\big[(\eta - f)_+\big] - \frac{\rho}{2}\cdot\Big(\eta - \min_{s\in\mathcal{S}}V(s)\Big)_+ + \eta\Big\}. \tag{2.3}$$

**Value gap between maximum and minimum.** Finally, another useful property of the robust value functions of an RMDP with TV robust sets is a fine characterization of the gap between the maximum and the minimum of the robust value function, which is first identified and utilized by Shi et al. (2023) for an infinite horizon RMDP with TV robust sets. In this work, we prove and use a similar result for the finite horizon case, concluded in the following proposition. The proof is in Appendix C.2.

**Proposition 2.6** (Gap between maximum and minimum). *Under Assumption 2.1 with the robust set specified by Definition 2.4, the robust value functions satisfies that*

$$\max_{(s,a)\in\mathcal{S}\times\mathcal{A}} Q_{h,P,\Phi}^\pi(s,a) - \min_{(s,a)\in\mathcal{S}\times\mathcal{A}} Q_{h,P,\Phi}^\pi(s,a) \le \min\big\{H, \rho^{-1}\big\},$$

$$\max_{s\in\mathcal{S}} V_{h,P,\Phi}^\pi(s) - \min_{s\in\mathcal{S}} V_{h,P,\Phi}^\pi(s) \le \min\big\{H, \rho^{-1}\big\},$$

*for any transition* $P = \{P_h\}_{h=1}^H \subset \mathcal{P}$, *any policy* $\pi$, *and any step* $h \in [H]$.

## 2.2 Robust RL with Interactive Data Collection

We study how to learn the optimal robust policy $\pi^\star$ in (2.1) from interactive data collection. Specifically, the learner is required to interact with *only* the *training environment*, i.e., $P^\star$, for some $K \in \mathbb{N}$ episodes. In each episode $k$, the learner adopts a policy $\pi^k$ to interact with the training environment $P^\star$ and to collect data. When the $k$-th episode ends, the learner updates its policy to $\pi^{k+1}$ based on historical data and proceeds to the subsequent $(k+1)$-th episode. The process ends after $K$ episodes.

**Sample complexity.** We use the notion of *sample complexity* as the key evaluation metric. For any given algorithm and predetermined accuracy level $\varepsilon > 0$, the sample complexity is the minimum number of episodes $K$ required for the algorithm to output an $\varepsilon$-optimal robust policy $\widehat{\pi}$ satisfying $V^\star_{1,P^\star,\mathbf{\Phi}}(s_1) - V^{\widehat{\pi}}_{1,P^\star,\mathbf{\Phi}}(s_1) \leq \varepsilon$. The goal is to design algorithms whose sample complexity has small or even optimal dependence on the problem parameters $S, A, H, \rho$, and $1/\varepsilon$.

**Online regret.** Another evaluation metric that is related to the minimization of sample complexity is the *online regret*, which is the cumulative difference between the optimal robust policy $\pi^\star$ and the executed policies $\{\pi^k\}_{k=1}^K$. Formally, we define $\mathrm{Regret}_{\mathbf{\Phi}}(K) := \sum_{k=1}^K V^\star_{1,P^\star,\mathbf{\Phi}}(s_1) - V^{\pi^k}_{1,P^\star,\mathbf{\Phi}}(s_1)$. The goal is to design algorithms that can achieve a sublinear-in-$K$ regret with small dependence on $S, A, H, \rho$. It turns out that any sublinear-regret algorithm can be easily converted to a polynomial-sample complexity algorithm by applying the standard online-to-batch conversion (Jin et al., 2018).

## 3 A Hardness Result: The Curse of Support Shift

Unfortunately, we show in this section that in general such a problem of robust RL with online data collection is *impossible* – there exists a simple class of two RMDPs such that an $\Omega(K)$-online regret lower bound exists. However, previous works on robust RL with a generative model or offline data with good coverage do provide sample-efficient ways to find the optimal robust policy for this class of RMDPs. This is a separation between robust RL with interactive data collection and generative model/offline data. Please see also Figure 1 for an illustration of the example.

**Example 3.1** (Hard example of robust RL with interactive data collection). *Consider two RMDPs $\mathcal{M}_0$ and $\mathcal{M}_1$ which only differ in their nominal transition kernels. The state space is $\mathcal{S} = \{s_{\mathrm{good}}, s_{\mathrm{bad}}\}$, and the action space is $\mathcal{A} = \{0, 1\}$. The horizon length $H = 3$. The reward function $R$ always is $1$ at the good state $s_{\mathrm{good}}$ and is $0$ at the bad state $s_{\mathrm{bad}}$, i.e.,*

$$R_h(s,a) = \begin{cases} 1, & s = s_{\mathrm{good}} \\ 0, & s = s_{\mathrm{bad}} \end{cases}, \quad \forall (a,h) \in \mathcal{A} \times [H].$$

*For the good state $s_{\mathrm{good}}$, the next state is always $s_{\mathrm{good}}$. For the bad state $s_{\mathrm{bad}}$, there is a chance to get to the good state $s_{\mathrm{good}}$, with the transition probability depending on the action it takes. Formally,*

$$P_h^{\star,\mathcal{M}_\theta}(s_{\mathrm{good}}|s_{\mathrm{good}}, a) = 1, \quad \forall (a,h) \in \mathcal{A} \times \{1, 2\}, \quad \forall \theta \in \{0, 1\},$$

$$P_2^{\star,\mathcal{M}_\theta}(s_{\mathrm{good}}|s_{\mathrm{bad}}, a) = \begin{cases} p, & a = \theta \\ q, & a = 1 - \theta \end{cases}, \quad \forall \theta \in \{0, 1\},$$

*where $p, q$ are two constants satisfying $0 < q < p < 1$. Intuitively, when at the bad state, the optimal action would result in a higher transition probability $p$ to the good state than the transition probability $q$ induced by the other action. Finally, we consider the robust set being specified by a total-variation distance ball centered at the nominal transition kernel, that is, for any $P$,*

$$\mathbf{\Phi}(P) = \bigotimes_{(s,a) \in \mathcal{S} \times \mathcal{A}} \mathcal{P}_\rho(s, a; P), \; \mathcal{P}_\rho(s, a; P) = \left\{ \widetilde{P}(\cdot) \in \Delta(\mathcal{S}) : D_{\mathrm{TV}}\big(\widetilde{P}(\cdot) \big\| P(\cdot|s,a)\big) \leq \rho \right\}, \text{(3.1)}$$

*where $\rho \in [0, q]$ is the parameter characterizing the size of the robust set. We set $s_1 = s_{\mathrm{good}}$.*

For this class of RMDPs, we have the following hardness result for doing robust RL with interactive data collection, an $\Omega(\rho \cdot K)$-online regret lower bound. The proof is in Appendix D.1.

**Theorem 3.2** (Hardness result (based on Example 3.1)). *There exists two RMDPs $\{\mathcal{M}_0, \mathcal{M}_1\}$, the following regret lower bound holds,*

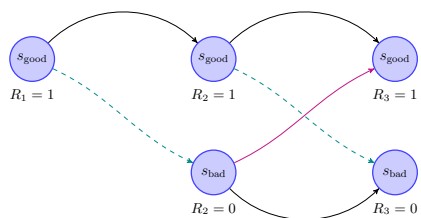

Figure 1: Illustration of the hard example in Example 3.1. The solid lines represent possible transitions of the nominal transition kernel. The dashed lines represent the transitions induced by the worst case transition kernel in the robust set. The red solid line represents the transition where the two RMDP instances differ in that different actions lead to higher transition probability from $s_{\text{bad}}$ to $s_{\text{good}}$. We notice that when starting from $s_1 = s_{\text{good}}$, the nominal transition kernel keeps the agent at $s_{\text{good}}$ and no information at $s_{\text{bad}}$ is revealed.

$$\inf_{\mathcal{ALG}} \sup_{\theta \in \{0,1\}} \mathbb{E}\left[\text{Regret}^{\mathcal{M}_\theta, \mathcal{ALG}}_{\boldsymbol{\Phi}}(K)\right] \geq \Omega(\rho \cdot HK),$$

where $\text{Regret}^{\mathcal{M}_\theta, \mathcal{ALG}}_{\boldsymbol{\Phi}}(K)$ *refers to the online regret of algorithm* $\mathcal{ALG}$ *for RMDP* $\mathcal{M}_\theta$.

The reason why any algorithm fails for this class of RMDPs is the *support shift* of the worst-case transition kernel. In robust RL, the performance of a policy $\pi$ is evaluated via the robust expected total rewards, or equivalently, the expected return under the most adversarial transition kernel $P^{\dagger,\pi}$. In such an example, as we explicitly show in the proof, when in the good state $s_{\text{good}}$, the worst-case transition kernel $P^{\dagger,\pi}$ would transit the state to $s_{\text{bad}}$ with a constant probability $\rho > 0$. But the state $s_{\text{bad}}$ is out of the scope of the data collection process because starting from $s_1 = s_{\text{good}}$ the nominal transition kernel always transits the state to $s_{\text{good}}$. As a result, the performance of the learned policy at the bad state $s_{\text{bad}}$ is not guaranteed, and inevitably incurs an $\Omega(\rho \cdot K)$-lower bound of regret, a hardness result. Furthermore, by strategically constructing RMDPs with the horizon $3H$ based on Example 3.1, we can derive a lower bound of $\Omega(\rho \cdot HK)$. See Appendix B.3 for more discussions.

## 4    A Solvable Case, Efficient Algorithm, and Sharp Analysis

Motivated by the hard instance (Example 3.1), we now investigate a special subclass of RMDPs with $\mathcal{S} \times \mathcal{A}$-rectangular total variation robust set that we show allows for doing sample-efficient robust RL through interactive data collection. In Section 4.1, we introduce the assumption we impose on the RMDP. We propose our algorithm design in Section 4.2, with theoretical analysis in Section 4.3.

### 4.1    Vanishing Minimal Value: Eliminating Support Shift

To overcome the difficulty of support shift identified in Section 3, we make the following *vanishing minimal value* assumption on the underlying RMDP.

**Assumption 4.1** (Vanishing minimal value)**.** *We assume that the underlying RMDP satisfies that* $\min_{s \in \mathcal{S}} V^\star_{1,P^\star,\boldsymbol{\Phi}}(s) = 0$. *Also, WLOG, we assume that the initial state* $s_1 \notin \arg\min_{s \in \mathcal{S}} V^\star_{1,P^\star,\boldsymbol{\Phi}}(s)$.

Assumption 4.1 imposes that the minimal robust expected total rewards over all possible initial states is 0. Assuming that the initial state $s_1 \notin \arg\min_{s \in \mathcal{S}} V^\star_{1,P^\star,\boldsymbol{\Phi}}(s)$ avoids making the problem trivial. A close look at Assumption 4.1 actually gives that the minimal robust value function of any policy $\pi$ at any step is zero, that is, $\min_{s \in \mathcal{S}} V^\pi_{h,P^\star,\boldsymbol{\Phi}}(s) = 0$ for any policy $\pi$ and any step $h \in [H]$. With this observation, the following proposition explains why this assumption helps to overcome the difficulty, with the proof of the proposition in Appendix C.3.

**Proposition 4.2** (Equivalent expression of TV robust set with vanishing minimal value)**.** *For any function* $V : \mathcal{S} \mapsto [0, H]$ *with* $\min_{s \in \mathcal{S}} V(s) = 0$, *we have that*

$$\mathbb{E}_{\mathcal{P}_\rho(s,a;P^\star_h)}[V] = \rho' \cdot \mathbb{E}_{\mathcal{B}_{\rho'}(s,a;P^\star_h)}[V], \quad with \quad \rho' = 1 - \frac{\rho}{2} > 0,$$

*where the TV robust set* $\mathcal{P}_\rho(s, a; P^\star_h)$ *is defined in* (3.1) *and the set* $\mathcal{B}_{\rho'}(s, a; P^\star_h)$ *is defined as*[2]

$$\mathcal{B}_{\rho'}(s, a; P^\star_h) = \left\{ \widetilde{P}(\cdot) \in \Delta(\mathcal{S}) : \sup_{s' \in \mathcal{S}} \frac{\widetilde{P}(s')}{P^\star_h(s'|s,a)} \leq \frac{1}{\rho'} \right\}.$$

---

[2]Here we implicitly define $\frac{0}{0} = 0$ and $\frac{a}{0} = \infty$ for any $a > 0$.

---

**Algorithm 1** OPtimistic RObust Value Iteration for TV Robust Set (OPROVI-TV)

1: **Initialize:** dataset $\mathbb{D} = \emptyset$.
2: **for** episode $k = 1, \cdots, K$ **do**
3:     Training environment transition estimation:
4:     Update the count functions $N_h^k(s,a,s')$ and $N_h^k(s,a)$ based on $\mathbb{D}$.
5:     Calculate the transition kernel estimator $\widehat{P}_h^k$ as $N_h^k(s,a,s')/(N_h^k(s,a) \vee 1)$.
6:     Optimistic robust planning:
7:     Set $\overline{V}_{H+1}^k = \underline{V}_{H+1}^k = 0$.
8:     **for** step $h = H, \cdots, 1$ **do**
9:         Set $\overline{Q}_h^k(\cdot, \cdot)$ and $\underline{Q}_h^k(\cdot, \cdot)$ as (4.2) and (4.3), with $\mathrm{bonus}_h^k(\cdot, \cdot)$ defined in (4.5).
10:         Set $\pi_h^k(\cdot|\cdot) = \mathrm{argmax}_{a \in \mathcal{A}} \overline{Q}_h^k(\cdot, a)$, $\overline{V}_h^k(\cdot) = \mathbb{E}_{\pi_h^k(\cdot|\cdot)}[\overline{Q}_h^k(\cdot, \cdot)]$, and $\underline{V}_h^k(\cdot) = \mathbb{E}_{\pi_h^k(\cdot|\cdot)}[\underline{Q}_h^k(\cdot, \cdot)]$.
11:     **end for**
12:     Execute the policy in training environment and collect data:
13:     Receive the initial state $s_1^k \in \mathcal{S}$.
14:     **for** step $h = 1, \cdots, H$ **do**
15:         Take action $a_h^k \sim \pi_h^k(\cdot|s_h^k)$, observe reward $R_h(s_h^k, a_h^k)$ and the next state $s_{h+1}^k$.
16:     **end for**
17:     Set $\mathbb{D}$ as $\mathbb{D} \cup \{(s_h^k, a_h^k, s_{h+1}^k)\}_{h=1}^H$.
18: **end for**
19: **Output:** Randomly (uniformly) return a policy from $\{\pi^k\}_{k=1}^K$.

---

As Proposition 4.2 indicates, under Assumption 4.1, the robust Bellman equations (Propositions 2.2 and 2.3) at step $h \in [H]$ is equivalent to taking an infimum over another robust set $\mathcal{B}_{\rho'}(s, a; P_h^\star)$ that shares the *same* support as the nominal transition kernel $P^\star(\cdot|s, a)$, discounted by a constant $\rho' < 1$. Intuitively, this new robust set rules out the difficulty originated in unseen states in training environments and the discount factor $\rho'$ hedges the difficulty from prohibitively small probability of reaching certain states that may appear often in the testing environments. This renders robust RL with interactive data collection possible. See Appendix B.4.1 for discussions/examples of Assumption 4.1.

### 4.2 Algorithm Design: OPROVI-TV

In this section, we propose our algorithm that solves robust RL with interactive data collection for RMDPs with $\mathcal{S} \times \mathcal{A}$-rectangular total-variation (TV) robust sets (Assumption 2.1 and Definition 2.4) and satisfying the vanishing minimal value assumption (Assumption 4.1). Our algorithm, OPtimistic RObust Value Iteration for TV Robust Set (OPROVI-TV, Algorithm 1), can automatically balance exploitation and exploration during the interactive data collecting process while managing the distributional robustness of the learned policy. The full algorithm OPROVI-TV is given in Algorithm 1.

**Step I: Training Environment Transition Estimation (Line 3 to 5).** At the beginning of each episode $k \in [K]$, we maintain an estimate of the transition kernel $P^\star$ of the training environment by using the historical data $\mathbb{D} = \{(s_h^\tau, a_h^\tau, s_{h+1}^\tau)\}_{\tau=1,h=1}^{k-1,H}$ collected from the interaction with the training environment. Specifically, we simply adopt a vanilla empirical estimator, defined as $\widehat{P}_h^k(s'|s, a) = N_h^k(s, a, s')/(N_h^k(s, a) \vee 1)$ for any $(s, a, h, s') \in \mathcal{S} \times \mathcal{A} \times \mathcal{S} \times [H]$, where the count functions $N_h^k(s, a, s')$ and $N_h^k(s, a)$ are calculated based on the current dataset $\mathbb{D}$ by $N_h^k(s, a, s') = \sum_{\tau=1}^{k-1} \mathbf{1}\{(s_h^\tau, a_h^\tau, s_{h+1}^\tau) = (s, a, s')\}$ and $N_h^k(s, a) = \sum_{s' \in \mathcal{S}} N_h^k(s, a, s')$ for any $(s, a, h, s') \in \mathcal{S} \times \mathcal{A} \times \mathcal{S} \times [H]$. This just coincides with the transition estimator adopted by existing non-robust online RL algorithms (Auer et al., 2008; Azar et al., 2017; Zhang et al., 2021).

**Step II: Optimistic Robust Planning (Line 6 to 11).** Given $\widehat{P}^k(\cdot|\cdot, \cdot)$ that estimates the training environment, we perform an optimistic robust planning to construct the policy $\pi^k$ to execute. Basically, the optimistic robust planning follows the robust Bellman optimal equation (Proposition 2.3) to approximate the optimal robust policy, but differs in that it maintains an upper bound and a lower bound of the optimal robust value function and chooses the policy that maximizes the optimistic

estimate to incentivize exploration during data collection. Here the purpose of maintaining the lower bound estimate is to facilitate the construction of the variance-aware optimistic bonus (see following), which helps to sharpen our theoretical analysis.

▷ *Simplifying the robust expectation.* To utilize the vanishing minimal value condition (Assumption 4.1), we take a closer look into the robust Bellman equation. By strong duality (Proposition 2.5), the robust expectation $\mathbb{E}_{\mathcal{P}_\rho(s,a;P)}[V]$ for any $V \in [0, H]$ satisfying $\min_{s \in \mathcal{S}} V(s) = 0$ is equivalent to

$$\mathbb{E}_{\mathcal{P}_\rho(s,a;P)}\big[V\big] = \sup_{\eta \in [0,H]} \left\{ -\mathbb{E}_{P(\cdot|s,a)}\Big[(\eta - V)_+\Big] + \left(1 - \frac{\rho}{2}\right) \cdot \eta \right\}. \tag{4.1}$$

Consequently, with a slight abuse of the notation, in the remaining of the paper, we **re-define** the operator $\mathbb{E}_{\mathcal{P}_\rho(s,a;P)}[V]$ as the right hand side of (4.1). Due to Assumption 4.1, the robust Bellman (optimal) equation (Proposition 2.2 and Proposition 2.3) still holds under this new definition.

▷ *Optimistic robust planning.* With this in mind, the optimistic robust planning goes as follows. Starting from $\overline{V}_{H+1}^k = \underline{V}_{H+1}^k = 0$, we recursively define that for any $(s, a) \in \mathcal{S} \times \mathcal{A}$,

$$\overline{Q}_h^k(s,a) = \min\left\{ R_h(s,a) + \mathbb{E}_{\mathcal{P}_\rho(s,a;\widehat{P}_h^k)}\Big[\overline{V}_{h+1}^k\Big] + \texttt{bonus}_h^k(s,a), \min\left\{H, \rho^{-1}\right\}\right\}, \tag{4.2}$$

$$\underline{Q}_h^k(s,a) = \max\left\{ R_h(s,a) + \mathbb{E}_{\mathcal{P}_\rho(s,a;\widehat{P}_h^k)}\Big[\overline{V}_{h+1}^k\Big] - \texttt{bonus}_h^k(s,a), 0\right\}, \tag{4.3}$$

where the robust expectation $\mathbb{E}_{\mathcal{P}_\rho(s,a;\widehat{P}_h^k)}$ follows the definition in the right hand side of (4.1), and the bonus function $\texttt{bonus}_h^k(s,a) \geq 0$ is defined later. Here we truncate the optimistic estimate $\overline{Q}_h^k$ via the upper bound $\min\{H, \rho^{-1}\}$ of the true optimal robust value function $Q_{h,P^\star,\Phi}^\star$. This truncation arises from the combined implication of Proposition 2.6 and the fact that $\min_{(s,a) \in \mathcal{S} \times \mathcal{A}} Q_{h,P^\star,\Phi}^\star(s,a) = 0$ under Assumption 4.1. As we establish in Lemma E.2, $\overline{Q}_h^k$ and $\underline{Q}_h^k$ form upper and lower bounds for $Q_{h,P^\star,\Phi}^\star$ and $Q_{h,P^\star,\Phi}^{\pi^k}$ under a proper choice of the bonus. After performing (4.2) and (4.3), we choose the data collection policy $\pi_h^k$ to be the optimal policy with respect to the optimistic estimator $\overline{Q}_h^k$ and define $\overline{V}_h^k$ and $\underline{V}_h^k$ accordingly by

$$\pi_h^k(\cdot|\cdot) = \underset{a \in \mathcal{A}}{\operatorname{argmax}}\, \overline{Q}_h^k(\cdot, a), \quad \overline{V}_h^k(s) = \mathbb{E}_{\pi_h^k(\cdot|s)}\Big[\overline{Q}_h^k(s, \cdot)\Big], \quad \underline{V}_h^k(s) = \mathbb{E}_{\pi_h^k(\cdot|s)}\Big[\underline{Q}_h^k(s, \cdot)\Big]. \tag{4.4}$$

We remark that the purpose of maintaining the lower bound estimate (4.3) is to facilitate the construction of the bonus and to help sharpen our theoretical analysis. The construction of the policy $\pi^k$ is still based on the optimistic estimator, which is why we call it optimistic robust planning. As indicated by theory, the optimistic robust planning can effectively guide the policy to explore uncertain *robust value function* estimates, striking a balance between exploration and exploitation while managing distributional robustness.

▷ *Bonus function.* The bonus function $\texttt{bonus}_h^k(s,a)$ is a Bernstein-style bound defined as

$$\sqrt{\frac{\mathbb{V}_{\widehat{P}_h^k(\cdot|s,a)}\Big[\big(\overline{V}_{h+1}^k + \underline{V}_{h+1}^k\big)/2\Big]c_1\iota}{N_h^k(s,a) \vee 1}} + \frac{2\mathbb{E}_{\widehat{P}_h^k(\cdot|s,a)}\Big[\overline{V}_{h+1}^k - \underline{V}_{h+1}^k\Big]}{H} + \frac{c_2 H^2 S\iota}{N_h^k(s,a) \vee 1} + \frac{1}{\sqrt{K}}, \tag{4.5}$$

where $\iota = \log(S^3 AH^2 K^{3/2}/\delta)$, $c_1, c_2 > 0$ are absolute constants, and $\delta$ signifies a pre-selected fail probability. Under (4.5), $\overline{Q}_h^k$ and $\underline{Q}_h^k$ become upper and lower bounds of the optimal robust value functions (Lemma E.2). More importantly, the bonus (4.5) is carefully designed for robust value functions such that the summation of this bonus term (especially the leading variance term in (4.5)) over time steps is well controlled, for which we also develop new analysis methods. This is critical for obtaining a sharp sample complexity of Algorithm 1.

## 4.3 Theoretical Guarantees

This section establishes the online regret and the sample complexity of `OPROVI-TV` (Algorithm 1). Our main result is the following, upper bounding the online regret of Algorithm 1, proved in Appendix E.

**Theorem 4.3** (Online regret of `OPROVI-TV`). *Given an RMDP with $\mathcal{S} \times \mathcal{A}$-rectangular total-variation robust set of radius $\rho \in [0,1)$ (Assumption 2.1 and Definition 2.4) satisfying Assumptions 4.1, choosing the bonus function as (4.5) with sufficiently large $c_1, c_2 > 0$, then with probability at least $1 - \delta$, Algorithm 1 satisfies*

$$\text{Regret}_{\mathbf{\Phi}}(K) \leq \mathcal{O}\left(\sqrt{\min\left\{H, \rho^{-1}\right\} H^2 SAK\iota'}\right),$$

*where $\iota' = \log^2(SAHK/\delta)$ and $\mathcal{O}(\cdot)$ hides absolute constants and lower order terms in $K$.*

Theorem 4.3 shows that Algorithm 1 enjoys a sublinear online regret of $\widetilde{\mathcal{O}}(\sqrt{K})$, meaning that it is able to approximately find the optimal robust policy through interactive data collection. This is in contrast with the general hardness result in Section 3 where sample-efficient learning is impossible in the worst case. Thus we show the effectiveness of the minimal value assumption for robust RL with interactive data collection. As a corollary, we have the following sample complexity bound for Algorithm 1. It is obtained directly from Theorem 4.3 and a standard online to batch conversion.

**Corollary 4.4** (Sample complexity of `OPROVI-TV`). *Under the same setup and conditions as in Theorem 4.3, with probability at least $1 - \delta$, Algorithm 1 can output an $\varepsilon$-optimal policy within*

$$\mathcal{O}\left(\min\left\{H, \rho^{-1}\right\} H^2 SA\iota''/\varepsilon^2\right) \tag{4.6}$$

*episodes, where $\iota'' = \log(SAH/\varepsilon\delta)$ and $\mathcal{O}(\cdot)$ hides absolute constants. The valid range of $\varepsilon$ satisfies $\varepsilon \in (0, c \cdot \min\{1, 1/(\rho H)\}]$ for some constant $c > 0$.*

We compare the sample complexity (4.6) with prior arts on non-robust online RL and robust RL with a generative model. On the one hand, (4.6) with $\rho = 0$ equals to $\widetilde{\mathcal{O}}(H^3 SA/\varepsilon^2)$, matching the minimax sample complexity lower bound for online RL in non-robust MDPs (Azar et al., 2017). This means that our algorithm design can naturally handle non-robust MDPs as a special case (please also see Remark B.4 for why one can reduce Algorithm 1 to general non-robust MDPs under Assumption 4.1). On the other hand, the previous work of Shi et al. (2023) for robust RL in infinite horizon RMDPs with a TV robust set and a generative model showcases a minimax optimal sample complexity of

$$\widetilde{\mathcal{O}}\left(\min\left\{H_\gamma, \rho^{-1}\right\} H_\gamma^2 SA/\varepsilon^2\right),$$

for $\rho \in [0,1)$, where we $H_\gamma := 1/(1-\gamma)$ is the effective horizon of the infinite $\gamma$-discounted RMDPs. As a result, the sample complexity (4.6) of Algorithm 1 matches their result. We highlight that our algorithm does not rely on a generative model and operates purely through interactive data collection.

**Extensions of Algorithm 1 and its theory.** In Appendix B.4.2, we extend Algorithm 1 to solve a new type of RMDPs whose robust set consists of transition probabilities with bounded ratio to the nominal kernel. The intuition is because it is equivalent to the $\mathcal{S} \times \mathcal{A}$-rectangular RMDP with a TV robust set and vanishing minimal value assumption in an appropriate sense (Proposition 4.2) Consequently, by a clever usage of Algorithm 1, we can also solve this new model sample-efficiently, as is shown in Corollary B.5. Such a result echoes our intuition on the curse of support shift.

## 5 Conclusions and future works

This work shows that in the absence of any structural assumptions, robust RL via interactive data collection necessarily induces a linear regret lower bound in the worst case due to the curse of support shift. Under the vanishing minimal value assumption, an assumption that is able to effectively rule out the potential support shift issues for RMDPs with a TV robust set, we propose a sample-efficient robust RL algorithm for those RMDPs with sharp analysis. Potential future works include extending to function approximation settings and other types of robust sets. See discussion in Appendix B.5.

## Acknowledgments and Disclosure of Funding

The material in this paper is based upon work supported by the Air Force Office of Scientific Research under award number FA9550-20-1-0397. Additional support is gratefully acknowledged from NSF 1915967, 2118199, 2229012, 2312204. The authors would like to thank the anonymous reviewers for their helpful comments. The authors would also like to thank Pan Xu and Zhishuai Liu for their feedback on an early draft of this work.

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

# A Related Works

We give a detailed discussion on the related works in this section.

**Robust reinforcement learning in robust Markov decision processes.** Robust RL is usually framed as a robust Markov decision process (RMDP) (Iyengar, 2005; El Ghaoui and Nilim, 2005; Wiesemann et al., 2013). There is a long line of work dedicated to the problem of how to solve for the optimal robust policy of a given RMDP, i.e., planning (Iyengar, 2005; El Ghaoui and Nilim, 2005; Xu and Mannor, 2010; Wang and Zou, 2022; Wang et al., 2022; Kuang et al., 2022; Wang et al., 2023a; Yu et al., 2023; Zhou et al., 2023; Li and Lan, 2023; Wang et al., 2023c; Ding et al., 2024). Recently, the community has also witnessed a growing body of work on sample-efficient robust RL in RMDPs with different data collection oracles, including the generative model setup (Yang et al., 2022; Panaganti and Kalathil, 2022; Si et al., 2023; Wang et al., 2023b; Yang et al., 2023b; Xu et al., 2023; Clavier et al., 2023; Wang et al., 2023d; Shi et al., 2023), offline setting (Zhou et al., 2021b; Panaganti et al., 2022; Shi and Chi, 2022; Ma et al., 2022; Blanchet et al., 2023; Liu and Xu, 2024b; Wang et al., 2024), and interactive data collection setting (Badrinath and Kalathil, 2021; Wang and Zou, 2021; Liu and Xu, 2024a).

Our work falls into the paradigm of sample-efficient robust RL with interactive data collection. Wang and Zou (2021) and Badrinath and Kalathil (2021) propose efficient online learning algorithms to obtain the optimal robust policy of an infinite horizon RMDP, but none of them handle the challenge of exploration in online RL by assuming the access to *explorative policies*. This assumption enables the learner to collect high-quality data essential for effective learning and decision-making. In contrast, our work focuses on developing efficient algorithms for the fully online setting, where there is no predefined exploration policy to use. Under this more challenging setting, we address the exploration challenge through algorithmic design rather than relying on assumed access to explorative policies.

During the preparation of this work, we are aware of several concurrent and independent works (Liu and Xu, 2024a,b; Wang et al., 2024), which study a different type of RMDPs known as $d$-rectangular linear MDPs (Ma et al., 2022; Blanchet et al., 2023). In particular, Liu and Xu (2024b) and Wang et al. (2024) consider the offline setting, while Liu and Xu (2024a) investigate robust RL through interactive data collection (off-dynamics learning), thus bearing closer relevance to our work. More specifically, under the existence of a "fail-state", the algorithm in Liu and Xu (2024a) can learn an $\varepsilon$-optimal robust policy with provable sample efficiency. In contrast, our work first explicitly uncovers the fundamental hardness of doing robust RL in RMDPs with a TV distance based robust set and without additional assumptions. To overcome the inherent difficulty, we adopt a vanishing minimal value assumption that strictly generalizes the "fail-state" assumption used in Liu and Xu (2024a). Moreover, our focus is on tabular $\mathcal{S} \times \mathcal{A}$-rectangular RMDPs, with customized algorithmic design and theoretical analysis which allow us to obtain a sharp sample complexity bound.

Finally, in Table 1, we compare the sample complexity of our algorithm with prior work on robust RL for RMDPs with $\mathcal{S} \times \mathcal{A}$-rectangular TV robust sets under various settings (generative model/offline dataset). We remark that the works of Panaganti and Kalathil (2022) and Blanchet et al. (2023) are in the paradigm of function approximation, and here we reduce their general sample complexity result to the tabular setup we consider.

**Sample-efficient online non-robust reinforcement learning.** Our work is also closely related to online non-robust RL, which is often formulated as a Markov decision process (MDP) with online data collection. For non-robust online RL, the key challenge is the exploration-exploitation tradeoff. There has been a long line of work (Azar et al., 2017; Dann et al., 2017; Jin et al., 2018; Zanette and Brunskill, 2019; Zhang et al., 2020, 2021; Ménard et al., 2021; Wu et al., 2022; Li et al., 2023; Zhang et al., 2023) addressing this challenge in the context of tabular MDPs, where the state space and action space are finite and also relatively small. In particular, many algorithms (e.g., UCBVI in Azar et al. (2017)) have been proven capable of finding an $\varepsilon$-optimal policy within $\widetilde{\mathcal{O}}(H^3SA/\varepsilon^2)$ sample complexity. Notably, a standard MDP corresponds to an RMDP with a TV robust set and $\rho = 0$, suggesting that OPROVI-TV can naturally achieve nearly minimax-optimality for non-robust RL. Moving beyond the tabular setups, recent works also investigate online non-robust RL with linear function approximation (Jin et al., 2020; Ayoub et al., 2020; Zhou et al., 2021a; Zhong and Zhang, 2023; Huang et al., 2023b; He et al., 2023; Agarwal et al., 2023) and even general function approximations (Jiang et al., 2017; Sun et al., 2019; Du et al., 2021; Jin et al., 2021; Foster et al.,

2021; Liu et al., 2022; Zhong et al., 2022; Liu et al., 2023; Huang et al., 2023a; Xu and Zeevi, 2023; Agarwal et al., 2023).

**Corruption robust reinforcement learning.** Generally speaking, our research is also related to another form of robust RL, namely corruption robust RL (Lykouris et al., 2021; Wei et al., 2022; Zhang et al., 2022; Ye et al., 2023a,b; Yang et al., 2023a; Ye et al., 2024). This branch of researches on robust RL addresses scenarios where training data is corrupted, presenting a distinct challenge from distributionally robust RL. The latter concerns testing time robustness, where the agent is evaluated in a perturbed environment after being trained on nominal data. These two forms of robust RL, while sharing the overarching goal to enhance agent resilience, operate within different contexts and confront distinct challenges. Thus, a direct comparison between these two types of robust RL is difficult because each addresses unique aspects of resilience.

# B Further Discussions

This section complements the main part of the paper by further commenting and discussing several aspects of the paper. Due to space limits, these important remarks are provided here.

## B.1 Discussions on Introduction (Section 1)

**About a generative model and a simulator.** A generative model here means a mechanism that when queried at some state, action, and time step, returns a sample of next state. Here we distinguish this notion with the notion of simulator or simulated environment which generally refers to a human-made training environment that mimics the real-world environment. With a generative model on hand, interactive data collection is no longer needed, but in a simulated environment, it is common to train a robust policy through interactive data collection in practice.

**The definition of total-variation robust set.** We notice that all of the previous work on sample-efficient robust RL in RMDPs with TV robust sets (Yang et al., 2022; Panaganti and Kalathil, 2022; Panaganti et al., 2022; Xu et al., 2023; Blanchet et al., 2023; Shi et al., 2023) relies on defining the TV distance through the general $f$-divergence so that a strong duality representation holds. But this implicitly requires the testing environment transition probability is absolute continuous w.r.t. the training environment transition probability. In this paper, we do not make such a restriction. We prove the same strong duality even if the absolute continuity does not hold. In fact, all the previous work can be directly extended to such TV distance definition via our more general strong duality result.

**An existing work.** We note that an existing work (Dong et al., 2022) also studies the problem of robust RL with interactive data collection. They study $\mathcal{S} \times \mathcal{A}$-rectangular RMDPs with a TV robust set, assuming that the support of the training environment transition is the full state space $\mathcal{S}$. They claim the existence of an algorithm that enjoys a $\widetilde{\mathcal{O}}(\sqrt{K})$-online regret. We point out that their proof exhibits an essential flaw (misuse of Lemma 12 therein) and therefore the regret they claim is invalid.

**The range of the robust set size $\rho$.** We do not signify the situation when $\rho = 1$ since in that case the TV robust set contains all possible transition probabilities, making the problem statistically trivial. In that case, no sample is needed.

## B.2 Discussions on Preliminaries (Section 2)

### B.2.1 Robust Markov Decision Processes

**Robust Bellman equations.** We remark that the original version of the robust Bellman equation (Iyengar, 2005) is for infinite horizon RMDPs and a customized proof of robust Bellman equation for finite horizon RMDPs (Proposition 2.2) can be found in Appendix A.1 of Blanchet et al. (2023). The robust Bellman optimal equation (Proposition 2.3) is then a corollary or can be proved similarly.

**Strong duality under TV distance robust set.** An essential property of the TV robust set is that the robust expectation involved in the robust Bellman equations (Propositions 2.2 and 2.3) has a duality representation that only uses the expectation under the nominal transition kernel. Previous

works, e.g., Yang et al. (2022), have proved such a result when the TV distance is defined through $f$-divergence. Here we extend such a result to the TV distance defined directly though (2.2) that allows a difference support between $p$ and $q$.

**Remark B.1.** *Despite all previous works on RMDPs with TV robust sets relying on the definition of TV distance $D_{\mathrm{TV}}(p(\cdot)\|q(\cdot))$ with absolute continuity of $p$ with respect to $q$ to obtain the strong duality representation in the form of (2.3), their results can be directly extended to TV distance that allows for different support between $p$ and $q$ thanks to Proposition 2.5.*

**Value gap between maximum.** We note that in the proof of Proposition 2.6, we actually show a tighter form of bound of the gap between the maximum and minimum as

$$\frac{1}{\rho} \cdot \left(1 - (1 - \rho)^H\right).$$

But in the sequel, we mainly use the form of $\min\{H, \rho^{-1}\}$ for its brevity and the fact of $(1 - (1 - \rho)^H)/\rho = \Theta\big(\min\{H, \rho^{-1}\}\big)$ in the sense that

$$c \cdot \min\left\{H, \rho^{-1}\right\} \le (1 - (1 - \rho)^H)/\rho \le \min\left\{H, \rho^{-1}\right\}$$

for any $H \ge H_0 \in \mathbb{N}_+$ and $\rho \in [0, 1]$ with some constant $c > 0$ that is independent of $(H, \rho)$.

In contrast with a crude bound of $H$, such a fine upper bound decreases when $\rho$ is large, which is essential to understanding the statistical limits of doing robust RL in RMDPs with TV robust sets.

### B.2.2 Robust RL with Interactive Data Collection

**Sample complexity.** The metric of *sample complexity* is connected with the sample complexity used in robust RL with generative models and offline settings (see related works for the references), wherein the sample complexity means the minimum number of generative samples or pre-collected offline data required to achieve $\varepsilon$-optimality. In contrast, here the sample complexity is measuring the least number of interactions with the training environment needed to learn $\pi^\star$, where no generative or offline sample is available. Such a learning protocol casts unique challenges on the algorithmic design and theoretical analysis to get the optimal sample complexity.

### B.3 Discussions on Hardness Result: The Curse of Support Shift (Section 3)

In contrast to the interactive data collection setting we consider, doing robust RL with a generative model or an offline dataset with good coverage properties does not face the difficulty we displayed through Example 3.1. It turns out that any RMDP with $\mathcal{S} \times \mathcal{A}$-rectangular total-variation robust set (including Example 3.1) can be solved in a sample-efficient manner therein, see Yang et al. (2022); Panaganti and Kalathil (2022); Panaganti et al. (2022); Xu et al. (2023); Blanchet et al. (2023); Shi et al. (2023) and Remark B.1. The intuitive reason is that, for the generative model setting, the learner can directly query any state-action pair to estimate the nominal transition kernel $P^\star$, and thus no support shift happens. The same reason holds for the offline setup with a good-coverage dataset.

There is a broader understanding of the curse of support shift that hinders the tractability of robust RL via interactive data collection. The concept of support shift could be comprehended within a broader context beyond the disjointness of certain parts of the support sets of the training and testing environments. Instead, ensuring a "high probability of disjointness" is enough to maintain the integrity of the hardness result. For instance, we can modify the state $s_{\mathrm{good}}$ in Example 3.1 so that it is no longer an absorbing state. Rather, $s_{\mathrm{good}}$ could transit to $s_{\mathrm{bad}}$ with a small probability, such as $2^{-H}$. This modification expands the support of the training environment to encompass the entire state space. Nevertheless, acquiring information about $s_{\mathrm{bad}}$ necessitates exponential samples, thereby preserving the hardness result.

### B.4 Discussions on A Solvable Case, Efficient Algorithm, and Sharp Analysis (Section 4)

#### B.4.1 Vanishing Minimal Value Assumption

**Another understanding of Assumption 4.1.** To understand this from another perspective, it could be shown that under the conclusions of Proposition 4.2, the robust value function of any

policy $\pi$ is equivalent to the robust value function of this policy under another *discounted* RMDP $(\mathcal{S}, \mathcal{A}, H, P^\star, R', \mathbf{\Phi}')$ with $R'_h(s,a) = (\rho')^{h-1} R_h(s,a)$ and $\mathbf{\Phi}'$ given by

$$\mathbf{\Phi}'(P) = \bigotimes_{(s,a) \in \mathcal{S} \times \mathcal{A}} \mathcal{B}_{\rho'}(s, a; P). \tag{B.1}$$

And therefore we are equivalently considering this new type of RMDPs. Please refer to Section B.4.2 for more discussions on the connections between the two types of RMDPs.

**Examples of Assumption 4.1.** In the sequel, we provide a concrete condition that makes Assumption 4.1 hold, which imposes that the state space of the RMDP has a "closed" subset of "fail-states" with zero rewards.

**Condition B.2** (Fail-states). *There exists a subset $\mathcal{S}_f \subset \mathcal{S}$ of fail states such that*

$$R_h(s,a) = 0, \quad P_h^\star(\mathcal{S}_f | s, a) = 1, \quad \forall (s, a, h) \in \mathcal{S}_f \times \mathcal{A} \times [H].$$

This type of "fail-states" condition is first proposed by Panaganti et al. (2022) (with $|\mathcal{S}_f| = 1$) to handle the computational issues for robust offline RL under function approximations (out of the scope of our work). In contrast, here we make the vanishing minimal value assumption in order to tackle the *support shift* or *extrapolation* issue for the interactive data collection setup. The comparison between the vanishing minimal value assumption (Assumption 4.1) and the "fail-states" condition (Condition B.2) is given below.

**Remark B.3** (Comparison between Assumption 4.1 and Condition B.2). *We first observe that Condition B.2 implies that $\min_{s \in \mathcal{S}} V_{h, P^\star, \mathbf{\Phi}}^\pi(s) = 0$ for any policy $\pi$ and step $h \in [H]$, therefore satisfying the minimal value assumption (Assumption 4.1). Conversely, the vanishing minimal value assumption in Assumption 4.1 is strictly more general than the fail-state condition in Condition B.2. To illustrate, one can consider an RMDP characterized by the state space $\mathcal{S} = \{s_1, s_2\}$, action space $\mathcal{A} = \{a_1\}$, time horizon $H = 2$, reward function $R_h(s,a) = \mathbf{1}\{s = s_2\}$, and transition probabilities defined as follows:*

$$P_1^\star(s_1 | s_1, a_1) = 1 - \rho, \quad P_1^\star(s_2 | s_1, a_1) = \rho, \quad P_1^\star(s_1 | s_2, a_1) = 0, \quad P_1^\star(s_2 | s_2, a_1) = 1,$$

*where $\rho$ is the radius of the robust set. It is evident that no fail-state emerges within such an RMDP structure. However, this RMDP satisfies the vanishing minimal value assumption since $V_{1, P^\star, \mathbf{\Phi}}^\star(s_1) = 0$.*

**Remark B.4** (Reduction to non-robust MDP without loss of generality). *It is noteworthy that assuming the vanishing minimal value (Assumption 4.1) or the presence of fail-states (Condition B.2) in the non-robust case ($\rho = 0$) is without loss of generality. This is achievable by expanding the prior state space $\mathcal{S}$ of MDP to include an additional state $s_f$, denoted as the fail-state. More importantly, this augmentation does not alter the optimal value or the optimal value function of the original MDP. Consequently, it becomes sufficient to seek the optimal policy within the augmented MDP, which satisfies the conditions of vanishing minimal value (Assumption 4.1) or the existence of fail-states (Condition B.2). This indicates that our algorithm and theoretical analysis in the sequel can be directly reduced to non-robust MDPs without additional assumptions.*

### B.4.2 Extensions to Robust Set with Bounded Transition Probability Ratio

In this section, we show that our algorithm design (Algorithm 1) can also be applied to $\mathcal{S} \times \mathcal{A}$-rectangular discounted RMDPs with robust sets given by (B.1) (i.e., bounded ratio between training and testing transition probabilities). We establish that our main theoretical result in Section 4.3 can imply a sublinear regret upper bound for this model, which means that this type of RMDPs can also be solved sample-efficiently by a clever usage of Algorithm 1. This coincides with our intuition on support shift in Section 4.1.

$\mathcal{S} \times \mathcal{A}$**-rectangular discounted RMDPs with robust set** (B.1). We first formally define the model we consider. We define a finite-horizon discounted RMDP as a finite-horizon RMDP $\mathcal{M}_\gamma = (\mathcal{S}, \mathcal{A}, H, P^\star, R_\gamma, \mathbf{\Phi}')$, where the robust set $\mathbf{\Phi}'$ is given by (B.1), i.e.,

$$\mathbf{\Phi}'(P) = \bigotimes_{(s,a) \in \mathcal{S} \times \mathcal{A}} \left\{ \widetilde{P}(\cdot) \in \Delta(\mathcal{S}) : \sup_{s' \in \mathcal{S}} \frac{\widetilde{P}(s')}{P_h^\star(s'|s,a)} \leq \frac{1}{\rho'} \right\} := \bigotimes_{(s,a) \in \mathcal{S} \times \mathcal{A}} \mathcal{B}_{\rho'}(s, a; P^\star). \tag{B.2}$$

This robust set contains transition probabilities that share the same support as the nominal transition kernel. The reward function $R_\gamma = \{\gamma^{h-1} \cdot R_h\}_{h=1}^H$, where $\gamma \in (0,1)$ is the discount factor and $R_h \in [0,1]$ is the true reward at step $h$. That is, the robust value function is now the worst case expected discounted total reward.

**Algorithm and regret bound.** Now we show that we can apply Algorithm 1 to solve robust RL in $\mathcal{S} \times \mathcal{A}$-rectangular discounted RMDPs with robust set (B.2) via interactive data collection.

As motivated by the discussions under Proposition 4.2, we define an auxiliary finite-horizon TV-RMDP $\widetilde{\mathcal{M}}$ as $\widetilde{\mathcal{M}} = (\widetilde{\mathcal{S}}, \mathcal{A}, H, \widetilde{P^\star}, \widetilde{R}, \widetilde{\mathbf{\Phi}})$ which include an additional "fail-state" $s_f$. More specifically, the state space $\widetilde{\mathcal{S}} = \mathcal{S} \cup \{s_f\}$. The transition kernel $\widetilde{P^\star}$ is defined as, for any step $h \in [H]$,

$$\widetilde{P}_h^\star(\cdot|s,a) = P_h^\star(\cdot|s,a), \quad \forall(s,a) \in \mathcal{S} \times \mathcal{A} \quad \text{and} \quad \widetilde{P}_h^\star(\cdot|s_f,a) = \delta_{s_f}(\cdot), \quad \forall a \in \mathcal{A}. \quad \text{(B.3)}$$

The reward function $\widetilde{R}$ is defined as, for any step $h \in [H]$,

$$\widetilde{R}_h(s,a) = \left(\frac{\gamma}{\rho'}\right)^{h-1} \cdot R_h(s,a), \quad \forall(s,a) \in \mathcal{S} \times \mathcal{A} \quad \text{and} \quad \widetilde{R}_h(s_f,a) = 0, \quad \forall a \in \mathcal{A}.$$

We suppose that the discount factor $\gamma \leq \rho'$ so that the reward function $\widetilde{R}_h \in [0,1]$. The robust mapping $\widetilde{\mathbf{\Phi}}$ is defined as, for any $\widetilde{P} : \widetilde{\mathcal{S}} \times \mathcal{A} \mapsto \Delta(\widetilde{\mathcal{S}})$,

$$\widetilde{\mathbf{\Phi}}(\widetilde{P}) = \bigotimes_{(s,a)\in\widetilde{\mathcal{S}}\times\mathcal{A}} \left\{ \widetilde{P}(\cdot) \in \Delta(\widetilde{\mathcal{S}}) : D_{\mathrm{TV}}\big(P(\cdot)\|\widetilde{P}(\cdot|s,a)\big) \leq \rho \right\}$$

$$:= \bigotimes_{(s,a)\in\widetilde{\mathcal{S}}\times\mathcal{A}} \widetilde{\mathcal{P}}_\rho(s,a;\widetilde{P}), \quad \rho = 2 - 2\rho'.$$

Therefore, $\widetilde{\mathcal{M}}$ is an RMDP with $\mathcal{S} \times \mathcal{A}$-rectangular TV robust set of radius $\rho$ and satisfying Assumption 4.1 (because it satisfies the "fail-state" Condition B.2). Furthermore, for any initial state $s_1 \in \widetilde{\mathcal{S}} \setminus \{s_f\} = \mathcal{S}$, the interaction with the transition kernel $\widetilde{P^\star}$ is equivalent to the interaction with the transition kernel $P^\star$ of the original RMDP $\mathcal{M}_\gamma$, since by the definition (B.3), starting from any $s \neq s_f$ the agent would follow the same dynamics as $P^\star$. What's more, for any policy $\widetilde{\pi}_h : \widetilde{\mathcal{S}} \mapsto \Delta(\mathcal{A})$ for $\widetilde{\mathcal{M}}$, it naturally induces the unique policy $\widetilde{\pi}_{\mathcal{S},h} : \mathcal{S} \mapsto \Delta(\mathcal{A})$ for the original RMDP $\mathcal{M}_\gamma$.

Therefore, we can run Algorithm 1 on the auxiliary RMDP $\widetilde{\mathcal{M}}$, starting from the initial state $s_1 \in \widetilde{\mathcal{S}} \setminus \{s_f\}$, which only needs the interaction with $P^\star$. Suppose the output policy by the algorithm is $\{\widetilde{\pi}^k\}_{k=1}^K$, then the following corollary shows the induced policy $\{\widetilde{\pi}_{\mathcal{S}}^k\}_{k=1}^K$ for the original RMDP $\mathcal{M}_\gamma$ enjoys a sublinear regret.

**Corollary B.5** (Online regret of Algorithm 1 for discounted RMDPs with robust sets (B.2)). *Consider an $\mathcal{S} \times \mathcal{A}$-rectangular $\gamma$-discounted RMDP with robust set (B.2) satisfying $0 \leq \gamma \leq \rho' \in (1/2, 1]$. There exists an algorithm $\mathcal{ALG}$ (specified by the above discussion) such that its online regret for this RMDP is bounded by*

$$\mathrm{Regret}_{\mathbf{\Phi}'}^{\mathcal{ALG}}(K) \leq \mathcal{O}\left(\sqrt{\min\{H, (2-2\rho')^{-1}\} H^2 SAK\iota'}\right),$$

*where $\iota' = \log^2(SAHK/\delta)$ and $\mathcal{O}(\cdot)$ hides absolute constants and lower order terms in $K$.*

*Proof of Corollary B.5.* See Appendix F.1 for a detailed proof of Corollary B.5. $\qquad\square$

Corollary B.5 shows that besides $\mathcal{S} \times \mathcal{A}$-rectangular RMDPs with TV robust set and vanishing minimal value assumption, the $\mathcal{S} \times \mathcal{A}$-rectangular discounted RMDP with robust set of bounded transition probability ratio (B.2) can also be solved sample-efficiently by robust RL via interactive data collection. This also echoes our intuition on the support shift issue in Section 4.1. Furthermore, the regret decays as $\rho'$ decays in which case the transition probability ratio bound becomes higher, i.e., the robust set becomes larger.

**Remark B.6.** *The upper bound in Corollary B.5 does not depend on the discount factor $\gamma$ since Algorithm 1 adopts a coarse bound of $\widetilde{R}_h \leq 1$. The upper bound can be directly improved to be $\gamma$-dependent using a tighter truncation in step (4.2) of Algorithm 1.*

## B.5 Discussions of Limitations and Future Works

In this work, we show that in the absence of any structural assumptions, robust RL through interactive data collection necessarily induces a linear regret lower bound in the worst case due to the curse of support shift. Meanwhile, under the vanishing minimal value assumption, an assumption that is able to effectively rule out the potential support shift issues for RMDPs with a TV robust set, we propose a sample-efficient robust RL algorithm for those RMDPs. We discuss some potential extensions here and the associated challenges next.

**Extension to function approximation setting.** The vanishing minimal value assumption also suffices for developing sample-efficient algorithms for $\mathcal{S} \times \mathcal{A}$-rectangular TV-robust-set RMDPs with linear or even general function approximation (Blanchet et al., 2023). Nonetheless, achieving the nearly optimal rate under general function approximation remains elusive.

**Extension to other types of robust set.** Beyond the TV distance based robust set we consider, recent literature on robust RL also investigate other types of $\phi$-divergence based robust set including KL divergence, $\chi^2$ distance (Yang et al., 2022; Shi and Chi, 2022; Blanchet et al., 2023; Xu et al., 2023; Shi et al., 2023). An interesting direction of future work is to investigate is it also possible and, if possible, can we design provably sample-efficient robust RL algorithms with interactive data collection for RMDPs with those types of robust sets. Notably, the KL divergence based robust set naturally does not suffer from the curse of support shifts that gives rise to the hardness for the TV robust set case. However, we find that there are other difficulties for robust RL in KL divergence based RMDPs through interactive data collection. Meanwhile, the optimal sample complexity for robust RL in RMDPs with KL divergence robust set is still elusive even in the offline learning setup (Shi and Chi, 2022). We leave the study of RMDPs with KL divergence robust set for future work.

# C Proofs for Properties of RMDPs with TV Robust Sets

## C.1 Proof of Proposition 2.5

To simplify the notations, we present the following lemma, which directly implies Proposition 2.5.

**Lemma C.1** (Strong duality for TV robust set). *The following duality for total variation robust set holds, for $f : \mathcal{S} \mapsto [0, H]$,*

$$\inf_{Q(\cdot): D_{\mathrm{TV}}(Q(\cdot)\|Q^\star(\cdot)) \leq \sigma} \mathbb{E}_{Q(\cdot)}[f] = \sup_{\eta \in [0,H]} \left\{ -\mathbb{E}_{Q^\star(\cdot)}\big[(\eta - f)_+\big] - \frac{\sigma}{2} \cdot \left(\eta - \min_{s \in \mathcal{S}} f(s)\right)_+ + \eta \right\},$$

*where $\sigma \in [0, 1]$ and the TV distance $D_{\mathrm{TV}}(Q(\cdot)\|Q^\star(\cdot))$ is defined as*

$$D_{\mathrm{TV}}(Q(\cdot)\|Q^\star(\cdot)) = \frac{1}{2} \sum_{s \in \mathcal{S}} |Q(s) - Q^\star(s)|.$$

*Proof of Lemma C.1.* First, we note that when $Q^\star(s) > 0$ for any $s \in \mathcal{S}$, i.e., any $Q(\cdot) \in \Delta(\mathcal{S})$ is absolute continuous w.r.t. $Q^\star(\cdot)$, it has been proved by Yang et al. (2022) that

$$\inf_{Q(\cdot): D_{\mathrm{TV}}(Q(\cdot)\|Q^\star(\cdot)) \leq \sigma} \mathbb{E}_{Q(\cdot)}[f] = \sup_{\eta \in \mathbb{R}} \left\{ -\mathbb{E}_{Q^\star(\cdot)}\big[(\eta - f)_+\big] - \frac{\sigma}{2} \cdot \left(\eta - \min_{s \in \mathcal{S}} f(s)\right)_+ + \eta \right\}.$$

Furthermore, as is shown in Lemma H.8 in Blanchet et al. (2023), the optimal dual variable $\eta^\star$ lies in $[0, H]$ when $f \in [0, H]$. Therefore, for $Q^\star(\cdot)$ such that $Q^\star(s) > 0$ for any $s \in \mathcal{S}$, we have

$$\inf_{Q(\cdot): D_{\mathrm{TV}}(Q(\cdot)\|Q^\star(\cdot)) \leq \sigma} \mathbb{E}_{Q(\cdot)}[f] = \sup_{\eta \in [0,H]} \left\{ -\mathbb{E}_{Q^\star(\cdot)}\big[(\eta - f)_+\big] - \frac{\sigma}{2} \cdot \left(\eta - \min_{s \in \mathcal{S}} f(s)\right)_+ + \eta \right\}.$$

Now for any $Q^\star(\cdot) \in \Delta(\mathcal{S})$, we can prove the same result by averaging $Q^\star(\cdot)$ with a uniform distribution and taking the limit. More specifically, denote $U(\cdot) \in \Delta(\mathcal{S})$ as the uniform distribution

on $\mathcal{S}$, i.e., $U(s) = 1/|\mathcal{S}|$ for any $s \in \mathcal{S}$. Consider the following distributionally robust optimization problem, for any $\epsilon \in [0, 1]$,

$$\mathsf{P}(\epsilon) := \inf_{Q(\cdot):D_{\mathrm{TV}}\left(Q(\cdot)\|(1-\epsilon)Q^\star(\cdot)+\epsilon\cdot U(\cdot)\right)\leq\sigma} \mathbb{E}_{Q(\cdot)}[f].$$

By our previous discussions, since $(1 - \epsilon)Q^\star(s) + \epsilon \cdot U(s) > 0$ for any $s \in \mathcal{S}$ and $\epsilon > 0$, we have that

$$\mathsf{P}(\epsilon) = \mathsf{D}(\epsilon), \quad \forall \epsilon \in (0, 1], \tag{C.1}$$

where the function $\mathsf{D}(\cdot) : [0, 1] \mapsto \mathbb{R}_+$ is defined as

$$\mathsf{D}(\epsilon) := \sup_{\eta\in[0,H]} \left\{ -(1-\epsilon) \cdot \mathbb{E}_{Q^\star(\cdot)}\left[(\eta-f)_+\right] - \epsilon \cdot \mathbb{E}_{U(\cdot)}\left[(\eta-f)_+\right] - \frac{\sigma}{2} \cdot \left(\eta - \min_{s\in\mathcal{S}} f(s)\right)_+ + \eta \right\}.$$

By the definition of $\mathsf{P}(\cdot)$ and $\mathsf{D}(\cdot)$, our goal is to prove that $\mathsf{P}(0) = \mathsf{D}(0)$. To this end, it suffices to prove that (i) $\lim_{\epsilon\to0+} \mathsf{D}(\epsilon)$ exists and $\lim_{\epsilon\to0+} \mathsf{D}(\epsilon) = \mathsf{D}(0)$; and (ii) $\lim_{\epsilon\to0+} \mathsf{P}(\epsilon) = \mathsf{P}(0)$. To prove (i), consider that for any $\epsilon > 0$, by the definition of $\mathsf{D}(\cdot)$,

$$|\mathsf{D}(0) - \mathsf{D}(\epsilon)| \leq \sup_{\eta\in[0,H]} \left\{ \epsilon \cdot \mathbb{E}_{Q^\star(\cdot)}\left[(\eta-f)_+\right] + \epsilon \cdot \mathbb{E}_{U(\cdot)}\left[(\eta-f)_+\right] \right\} \leq \epsilon \cdot 2H.$$

Since the right hand side tends to 0 as $\epsilon$ tends to 0, we know that $\lim_{\epsilon\to0+} \mathsf{D}(\epsilon)$ exists, $\lim_{\epsilon\to0+} \mathsf{D}(\epsilon) = \mathsf{D}(0)$. This also indicates that $\lim_{\epsilon\to0+} \mathsf{P}(\epsilon)$ exists due to (C.1). This proves (i). Now we prove (ii). Notice that since the set

$$\left\{Q(\cdot) \in \Delta(\mathcal{S}) : D_{\mathrm{TV}}\left(Q(\cdot)\|(1-\epsilon)Q^\star(\cdot) + \epsilon \cdot U(\cdot)\right) \leq \sigma\right\}$$

is a closed subset of $\mathbb{R}^{|\mathcal{S}|}$, and $\mathbb{E}_{Q(\cdot)}[f]$ is a continuous function of $Q(\cdot) \in \mathbb{R}^{|\mathcal{S}|}$ w.r.t. the $\|\cdot\|_2$-norm, we can denote the optimal solution to the optimization problem involved in $\mathsf{P}(\epsilon)$ as

$$Q^\dagger_\epsilon(\cdot) = \underset{Q(\cdot):D_{\mathrm{TV}}\left(Q(\cdot)\|(1-\epsilon)Q^\star(\cdot)+\epsilon\cdot U(\cdot)\right)\leq\sigma}{\arg\inf} \mathbb{E}_{Q(\cdot)}[f],$$

which also gives that

$$\mathsf{P}(\epsilon) = \mathbb{E}_{Q^\dagger_\epsilon(\cdot)}[f] = \sum_{s\in\mathcal{S}} Q^\dagger_\epsilon(s)f(s).$$

With these preparations, we are able to prove (ii). On the one hand, consider for any $\epsilon \in (0, 1]$,

$$D_{\mathrm{TV}}\left((1-\epsilon) \cdot Q^\dagger_0(\cdot) + \epsilon \cdot U(\cdot)\|(1-\epsilon) \cdot Q^\star(\cdot) + \epsilon \cdot U(\cdot)\right) \leq (1-\epsilon) \cdot \sigma \leq \sigma.$$

Therefore, for any $\epsilon \in (0, 1]$, it holds that

$$\begin{aligned}
\mathsf{P}(\epsilon) &= \inf_{Q(\cdot):D_{\mathrm{TV}}\left(Q(\cdot)\|(1-\epsilon)Q^\star(\cdot)+\epsilon\cdot U(\cdot)\right)\leq\sigma} \mathbb{E}_{Q(\cdot)}[f] \\
&\leq \mathbb{E}_{(1-\epsilon)\cdot Q^\dagger_0(\cdot)+\epsilon\cdot U(\cdot)}[f] = (1-\epsilon) \cdot \mathbb{E}_{Q^\dagger_0}[f] + \epsilon \cdot \mathbb{E}_{U(\cdot)}[f],
\end{aligned}$$

which implies that

$$\lim_{\epsilon\to0+} \mathsf{P}(\epsilon) \leq \mathbb{E}_{Q^\dagger_0}[f] = \mathsf{P}(0). \tag{C.2}$$

On the other hand, for any $\epsilon \in (0, 1]$,

$$\begin{aligned}
\sigma &\geq \frac{1}{2} \sum_{s\in\mathcal{S}} \left|Q^\dagger_\epsilon(s) - (1-\epsilon) \cdot Q^\star(s) - \epsilon \cdot U(s)\right| \\
&\geq (1-\epsilon) \cdot D_{\mathrm{TV}}(Q^\dagger_\epsilon(\cdot)\|Q^\star(\cdot)) - \epsilon \cdot D_{\mathrm{TV}}(Q^\dagger_\epsilon(\cdot)\|U(\cdot)),
\end{aligned}$$

and by using $D_{\mathrm{TV}}(Q^\dagger_\epsilon(\cdot)\|U(\cdot)) \leq 1$, we obtain that

$$D_{\mathrm{TV}}(Q^\dagger_\epsilon(\cdot)\|Q^\star(\cdot)) \leq \frac{\sigma + \epsilon}{1 - \epsilon}. \tag{C.3}$$

Consider a sequence of $\{\epsilon_i\}_{i=1}^\infty$ converging to 0, i.e., $\lim_{i\to 0+}\epsilon_i = 0$. Since $\{Q_{\epsilon_i}^\dagger(\cdot)\}_{i=1}^\infty$ is a sequence contained in a compact subset of $\mathbb{R}^{|\mathcal{S}|}$, it has a converging (w.r.t. $\|\cdot\|_2$) subsequence denoted by $\{Q_{\epsilon_{i_k}}^\dagger(\cdot)\}_{k=1}^\infty$ whose limit is denoted as $Q^\dagger(\cdot) \in \Delta(\mathcal{S})$. By (C.3), we know that

$$D_{\mathrm{TV}}(Q_{\epsilon_{i_k}}^\dagger(\cdot)\|Q^\star(\cdot)) \leq \frac{\sigma + \epsilon_{i_k}}{1 - \epsilon_{i_k}}. \tag{C.4}$$

Taking limit on both sides of (C.4) (limit of LHS exists since the TV distance is a continuous function (w.r.t. $\|\cdot\|_2$) of its first entry and the limit of RHS obviously exists), we obtain that

$$D_{\mathrm{TV}}(Q^\dagger(\cdot)\|Q^\star(\cdot)) \leq \sigma. \tag{C.5}$$

Now we can arrive at the following,

$$\begin{aligned}
\lim_{\epsilon\to 0+} \mathsf{P}(\epsilon) &= \lim_{\epsilon\to 0+} \mathbb{E}_{Q_\epsilon^\dagger(\cdot)}[f] = \lim_{k\to 0+} \mathbb{E}_{Q_{\epsilon_{i_k}}^\dagger(\cdot)}[f] = \mathbb{E}_{Q^\dagger(\cdot)}[f] \\
&\geq \inf_{Q(\cdot):D_{\mathrm{TV}}(Q(\cdot)\|Q^\star(\cdot))\leq\sigma} \mathbb{E}_{Q(\cdot)}[f] = \mathsf{P}(0),
\end{aligned} \tag{C.6}$$

where the first and the last equality follows from the definition of $\mathsf{P}(\cdot)$, the second equality follows from the choice of the sequence $\{\epsilon_{i_k}\}_{k=1}^\infty$ that converges to 0, the third equality is due to the continuity of $\mathbb{E}_{Q(\cdot)}[f]$ of $Q(\cdot)$ (w.r.t. $\|\cdot\|_2$), and the inequality follows from (C.5). Finally, with (C.2) and (C.6), we conclude that

$$\lim_{\epsilon\to 0+} \mathsf{P}(\epsilon) = \mathsf{P}(0),$$

which proves (ii). Consequently, by (i) and (ii)

$$\mathsf{P}(0) = \lim_{\epsilon\to 0+} \mathsf{P}(\epsilon) = \lim_{\epsilon\to 0+} \mathsf{D}(\epsilon) = \mathsf{D}(0).$$

Recalling the definitions of $\mathsf{P}(\cdot)$ and $\mathsf{D}(\cdot)$, we conclude the proof of Lemma C.1. $\qquad\square$

## C.2 Proof of Proposition 2.6

*Proof of Proposition 2.6.* Here we prove a stronger result that for any policy $\pi$ and step $h \in [H]$

$$\max_{(s,a)\in\mathcal{S}\times\mathcal{A}} Q_{h,P,\Phi}^\pi(s,a) - \min_{(s,a)\in\mathcal{S}\times\mathcal{A}} Q_{h,P,\Phi}^\pi(s,a) \leq \frac{1}{\rho}\cdot\left(1 - (1-\rho)^{H-h+1}\right), \tag{C.7}$$

$$\max_{s\in\mathcal{S}} V_{h,P,\Phi}^\pi(s) - \min_{s\in\mathcal{S}} V_{h,P,\Phi}^\pi(s) \leq \frac{1}{\rho}\cdot\left(1 - (1-\rho)^{H-h+1}\right). \tag{C.8}$$

First, we note that for the last step $h = H$, (C.7) and (C.8) naturally hold since $R_H \in [0,1]$. Now suppose that (C.8) hold for some step $h + 1$. By robust Bellman equation (Proposition 2.2), we have that

$$\begin{aligned}
Q_{h,P^\star,\Phi}^\pi(s,a) &= R_h(s,a) + \mathbb{E}_{\mathcal{P}_\rho(s,a;P_h^\star)}\left[V_{h+1,P^\star,\Phi}^\pi\right] \\
&\leq 1 + \mathbb{E}_{\mathcal{P}_\rho(s,a;P_h^\star)}\left[V_{h+1,P^\star,\Phi}^\pi\right], \quad \forall(s,a)\in\mathcal{S}\times\mathcal{A},
\end{aligned} \tag{C.9}$$

where the inequality uses the fact that $R_h \leq 1$. Now we denote the state with the least robust value as

$$s_0 \in \operatorname*{argmin}_{s\in\mathcal{S}} V_{h+1,P^\star,\Phi}^\pi(s). \tag{C.10}$$

Inspired by Shi et al. (2023), we choose a transition kernel $\widetilde{P}_h$ satisfying that

$$\left\|\widetilde{P}_h(\cdot|s,a)\right\|_1 = 1 - \rho, \quad P_h^\star(s'|s,a) \geq \widetilde{P}_h(s'|s,a) \geq 0, \quad \forall(s,a,s')\in\mathcal{S}\times\mathcal{A}\times\mathcal{S},$$

which implies that

$$D_{\mathrm{TV}}\left(\widetilde{P}_h(\cdot|s,a) + \rho\cdot\delta_{s_0}(\cdot)\,\middle\|\,P_h^\star(\cdot|s,a)\right) \leq \rho, \quad \forall(s,a)\in\mathcal{S}\times\mathcal{A}.$$

Here $\delta_{s_0}(\cdot)$ is the point measure centered at $s_0$ defined in (C.10). Combined with (C.9), we have that

$$
\begin{aligned}
Q^\pi_{h,P^\star,\boldsymbol{\Phi}}(s,a) &\leq 1 + \mathbb{E}_{\widetilde{P}_h(\cdot|s,a)+\rho\cdot\delta_{s_0}(\cdot)}\left[V^\pi_{h+1,P^\star,\boldsymbol{\Phi}}\right] \\
&= 1 + \mathbb{E}_{\widetilde{P}_h(\cdot|s,a)}\left[V^\pi_{h+1,P^\star,\boldsymbol{\Phi}}\right] + \rho \cdot V^\pi_{h+1,P^\star,\boldsymbol{\Phi}}(s_0) \\
&\leq 1 + (1-\rho) \cdot \max_{s\in\mathcal{S}} V^\pi_{h+1,P^\star,\boldsymbol{\Phi}}(s) + \rho \cdot \min_{s\in\mathcal{S}} V^\pi_{h+1,P^\star,\boldsymbol{\Phi}}(s). \qquad \text{(C.11)}
\end{aligned}
$$

Consequently from (C.11), we further obtain that for any $(s,a)\in\mathcal{S}\times\mathcal{A}$,

$$
\begin{aligned}
Q^\pi_{h,P^\star,\boldsymbol{\Phi}}(s,a) &- \min_{(s,a)\in\mathcal{S}\times\mathcal{A}} Q^\pi_{h,P^\star,\boldsymbol{\Phi}}(s,a) \\
&\leq 1 + (1-\rho) \cdot \max_{s\in\mathcal{S}} V^\pi_{h+1,P^\star,\boldsymbol{\Phi}}(s) + \rho \cdot \min_{s\in\mathcal{S}} V^\pi_{h+1,P^\star,\boldsymbol{\Phi}}(s) - \min_{(s,a)\in\mathcal{S}\times\mathcal{A}} Q^\pi_{h,P^\star,\boldsymbol{\Phi}}(s,a) \\
&= 1 + (1-\rho) \cdot \left(\max_{s\in\mathcal{S}} V^\pi_{h+1,P^\star,\boldsymbol{\Phi}}(s) - \min_{s\in\mathcal{S}} V^\pi_{h+1,P^\star,\boldsymbol{\Phi}}(s)\right) \\
&\quad + \min_{s\in\mathcal{S}} V^\pi_{h+1,P^\star,\boldsymbol{\Phi}}(s) - \min_{(s,a)\in\mathcal{S}\times\mathcal{A}} Q^\pi_{h,P^\star,\boldsymbol{\Phi}}(s,a) \\
&\leq 1 + (1-\rho) \cdot \left(\max_{s\in\mathcal{S}} V^\pi_{h+1,P^\star,\boldsymbol{\Phi}}(s) - \min_{s\in\mathcal{S}} V^\pi_{h+1,P^\star,\boldsymbol{\Phi}}(s)\right), \qquad \text{(C.12)}
\end{aligned}
$$

where the first inequality uses (C.11) and the last inequality uses the following fact,

$$
\begin{aligned}
\min_{(s,a)\in\mathcal{S}\times\mathcal{A}} Q^\pi_{h,P^\star,\boldsymbol{\Phi}}(s,a) &= \min_{(s,a)\in\mathcal{S}\times\mathcal{A}} \left\{R_h(s,a) + \mathbb{E}_{\mathcal{P}_\rho(s,a;P^\star_h)}\left[V^\pi_{h+1,P^\star,\boldsymbol{\Phi}}\right]\right\} \\
&\geq \min_{s\in\mathcal{S}} V^\pi_{h+1,P^\star,\boldsymbol{\Phi}}(s).
\end{aligned}
$$

Now applying the assumption that (C.8) holds at step $h+1$ to the right hand side of (C.12), we obtain that

$$
\begin{aligned}
\max_{(s,a)\in\mathcal{S}\times\mathcal{A}} Q^\pi_{h,P^\star,\boldsymbol{\Phi}}(s,a) - \min_{(s,a)\in\mathcal{S}\times\mathcal{A}} Q^\pi_{h,P^\star,\boldsymbol{\Phi}}(s,a) &\leq 1 + \frac{1-\rho}{\rho} \cdot \left(1 - (1-\rho)^{H-h}\right) \\
&= \frac{1}{\rho} \cdot \left(1 - (1-\rho)^{H-h+1}\right).
\end{aligned}
$$

Thus given (C.8) at step $h+1$, we can derive (C.7) at step $h$. Now by noticing that

$$
\min_{(s,a)\in\mathcal{S}\times\mathcal{A}} Q^\pi_{h,P^\star,\boldsymbol{\Phi}}(s,a) \leq \min_{s\in\mathcal{S}} V^\pi_{h,P^\star,\boldsymbol{\Phi}}(s) \leq \max_{s\in\mathcal{S}} V^\pi_{h,P^\star,\boldsymbol{\Phi}}(s) \leq \max_{(s,a)\in\mathcal{S}\times\mathcal{A}} Q^\pi_{h,P^\star,\boldsymbol{\Phi}}(s,a),
$$

we can conclude that (C.8) also holds at step $h$. As a result, by an induction argument, we finish the proof of Proposition 2.6. $\qquad \square$

## C.3  Proof of Proposition 4.2

*Proof of Proposition 4.2.* We consider some fixed $(s,a,h)\in\mathcal{S}\times\mathcal{A}\times[H]$ throughout proof. By Lemma C.1, we have that

$$
\begin{aligned}
\mathbb{E}_{\mathcal{P}_\rho(s,a;P^\star_h)}[V] &= \sup_{\eta\in\mathbb{R}} \left\{-\mathbb{E}_{P^\star_h(\cdot|s,a)}\left[(\eta-V)_+\right] - \frac{\rho}{2} \cdot \left(\eta - \min_{s\in\mathcal{S}} V(s)\right)_+ + \eta\right\} \\
&= \sup_{\eta\in[0,H]} \left\{-\mathbb{E}_{P^\star_h(\cdot|s,a)}\left[(\eta-V)_+\right] - \frac{\rho}{2} \cdot \left(\eta - \min_{s\in\mathcal{S}} V(s)\right)_+ + \eta\right\} \\
&= \sup_{\eta\in[0,H]} \left\{-\mathbb{E}_{P^\star_h(\cdot|s,a)}\left[(\eta-V)_+\right] + \left(1 - \frac{\rho}{2}\right) \cdot \eta\right\}, \qquad \text{(C.13)}
\end{aligned}
$$

where the second equality follows from the fact the optimal dual variable $\eta^\star$ is in $[0,H]$ when $V\in[0,H]$ (see e.g., Lemma H.8 in Blanchet et al. (2023)), and the last equality is obtained by the fact that $\min_{s\in\mathcal{S}} V(s) = 0$.

**Part (i).** For any $\eta \in [0, H]$ and $Q \in \mathcal{B}_{\rho'}(s, a; P_h^\star)$, we have that

$$-\mathbb{E}_{P_h^\star(\cdot|s,a)}\big[(\eta - V)_+\big] + \Big(1 - \frac{\rho}{2}\Big) \cdot \eta \leq \Big(1 - \frac{\rho}{2}\Big) \cdot \Big(-\mathbb{E}_{Q(\cdot)}\big[(\eta - V)_+\big] + \eta\Big)$$

$$\leq \Big(1 - \frac{\rho}{2}\Big) \cdot \Big(-\mathbb{E}_{Q(\cdot)}\big[\eta - V\big] + \eta\Big)$$

$$= \Big(1 - \frac{\rho}{2}\Big) \cdot \mathbb{E}_{Q(\cdot)}\big[V\big], \tag{C.14}$$

where the first inequality uses the definition of $\mathcal{B}_{\rho'}(s, a; P_h^\star)$, the second equality follows from the fact that $(x)_+ \geq x$. Furthermore, by (C.14) we have that

$$\sup_{\eta \in [0, H]} \Big\{ -\mathbb{E}_{P_h^\star(\cdot|s,a)}\big[(\eta - V)_+\big] + \Big(1 - \frac{\rho}{2}\Big) \cdot \eta \Big\} \leq \Big(1 - \frac{\rho}{2}\Big) \cdot \inf_{Q \in \mathcal{B}_{\rho'}(s,a;P_h^\star)} \mathbb{E}_{Q(\cdot)}\big[V\big]. \tag{C.15}$$

Combining (C.13) and (C.15), we have that

$$\mathbb{E}_{\mathcal{P}_\rho(s,a;P_h^\star)}\big[V\big] \leq \rho' \cdot \mathbb{E}_{\mathcal{B}_{\rho'}(s,a;P_h^\star)}\big[V\big].$$

**Part (ii).** Since $\rho \in [0, 1]$, we know that there exists a $\widetilde{\eta} \in [0, H]$ such that

$$\sum_{s':V(s')<\widetilde{\eta}} P_h^\star(s'|s,a) \leq 1 - \frac{\rho}{2} \leq \sum_{s':V(s')\leq\widetilde{\eta}} P_h^\star(s'|s,a),$$

which further implies that we have the following interpolation for some $\lambda \in [0, 1]$:

$$1 - \frac{\rho}{2} = \lambda \sum_{s':V(s')<\widetilde{\eta}} P_h^\star(s'|s,a) + (1 - \lambda) \sum_{s':V(s')\leq\widetilde{\eta}} P_h^\star(s'|s,a).$$

We define a probability measure $\widetilde{P}^\star \in \Delta(\mathcal{S})$ as

$$\widetilde{P}_h^\star = \frac{\lambda P_h^\star(s'|s,a) \cdot \mathbf{1}\{V(s') > \widetilde{\eta}\} + (1 - \lambda)P_h^\star(s'|s,a) \cdot \mathbf{1}\{V(s') \geq \widetilde{\eta}\}}{1 - \frac{\rho}{2}}. \tag{C.16}$$

It is not difficult to verify that $\widetilde{P}_h^\star \in \mathcal{B}_{\rho'}(s, a; P_h^\star)$. Hence, we have

$$\Big(1 - \frac{\rho}{2}\Big) \cdot \mathbb{E}_{\mathcal{B}_{\rho'}(s,a;P_h^\star)}[V] \leq \Big(1 - \frac{\rho}{2}\Big) \cdot \mathbb{E}_{\widetilde{P}_h^\star(\cdot)}[V]$$

$$= \Big(1 - \frac{\rho}{2}\Big) \cdot \mathbb{E}_{\widetilde{P}_h^\star(\cdot)}[V - \widetilde{\eta}] + \Big(1 - \frac{\rho}{2}\Big) \cdot \widetilde{\eta}$$

$$= -\mathbb{E}_{P_h^\star(\cdot|s,a)}\big[(\widetilde{\eta} - V)_+\big] + \Big(1 - \frac{\rho}{2}\Big) \cdot \widetilde{\eta}, \tag{C.17}$$

where the last inequality uses the definition of $\widetilde{P}_h^\star$ in (C.16). Furthermore, by (C.17) we have that

$$\rho' \cdot \mathbb{E}_{\mathcal{B}_{\rho'}(s,a;P_h^\star)}\big[V\big] \leq \sup_{\eta \in [0, H]} \Big\{ -\mathbb{E}_{P_h^\star(\cdot|s,a)}\big[(\eta - V)_+\big] + \Big(1 - \frac{\rho}{2}\Big) \cdot \eta \Big\} \tag{C.18}$$

$$= \mathbb{E}_{\mathcal{P}_\rho(s,a;P_h^\star)}\big[V\big],$$

where the equality follows from (C.13).

**Combining Part (i) and Part (ii).** Finally, combining (C.15) and (C.18), we prove Proposition 4.2.
$\square$

# D  Proofs for Hardness Results

## D.1  Proof of Theorem 3.2

*Proof of Theorem 3.2.* We first explicitly give the expressions of the robust value functions in Example 3.1, based on which we derive the desired online regret lower bound.

**Robust value function.** Firstly, we can explicitly write down the expression of the robust value functions for any policy $\pi$ under Example 3.1, i.e., $V^\pi_{h,P^\star,\mathcal{M}_\theta,\mathbf{\Phi}}$ and $Q^\pi_{h,P^\star,\mathcal{M}_\theta,\mathbf{\Phi}}$. From now on we fix a policy $\pi$.

*For step $h = 3$*, the robust value function is the reward received. We can directly obtain for any $a \in \mathcal{A}$,

$$Q^\pi_{3,P^\star,\mathcal{M}_\theta,\mathbf{\Phi}}(s_{\text{good}}, a) = V^\pi_{3,P^\star,\mathcal{M}_\theta,\mathbf{\Phi}}(s_{\text{good}}) = 1, \tag{D.1}$$

$$Q^\pi_{3,P^\star,\mathcal{M}_\theta,\mathbf{\Phi}}(s_{\text{bad}}, a) = V^\pi_{3,P^\star,\mathcal{M}_\theta,\mathbf{\Phi}}(s_{\text{bad}}) = 0.$$

*For step $h = 2$*, by the robust Bellman equation (Proposition 2.2), we have that for the good state $s_{\text{good}}$,

$$Q^\pi_{2,P^\star,\mathcal{M}_\theta,\mathbf{\Phi}}(s_{\text{good}}, a) \tag{D.2}$$

$$= 1 + \inf_{P \in \mathcal{P}_\rho(s_{\text{good}},a;P_2^{\star,\mathcal{M}_\theta})} \mathbb{E}_{P(\cdot)}\left[V^\pi_{3,P^\star,\mathcal{M}_\theta,\mathbf{\Phi}}\right] = 1 + (1 - \rho), \quad \forall a \in \mathcal{A},$$

where the last equality is because $V^\pi_{3,P^\star,\mathcal{M}_\theta,\mathbf{\Phi}}$ takes the minimal value 0 at the bad state $s_{\text{bad}}$ and thus the most adversarial transition distribution is achieved at

$$P^\dagger(s') = (1 - \rho) \cdot \mathbf{1}\{s' = s_{\text{good}}\} + \rho \cdot \mathbf{1}\{s' = s_{\text{bad}}\}.$$

Similarly, we have that for the bad state $s_{\text{bad}}$,

$$Q^\pi_{2,P^\star,\mathcal{M}_\theta,\mathbf{\Phi}}(s_{\text{bad}}, a) = 0 + \inf_{P \in \mathcal{P}_\rho(s_{\text{bad}},a;P_2^{\star,\mathcal{M}_\theta})} \mathbb{E}_{P(\cdot)}\left[V^\pi_{3,P^\star,\mathcal{M}_\theta,\mathbf{\Phi}}\right] \tag{D.3}$$

$$= \begin{cases} p - \rho, & a = \theta \\ q - \rho, & a = 1 - \theta \end{cases}.$$

Finally by the robust Bellman equation again, we have that

$$V^\pi_{2,P^\star,\mathcal{M}_\theta,\mathbf{\Phi}}(s_{\text{good}}) = 1 + (1 - \rho),$$

$$V^\pi_{2,P^\star,\mathcal{M}_\theta,\mathbf{\Phi}}(s_{\text{bad}}) = \pi_2(\theta|s_{\text{bad}}) \cdot (p - \rho) + \pi_2(1 - \theta|s_{\text{bad}}) \cdot (q - \rho).$$

Notice that by $q < p$ we know that $V^\pi_{2,P^\star,\mathcal{M}_\theta,\mathbf{\Phi}}(s_{\text{bad}}) < p - \rho < 1 + (1 - \rho) < V^\pi_{2,P^\star,\mathcal{M}_\theta,\mathbf{\Phi}}(s_{\text{good}})$.

*For step $h = 1$*, we consider the robust values on the initial state $s_1 = s_{\text{good}}$, by robust Bellman equation,

$$Q^\pi_{1,P^\star,\mathcal{M}_\theta,\mathbf{\Phi}}(s_{\text{good}}, a) = 1 + \inf_{P \in \mathcal{P}_\rho(s_{\text{good}},a;P_1^{\star,\mathcal{M}_\theta})} \mathbb{E}_{P(\cdot)}\left[V^\pi_{2,P^\star,\mathcal{M}_\theta,\mathbf{\Phi}}\right] \tag{D.4}$$

$$= 1 + (1 - \rho) \cdot \left[1 + (1 - \rho)\right]$$

$$+ \rho \cdot \left[\pi_2(\theta|s_{\text{bad}}) \cdot (p - \rho) + \pi_2(1 - \theta|s_{\text{bad}}) \cdot (q - \rho)\right], \quad \forall a \in \mathcal{A}.$$

By robust Bellman equation, we also derive $V^\pi_{1,P^\star,\mathcal{M}_\theta,\mathbf{\Phi}}(s_{\text{good}}) = Q^\pi_{1,P^\star,\mathcal{M}_\theta,\mathbf{\Phi}}(s_{\text{good}}, a)$ for $\forall a \in \mathcal{A}$.

**Lower bound the online regret under Example 3.1.** With all the previous preparation, we can lower bound the online regret for robust RL with interactive data collection in Example 3.1. But first, we present the following general lemma.

**Lemma D.1** (Performance difference lemma for robust value function). *For any RMDP satisfying Assumption 2.1 and any policy $\pi$, the following inequality holds,*

$$V^{\pi^\star}_{1,P^\star,\mathbf{\Phi}}(s) - V^\pi_{1,P^\star,\mathbf{\Phi}}(s)$$

$$\geq \mathbb{E}_{(P^{\pi^\star,\dagger},\pi^\star)}\left[\sum_{h=1}^{H}\sum_{a \in \mathcal{A}}\left(\pi^\star_h(a|s_h) - \pi_h(a|s_h)\right) \cdot Q^\pi_{h,P^\star,\mathbf{\Phi}}(s_h, a)\,\middle|\,s_1 = s\right],$$

*where the expectation is taken with respect to the trajectories induced by policy $\pi^\star$, transition kernel $P^{\pi^\star,\dagger}$. Here the transition kernel $P^{\pi^\star,\dagger}$ is defined as*

$$P^{\pi^\star,\dagger}_h(\cdot|s,a) = \operatorname*{arginf}_{P \in \mathcal{P}(s,a;P_h^\star)} \mathbb{E}_{P(\cdot)}\left[V^{\pi^\star}_{h+1,P^\star,\mathbf{\Phi}}\right],$$

*where $\mathcal{P}(s,a;P_h^\star)$ is the robust set for state-action pair $(s,a)$ (see Assumption 2.1).*

*Proof of Lemma D.1.* Please refer to Appendix D.2 for a detailed proof of Lemma D.1. □

Now back to Example 3.1, our previous calculation actually shows that, by (D.1) for step $h = 3$,

$$\sum_{a \in \mathcal{A}} \left( \pi_3^{\star, \mathcal{M}_\theta}(a|s_3) - \pi_3(a|s_3) \right) \cdot Q_{3, P^\star, \mathcal{M}_\theta, \mathbf{\Phi}}^\pi(s_3, a) = 0, \quad \forall s_3 \in \{s_{\text{good}}, s_{\text{bad}}\}. \quad \text{(D.5)}$$

and by (D.4) we also have that for step $h = 1$,

$$\sum_{a \in \mathcal{A}} \left( \pi_1^{\star, \mathcal{M}_\theta}(a|s_1) - \pi_1(a|s_1) \right) \cdot Q_{1, P^\star, \mathcal{M}_\theta, \mathbf{\Phi}}^\pi(s_1, a) = 0, \quad \text{where} \quad s_1 = s_{\text{good}}. \quad \text{(D.6)}$$

Finally, let's consider step $h = 2$. By (D.2), we have that for the good state, it holds that

$$\sum_{a \in \mathcal{A}} \left( \pi_2^{\star, \mathcal{M}_\theta}(a|s_{\text{good}}) - \pi_2(a|s_{\text{good}}) \right) \cdot Q_{2, P^\star, \mathcal{M}_\theta, \mathbf{\Phi}}^\pi(s_{\text{good}}, a) = 0, \quad \text{(D.7)}$$

Meanwhile, by (D.3), we have that for the bad state, it holds that (recall that $q < p$)

$$\sum_{a \in \mathcal{A}} \left( \pi_2^{\star, \mathcal{M}_\theta}(a|s_{\text{bad}}) - \pi_2(a|s_{\text{bad}}) \right) \cdot Q_{2, P^\star, \mathcal{M}_\theta, \mathbf{\Phi}}^\pi(s_{\text{bad}}, a)$$

$$= \max \left\{ p - \rho, q - \rho \right\} - \left( \pi_2(\theta|s_{\text{bad}}) \cdot (p - \rho) + \pi_2(1 - \theta|s_{\text{bad}}) \cdot (q - \rho) \right)$$

$$= p - \rho - \left( \pi_2(\theta|s_{\text{bad}}) \cdot (p - \rho) + \pi_2(1 - \theta|s_{\text{bad}}) \cdot (q - \rho) \right)$$

$$= \frac{p - q}{2} \cdot \left( \left| \pi_2^{\star, \mathcal{M}_\theta}(\theta|s_{\text{bad}}) - \pi_2(\theta|s_{\text{bad}}) \right| + \left| \pi_2^{\star, \mathcal{M}_\theta}(1 - \theta|s_{\text{bad}}) - \pi_2(1 - \theta|s_{\text{bad}}) \right| \right)$$

$$= (p - q) \cdot D_{\text{TV}} \left( \pi_2^{\star, \mathcal{M}_\theta}(\cdot|s_{\text{bad}}) \big\| \pi_2(\cdot|s_{\text{bad}}) \right), \quad \text{(D.8)}$$

where according to (D.3) the optimal policy of $\mathcal{M}_\theta$ at $h = 2$ and $s_{\text{bad}}$ is $\pi_2^{\star, \mathcal{M}_\theta}(\theta|s_{\text{bad}}) = 1$. Now combining (D.5), (D.6), (D.7), and (D.8) with Lemma D.1, we can conclude that

$$V_{1, P^\star, \mathcal{M}_\theta, \mathbf{\Phi}}^{\pi^{\star, \mathcal{M}_\theta}}(s_{\text{good}}) - V_{1, P^\star, \mathcal{M}_\theta, \mathbf{\Phi}}^\pi(s_{\text{good}})$$

$$\geq \mathbb{E}_{a_1 \sim \pi_1^{\star, \mathcal{M}_\theta}(\cdot|s_{\text{good}}), s_2 \sim P_1^{\pi^{\star, \mathcal{M}_\theta}, \dagger}(\cdot|s_{\text{good}}, a_1)} \left[ \sum_{a \in \mathcal{A}} \left( \pi_2^\star(a|s_2) - \pi_2(a|s_2) \right) \cdot Q_{2, P^\star, \mathcal{M}_\theta, \mathbf{\Phi}}^\pi(s_2, a) \right]$$

$$= P_1^{\pi^{\star, \mathcal{M}_\theta}, \dagger}(s_{\text{bad}}|s_{\text{good}}, 0) \cdot (p - q) \cdot D_{\text{TV}} \left( \pi_2^{\star, \mathcal{M}_\theta}(\cdot|s_{\text{bad}}) \big\| \pi_2(\cdot|s_{\text{bad}}) \right), \quad \text{(D.9)}$$

where the adversarial transition kernel $P_1^{\pi^{\star, \mathcal{M}_\theta}, \dagger}$ is given by

$$P_1^{\pi^{\star, \mathcal{M}_\theta}, \dagger}(\cdot|s_{\text{good}}, 0) = \operatorname*{argmin}_{P \in \mathcal{P}(s_{\text{good}}, 0; P_1^{\star, \mathcal{M}_\theta})} \mathbb{E}_{P(\cdot)} \left[ V_{2, P^\star, \mathcal{M}_\theta, \mathbf{\Phi}}^{\pi^{\star, \mathcal{M}_\theta}} \right]$$

$$= (1 - \rho) \cdot \mathbf{1}\{\cdot = s_{\text{good}}\} + \rho \cdot \mathbf{1}\{\cdot = s_{\text{bad}}\}. \quad \text{(D.10)}$$

Consequently, taking (D.10) back into (D.9), we have that

$$V_{1, P^\star, \mathcal{M}_\theta, \mathbf{\Phi}}^{\pi^{\star, \mathcal{M}_\theta}}(s_{\text{good}}) - V_{1, P^\star, \mathcal{M}_\theta, \mathbf{\Phi}}^\pi(s_{\text{good}}) \geq \rho \cdot (p - q) \cdot D_{\text{TV}} \left( \pi_2^{\star, \mathcal{M}_\theta}(\cdot|s_{\text{bad}}) \big\| \pi_2(\cdot|s_{\text{bad}}) \right).$$

This implies that for any algorithm executing $\pi^1, \cdots, \pi^K$, its online regret is lower bounded by the following,

$$\text{Regret}_{\mathbf{\Phi}}^{\mathcal{M}_\theta, \mathcal{ALG}}(K) = \sum_{k=1}^K V_{1, P^\star, \mathcal{M}_\theta, \mathbf{\Phi}}^{\pi^{\star, \mathcal{M}_\theta}}(s_{\text{good}}) - V_{1, P^\star, \mathcal{M}_\theta, \mathbf{\Phi}}^{\pi^k}(s_{\text{good}})$$

$$\geq \rho \cdot (p - q) \cdot \sum_{k=1}^K D_{\text{TV}} \left( \pi_2^{\star, \mathcal{M}_\theta}(\cdot|s_{\text{bad}}) \big\| \pi_2^k(\cdot|s_{\text{bad}}) \right).$$

However, since in RMDPs of Example 3.1, the online interaction process is always kept in $s_{\text{good}}$ and there is no information on $\theta$ which can only be accessed at $(s, h) = (s_{\text{bad}}, 2)$. As a result, the estimates $\pi_2^k(\cdot|s_{\text{bad}})$ of $\pi_2^{\star,\mathcal{M}_\theta}(\cdot|s_{\text{bad}}) = \mathbf{1}\{\cdot = \theta\}$ can do no better than a random guess. Put it formally, consider that

$$
\sup_{\theta \in \{0,1\}} \mathbb{E}_{\mathcal{M}_\theta, \mathcal{ALG}} \left[ \text{Regret}_{\mathbf{\Phi}}^{\mathcal{M}_\theta, \mathcal{ALG}}(K) \right]
$$

$$
\geq \rho \cdot (p - q) \cdot \sup_{\theta \in \{0,1\}} \mathbb{E}_{\mathcal{M}_\theta, \mathcal{ALG}} \left[ \sum_{k=1}^{K} D_{\text{TV}} \left( \pi_2^{\star,\mathcal{M}_\theta}(\cdot|s_{\text{bad}}) \big\| \pi_2^k(\cdot|s_{\text{bad}}) \right) \right]
$$

$$
= \rho \cdot (p - q) \cdot \sup_{\theta \in \{0,1\}} \sum_{k=1}^{K} \mathbb{E}_{\mathcal{ALG}} \left[ \pi_2^k(1 - \theta|s_{\text{bad}}) \right]. \tag{D.11}
$$

Here in the last equality we can drop the subscription of $\mathcal{M}_\theta$ because the algorithm outputs $\pi_2^k$ independent of the $\theta$ due to our previous discussion. Notice that

$$
\sum_{\theta \in \{0,1\}} \sum_{k=1}^{K} \mathbb{E}_{\mathcal{ALG}} \left[ \pi_2^k(1 - \theta|s_{\text{bad}}) \right] = \sum_{k=1}^{K} \sum_{\theta \in \{0,1\}} \mathbb{E}_{\mathcal{ALG}} \left[ \pi_2^k(1 - \theta|s_{\text{bad}}) \right] = \sum_{k=1}^{K} 1 = K,
$$

which further indicates that

$$
\sup_{\theta \in \{0,1\}} \sum_{k=1}^{K} \mathbb{E}_{\mathcal{ALG}} \left[ \pi_2^k(1 - \theta|s_{\text{bad}}) \right] \geq \frac{K}{2}. \tag{D.12}
$$

Therefore, by combining (D.11) and (D.12), we conclude that

$$
\inf_{\mathcal{ALG}} \sup_{\theta \in \{0,1\}} \mathbb{E}_{\mathcal{M}_\theta, \mathcal{ALG}} \left[ \text{Regret}_{\mathbf{\Phi}}^{\mathcal{M}_\theta, \mathcal{ALG}}(K) \right] \geq (p - q) \cdot \frac{\rho K}{2}.
$$

This is the desired online regret lower bound of $\Omega(\rho \cdot K)$ for the RMDPs presented in Example 3.1. Furthermore, we can construct two RMDPs $\{\widetilde{\mathcal{M}}_0, \widetilde{\mathcal{M}}_1\}$ with horizon $3H$ by concatenating $H$ RMDPs $\{\mathcal{M}_0, \mathcal{M}_1\}$ presented in Example 3.1. Notably, at any steps $\{3i + 1\}_{i=0}^{H-1}$, we define

$$
R_{3i+1}(s_{\text{bad}}, a) = 1, \qquad P_{3i+1}^{\star, \widetilde{\mathcal{M}}_\theta}(s_{\text{good}}|s_{\text{bad}}, a) = 1, \quad \forall (a, \theta) \in \mathcal{A} \times \{0,1\}.
$$

Then we have

$$
\inf_{\mathcal{ALG}} \sup_{\theta \in \{0,1\}} \mathbb{E}_{\widetilde{\mathcal{M}}_\theta, \mathcal{ALG}} \left[ \text{Regret}_{\mathbf{\Phi}}^{\widetilde{\mathcal{M}}_\theta, \mathcal{ALG}}(K) \right] \geq H \cdot \Omega(\rho \cdot K) = \Omega(\rho \cdot HK),
$$

which completes the proof of Theorem 3.2. $\qquad\square$

## D.2 Proof of Lemma D.1

*Proof of Lemma D.1.* For any step $h \in [H]$, we have that by robust Bellman equation (Proposition 2.2),

$$
Q_{h,P^\star,\mathbf{\Phi}}^{\pi^\star}(s,a) - Q_{h,P^\star,\mathbf{\Phi}}^{\pi}(s,a) = \mathbb{E}_{\mathcal{P}_\rho(s,a;P_h^\star)} \left[ V_{h+1,P^\star,\mathbf{\Phi}}^{\pi^\star} \right] - \mathbb{E}_{\mathcal{P}_\rho(s,a;P_h^\star)} \left[ V_{h+1,P^\star,\mathbf{\Phi}}^{\pi} \right].
$$

By the definition of the transition kernel $P^{\pi^\star,\dagger}$ in Lemma D.1 and the property of infimum, we have that

$$
Q_{h,P^\star,\mathbf{\Phi}}^{\pi^\star}(s,a) - Q_{h,P^\star,\mathbf{\Phi}}^{\pi}(s,a) \geq \mathbb{E}_{P_h^{\pi^\star,\dagger}(\cdot|s,a)} \left[ V_{h+1,P^\star,\mathbf{\Phi}}^{\pi^\star} \right] - \mathbb{E}_{P_h^{\pi^\star,\dagger}(\cdot|s,a)} \left[ V_{h+1,P^\star,\mathbf{\Phi}}^{\pi} \right]
$$

$$
= \mathbb{E}_{P_h^{\pi^\star,\dagger}(\cdot|s,a)} \left[ V_{h+1,P^\star,\mathbf{\Phi}}^{\pi^\star} - V_{h+1,P^\star,\mathbf{\Phi}}^{\pi} \right]. \tag{D.13}
$$

By robust Bellman equation (Proposition 2.2) and (D.13), we further obtain that

$$
V_{h,P^\star,\mathbf{\Phi}}^{\pi^\star}(s) - V_{h,P^\star,\mathbf{\Phi}}^{\pi}(s) = \mathbb{E}_{\pi_h^\star(\cdot|s)} \left[ Q_{h,P^\star,\mathbf{\Phi}}^{\pi^\star}(s,\cdot) \right] - \mathbb{E}_{\pi_h(\cdot|s)} \left[ Q_{h,P^\star,\mathbf{\Phi}}^{\pi}(s,\cdot) \right]
$$

$$
= \mathbb{E}_{\pi_h^\star(\cdot|s)} \left[ Q_{h,P^\star,\mathbf{\Phi}}^{\pi}(s,\cdot) \right] - \mathbb{E}_{\pi_h(\cdot|s)} \left[ Q_{h,P^\star,\mathbf{\Phi}}^{\pi}(s,\cdot) \right]
$$

$$
+ \mathbb{E}_{\pi_h^\star(\cdot|s)} \left[ Q_{h,P^\star,\mathbf{\Phi}}^{\pi^\star}(s,\cdot) \right] - \mathbb{E}_{\pi_h^\star(\cdot|s)} \left[ Q_{h,P^\star,\mathbf{\Phi}}^{\pi}(s,\cdot) \right]
$$

$$
\geq \sum_{a \in \mathcal{A}} \left( \pi_h^\star(a|s) - \pi_h(a|s) \right) \cdot Q_{h,P^\star,\mathbf{\Phi}}^{\pi}(s,a)
$$

$$
+ \mathbb{E}_{a \sim \pi_h^\star(\cdot|s), P_h^{\pi^\star,\dagger}(\cdot|s,a)} \left[ V_{h,P^\star,\mathbf{\Phi}}^{\pi} - V_{h,P^\star,\mathbf{\Phi}}^{\pi} \right]. \tag{D.14}
$$

Thus by recursively applying (D.14) over $h \in [H]$, we can conclude that

$$V_{1,P^\star,\mathbf{\Phi}}^{\pi^\star}(s) - V_{1,P^\star,\mathbf{\Phi}}^{\pi}(s)$$
$$\geq \mathbb{E}_{(P^{\pi^\star,\dagger},\pi^\star)}\left[\sum_{h=1}^{H}\sum_{a\in\mathcal{A}}\left(\pi_h^\star(a|s_h) - \pi_h(a|s_h)\right)\cdot Q_{h,P^\star,\mathbf{\Phi}}^{\pi}(s_h,a)\,\middle|\,s_1 = s\right],$$

which completes the proof of Lemma D.1. $\qquad\square$

# E  Proofs for Theoretical Analysis of OPROVI-TV

In this section, we prove our main theoretical results (Theorem 4.3). In Appendix E.1, we outline the proof of the theorem. In Appendix E.2, we list all the key lemmas used in the proof of the theorem. We defer the proof of all the lemmas to subsequent sections (Appendices E.3 to E.8).

Before presenting all the proofs, we define the typical event $\mathcal{E}$ as

$$\mathcal{E} = \left\{ \left|\left(\mathbb{E}_{P_h^\star(\cdot|s,a)} - \mathbb{E}_{\widehat{P}_h^k(\cdot|s,a)}\right)\left[\left(\eta - V_{h+1,P^\star,\mathbf{\Phi}}^\star\right)_+\right]\right| \right.$$
$$\leq \sqrt{\frac{\mathbb{V}_{\widehat{P}_h^k(\cdot|s,a)}\left[\left(\eta - V_{h+1,P^\star,\mathbf{\Phi}}^\star\right)_+\right]\cdot c_1\iota}{N_h^k(s,a)\vee 1}} + \frac{c_2 H\iota}{N_h^k(s,a)\vee 1},$$
$$\left|P_h^\star(s'|s,a) - \widehat{P}_h(s'|s,a)\right| \leq \sqrt{\frac{\min\left\{P_h^\star(s'|s,a), \widehat{P}_h^k(s'|s,a)\right\}\cdot c_1\iota}{N_h^k(s,a)\vee 1}} + \frac{c_2\iota}{N_h^k(s,a)\vee 1},$$
$$\left.\forall(s,a,s',h,k)\in\mathcal{S}\times\mathcal{A}\times\mathcal{S}\times[H]\times[K],\ \forall\eta\in\mathcal{N}_{1/(S\sqrt{K})}\big([0,H]\big)\right\},$$

where $\iota = \log(S^3 A H^2 K^{3/2}/\delta)$, $c_1, c_2 > 0$ are two absolute constants, $\mathcal{N}_{1/S\sqrt{K}}([0,H])$ denotes an $1/S\sqrt{K}$-cover of the interval $[0,H]$.

**Lemma E.1** (Typical event). *For the typical event $\mathcal{E}$ defined in* (E.35)*, it holds that $\mathbb{P}(\mathcal{E}) \geq 1 - \delta$.*

*Proof of Lemma E.1.* This is a direct application of Bernstein inequality and its empirical version (Maurer and Pontil, 2009), together with a union bound over $(s,a,s',h,k,\eta)\in\mathcal{S}\times\mathcal{A}\times\mathcal{S}\times[H]\times[K]\times\mathcal{N}_{1/(S\sqrt{K})}([0,H])$. Note that the size of $\mathcal{N}_{1/(S\sqrt{K})}([0,H])$ is of order $SH\sqrt{K}$. $\qquad\square$

In this section, we always let the event $\mathcal{E}$ hold, which by Lemma E.1 is of probability at least $1 - \delta$.

## E.1  Proof of Theorem 4.3

*Proof of Theorem 4.3.* With Lemma E.2 (optimism and pessimism), we upper bound the regret as

$$\text{Regret}_{\mathbf{\Phi}}(K) = \sum_{k=1}^{K} V_{1,P^\star,\mathbf{\Phi}}^\star(s_1) - V_{1,P^\star,\mathbf{\Phi}}^{\pi^k}(s_1) \leq \sum_{k=1}^{K} \overline{V}_1^k(s_1) - \underline{V}_1^k(s_1). \tag{E.1}$$

In the sequel, we break our proof into three steps.

**Step 1: upper bounding** (E.1). According to the choice of $\overline{Q}_h^k, \underline{Q}_h^k, \overline{V}_h^k, \underline{V}_h^k$ in (4.2), (4.3), and (4.4), let's consider that for any $(h,k)\in[H]\times[K]$ and $(s,a)\in\mathcal{S}\times\mathcal{A}$,

$$\overline{Q}_h^k(s,a) - \underline{Q}_h^k(s,a)$$
$$= \min\left\{R_h(s,a) + \mathbb{E}_{\mathcal{P}_\rho(s,a;\widehat{P}_h^k)}\left[\overline{V}_{h+1}^k\right] + \text{bonus}_h^k(s,a),\ \min\left\{H,\rho^{-1}\right\}\right\}$$

$$- \max\left\{ R_h(s,a) + \mathbb{E}_{\mathcal{P}_\rho(s,a;\widehat{P}_h^k)}\left[\overline{V}_{h+1}^k\right] - \mathtt{bonus}_h^k(s,a), 0 \right\}$$

$$\leq \mathbb{E}_{\mathcal{P}_\rho(s,a;\widehat{P}_h^k)}\left[\overline{V}_{h+1}^k\right] - \mathbb{E}_{\mathcal{P}_\rho(s,a;\widehat{P}_h^k)}\left[\underline{V}_{h+1}^k\right] + 2 \cdot \mathtt{bonus}_h^k(s,a)$$

$$= \underbrace{\mathbb{E}_{\mathcal{P}_\rho(s,a;\widehat{P}_h^k)}\left[\overline{V}_{h+1}^k\right] - \mathbb{E}_{\mathcal{P}_\rho(s,a;P_h^\star)}\left[\overline{V}_{h+1}^k\right] + \mathbb{E}_{\mathcal{P}_\rho(s,a;P_h^\star)}\left[\underline{V}_{h+1}^k\right] - \mathbb{E}_{\mathcal{P}_\rho(s,a;\widehat{P}_h^k)}\left[\underline{V}_{h+1}^k\right]}_{\text{Term (i)}}$$

$$+ \underbrace{\mathbb{E}_{\mathcal{P}_\rho(s,a;P_h^\star)}\left[\overline{V}_{h+1}^k\right] - \mathbb{E}_{\mathcal{P}_\rho(s,a;P_h^\star)}\left[\underline{V}_{h+1}^k\right]}_{\text{Term (ii)}} + 2 \cdot \mathtt{bonus}_h^k(s,a). \tag{E.2}$$

**Step 1.1: upper bounding Term (i).** By using a Bernstein-style concentration argument customized for TV robust expectations (Lemma E.3), we can bound Term (i) by the bonus function, i.e.,

$$\text{Term (i)} \leq 2 \cdot \mathtt{bonus}_h^k(s,a). \tag{E.3}$$

**Step 1.2: upper bounding Term (ii).** By our definition of the operator $\mathbb{E}_{\mathcal{P}_\rho(s,a;P_h^\star)}[V]$ in (4.1), we have

$$\text{Term (ii)} = \sup_{\eta \in [0,H]} \left\{ -\mathbb{E}_{P_h^\star(\cdot|s,a)}\left[\left(\eta - \overline{V}_{h+1}^k\right)_+\right] + \left(1 - \frac{\rho}{2}\right) \cdot \eta \right\}$$

$$- \sup_{\eta \in [0,H]} \left\{ -\mathbb{E}_{P_h^\star(\cdot|s,a)}\left[\left(\eta - \underline{V}_{h+1}^k\right)_+\right] + \left(1 - \frac{\rho}{2}\right) \cdot \eta \right\}$$

$$\leq \sup_{\eta \in [0,H]} \left\{ \mathbb{E}_{P_h^\star(\cdot|s,a)}\left[\left(\eta - \underline{V}_{h+1}^k\right)_+ - \left(\eta - \overline{V}_{h+1}^k\right)_+\right] \right\}. \tag{E.4}$$

By Lemma E.2 which shows that $\overline{V}_{h+1}^k \geq \underline{V}_{h+1}^k$ and the fact that $(\eta - x)_+ - (\eta - y)_+ \leq y - x$ for any $y > x$, we can further upper bound the right hand side of (E.4) by

$$\text{Term (ii)} \leq \mathbb{E}_{P_h^\star(\cdot|s,a)}\left[\overline{V}_{h+1}^k - \underline{V}_{h+1}^k\right]. \tag{E.5}$$

**Step 1.3: combining the upper bounds.** Now combining (E.3) and (E.5) with (E.2), we have that

$$\overline{Q}_h^k(s,a) - \underline{Q}_h^k(s,a) \leq \mathbb{E}_{P_h^\star(\cdot|s,a)}\left[\overline{V}_{h+1}^k - \underline{V}_{h+1}^k\right] + 4 \cdot \mathtt{bonus}_h^k(s,a).$$

By Lemma E.4, we can upper bound the bonus function, and after rearranging terms we further obtain that

$$\overline{Q}_h^k(s,a) - \underline{Q}_h^k(s,a) \leq \left(1 + \frac{12}{H}\right) \cdot \mathbb{E}_{P_h^\star(\cdot|s,a)}\left[\overline{V}_{h+1}^k - \underline{V}_{h+1}^k\right]$$

$$+ 4\sqrt{\frac{\mathbb{V}_{P_h^\star(\cdot|s,a)}\left[V_{h+1,P^\star,\Phi}^{\pi^k}\right] \cdot c_1 \iota}{N_h^k(s,a) \vee 1}} + \frac{4c_2 H^2 S \iota}{N_h^k(s,a) \vee 1} + \frac{4}{\sqrt{K}}, \tag{E.6}$$

where $c_1, c_2 > 0$ are two absolute constants. For the sake of brevity, we introduce the following notations of differences, for any $(h,k) \in [H] \times [K]$,

$$\Delta_h^k := \overline{V}_h^k(s_h^k) - \underline{V}_h^k(s_h^k),$$

$$\zeta_h^k := \Delta_h^k - \left(\overline{Q}_h^k(s_h^k, a_h^k) - \underline{Q}_h^k(s_h^k, a_h^k)\right), \tag{E.7}$$

$$\xi_h^k := \mathbb{E}_{P_h^\star(\cdot|s_h^k, a_h^k)}\left[\overline{V}_h^k - \underline{V}_h^k\right] - \Delta_{h+1}^k. \tag{E.8}$$

If we further define the filtration $\{\mathcal{F}_{h,k}\}_{(h,k) \in [H] \times [K]}$ as

$$\mathcal{F}_{h,k} = \sigma\left(\{(s_i^\tau, a_i^\tau)\}_{(i,\tau) \in [H] \times [k-1]} \bigcup \{(s_i^k, a_i^k)\}_{i \in [h-1]} \bigcup \{s_h^k\}\right),$$

then we can find that $\{\zeta_h^k\}_{(h,k)\in[H]\times[K]}$ is a martingale difference sequence with respect to $\{\mathcal{F}_{h,k}\}_{(h,k)\in[H]\times[K]}$ and $\{\xi_h^k\}_{(h,k)\in[H]\times[K]}$ is a martingale difference sequence with respect to $\{\mathcal{F}_{h,k}\cup\{a_h^k\}\}\}_{(h,k)\in[H]\times[K]}$. Also, we further have that

$$\Delta_h^k = \zeta_h^k + \left(\overline{Q}_h^k(s_h^k,a_h^k) - \underline{Q}_h^k(s_h^k,a_h^k)\right) \tag{E.9}$$

$$\leq \zeta_h^k + \left(1+\frac{12}{H}\right)\cdot \mathbb{E}_{P_h^\star(\cdot|s_h^k,a_h^k)}\left[\overline{V}_{h+1}^k - \underline{V}_{h+1}^k\right]$$

$$+ 4\sqrt{\frac{\mathbb{V}_{P_h^\star(\cdot|s,a)}\left[V_{h+1,P^\star,\Phi}^{\pi^k}\right]\cdot c_1\iota}{N_h^k(s_h^k,a_h^k)\vee 1}} + \frac{4c_2 H^2 S\iota}{N_h^k(s_h^k,a_h^k)\vee 1} + \frac{4}{\sqrt{K}}$$

$$= \zeta_h^k + \left(1+\frac{12}{H}\right)\cdot \xi_h^k + \left(1+\frac{12}{H}\right)\cdot \Delta_{h+1}^k +$$

$$4\sqrt{\frac{\mathbb{V}_{P_h^\star(\cdot|s,a)}\left[V_{h+1,P^\star,\Phi}^{\pi^k}\right]\cdot c_1\iota}{N_h^k(s_h^k,a_h^k)\vee 1}} + \frac{4c_2 H^2 S\iota}{N_h^k(s_h^k,a_h^k)\vee 1} + \frac{4}{\sqrt{K}},$$

where the inequality applies (E.6). Recursively applying (E.9) and using the fact that $(1+\frac{12}{H})^h \leq (1+\frac{12}{H})^H \leq c$ for some absolute constant $c > 0$, we can upper bound the right hand side of (E.1) as

$$\text{Regret}_{\Phi}(K) \leq \sum_{k=1}^K \Delta_1^k \leq C_1 \cdot \sum_{k=1}^K \sum_{h=1}^H (\zeta_h^k + \xi_h^k)$$

$$+ \sqrt{\frac{\mathbb{V}_{P_h^\star(\cdot|s,a)}\left[V_{h+1,P^\star,\Phi}^{\pi^k}\right]\cdot \iota}{N_h^k(s_h^k,a_h^k)\vee 1}} + \frac{H^2 S\iota}{N_h^k(s_h^k,a_h^k)\vee 1} + \frac{1}{\sqrt{K}}. \tag{E.10}$$

where $C_1 > 0$ is an absolute constant.

**Step 2: controlling the summation of variance terms.** In view of (E.10), it suffices to upper bound its right hand side. The key difficulty is the analysis of the summation of the variance terms, which we focus on now. By Cauchy-Schwartz inequality,

$$\sum_{k=1}^K \sum_{h=1}^H \sqrt{\frac{\mathbb{V}_{P_h^\star(\cdot|s_h^k,a_h^k)}\left[V_{h+1,P^\star,\Phi}^{\pi^k}\right]}{N_h^k(s_h^k,a_h^k)\vee 1}}$$

$$\leq \sqrt{\sum_{k=1}^K \sum_{h=1}^H \mathbb{V}_{P_h^\star(\cdot|s_h^k,a_h^k)}\left[V_{h+1,P^\star,\Phi}^{\pi^k}\right] \cdot \sum_{k=1}^K \sum_{h=1}^H \frac{1}{N_h^k(s_h^k,a_h^k)\vee 1}}. \tag{E.11}$$

On the right hand side of (E.11), the summation of the inverse of the count function is a well bounded term (Lemma E.13). So the key is to upper bound the the summation of the variance of the robust value functions to obtain a sharp bound. To this end, we invoke Lemma E.5 to obtain that with probability at least $1-\delta$,

$$\sum_{k=1}^K \sum_{h=1}^H \mathbb{V}_{P_h^\star(\cdot|s_h^k,a_h^k)}\left[V_{h+1,P^\star,\Phi}^{\pi^k}\right]$$

$$\leq C_2 \cdot \left(\min\{H,\rho^{-1}\}\cdot HK + \min\{H,\rho^{-1}\}^3 \cdot H\iota\right), \tag{E.12}$$

where $C_2 > 0$ is an absolute constant. With inequality (E.12) and Lemma E.13 that

$$\sum_{k=1}^K \sum_{h=1}^H \frac{1}{N_h^k(s_h^k,a_h^k)\vee 1} \leq C_2' \cdot HSA\iota,$$

with $C_2' > 0$ being another constant, we can upper bound the summation of the variance terms (E.11) as

$$\sum_{k=1}^K \sum_{h=1}^H \sqrt{\frac{\mathbb{V}_{P_h^\star(\cdot|s_h^k,a_h^k)}\left[V_{h+1,P^\star,\Phi}^{\pi^k}\right]}{N_h^k(s_h^k,a_h^k)\vee 1}}$$

$$\leq C_3 \sqrt{\min\left\{H, \rho^{-1}\right\} \cdot H^2 SAK\iota + \min\left\{H, \rho^{-1}\right\}^3 \cdot H^2 SA\iota^2}. \tag{E.13}$$

where $C_3 > 0$ is also an absolute constant.

**Step 3: finishing the proof.** With (E.10) and (E.13), it suffices to control the remaining terms. For the summation of the martingale difference terms, notice that by the definitions in (E.7) and (E.8), both $\zeta_h^k$ and $\xi_h^k$ are bounded by $\min\{H, \rho^{-1}\}$ according to (4.2) and Lemma E.2 (optimism and pessimism). As a result, using Azuma-Hoeffding inequality, with probability at least $1 - \delta$

$$\sum_{k=1}^{K}\sum_{h=1}^{H}(\zeta_h^k + \xi_h^k) \leq C_4 \cdot \min\left\{H, \rho^{-1}\right\} \cdot \sqrt{HK\iota},$$

where $C_4 > 0$ is an absolute constant. For the summation of the inverse of the count function in (E.10), it suffices to invoke again Lemma E.13. Combining all together, with probability at least $1 - 3\delta$, we have

$$\mathrm{Regret}_{\mathbf{\Phi}}(K) \leq C_5 \cdot \Bigg( \sqrt{\min\left\{H, \rho^{-1}\right\} \cdot H^2 SAK\iota^2 + \min\left\{H, \rho^{-1}\right\}^3 \cdot H^2 SA\iota^3}$$

$$+ \min\left\{H, \rho^{-1}\right\} \cdot \sqrt{HK\iota} + H^3 S^2 A\iota^2 + H\sqrt{K} \Bigg)$$

$$= \mathcal{O}\left(\sqrt{\min\left\{H, \rho^{-1}\right\} \cdot H^2 SAK\iota'}\right),$$

where $C_5 > 0$ is an absolute constant and $\iota' = \log^2(SAHK/\delta)$. This completes the proof of Theorem 4.3. $\qquad\square$

### E.2 Key Lemmas

**Lemma E.2** (Optimistic and pessimistic estimation of the robust values). *By setting the $\mathrm{bonus}_h^k$ as in (4.5), then under the typical event $\mathcal{E}$, it holds that*

$$\underline{Q}_h^k(s,a) \leq Q_{h,P^\star,\mathbf{\Phi}}^{\pi^k}(s,a) \leq Q_{h,P^\star,\mathbf{\Phi}}^\star(s,a) \leq \overline{Q}_h^k(s,a),$$

$$\underline{V}_h^k(s) \leq V_{h,P^\star,\mathbf{\Phi}}^{\pi^k}(s) \leq V_{h,P^\star,\mathbf{\Phi}}^\star(s) \leq \overline{V}_h^k(s), \tag{E.14}$$

*for any $(s,a,h,k) \in \mathcal{S} \times \mathcal{A} \times [H] \times [K]$.*

*Proof of Lemma E.2.* See Appendix E.3 for a detailed proof. $\qquad\square$

**Lemma E.3** (Proper bonus for TV robust sets and optimistic and pessimistic value estimators). *By setting the $\mathrm{bonus}_h^k$ as in (4.5), then under the typical event $\mathcal{E}$, it holds that*

$$\mathbb{E}_{\mathcal{P}_\rho(s,a;\widehat{P}_h^k)}\left[\overline{V}_{h+1}^k\right] - \mathbb{E}_{\mathcal{P}_\rho(s,a;P_h^\star)}\left[\overline{V}_{h+1}^k\right] + \mathbb{E}_{\mathcal{P}_\rho(s,a;P_h^\star)}\left[\underline{V}_{h+1}^k\right] - \mathbb{E}_{\mathcal{P}_\rho(s,a;\widehat{P}_h^k)}\left[\underline{V}_{h+1}^k\right]$$

$$\leq 2 \cdot \mathrm{bonus}_h^k(s,a),$$

*Proof of Lemma E.3.* See Appendix E.4 for a detailed proof. $\qquad\square$

**Lemma E.4** (Control of the bonus term). *Under the typical event $\mathcal{E}$, the $\mathrm{bonus}_h^k$ in (4.5) is bounded by*

$\mathrm{bonus}_h^k(s,a)$

$$\leq \sqrt{\frac{\mathbb{V}_{P_h^\star(\cdot|s,a)}\left[V_{h+1,P^\star,\mathbf{\Phi}}^{\pi^k}\right] \cdot c_1\iota}{N_h^k(s,a) \vee 1}} + \frac{4 \cdot \mathbb{E}_{P_h^\star(\cdot|s,a)}\left[\overline{V}_{h+1}^k - \underline{V}_{h+1}^k\right]}{H} + \frac{c_2 H^2 S\iota}{N_h^k(s,a) \vee 1} + \frac{1}{\sqrt{K}},$$

*where $\iota = \log(S^3 AH^2 K^{3/2}/\delta)$ and $c_1, c_2 > 0$ are absolute constants.*

*Proof of Lemma E.4.* See Appendix E.5 for a detailed proof. $\qquad\square$

**Lemma E.5** (Total variance law for robust MDP with TV robust sets). *With probability at least $1 - \delta$, the following inequality holds*

$$\sum_{k=1}^{K} \sum_{h=1}^{H} \mathbb{V}_{P_h^\star(\cdot|s_h^k, a_h^k)}\left[V_{h+1, P^\star, \mathbf{\Phi}}^{\pi^k}\right] \leq c_3 \cdot \left(\min\{H, \rho^{-1}\} \cdot HK + \min\{H, \rho^{-1}\}^3 \cdot H\iota\right).$$

*where $\iota = \log(S^3 A H^2 K^{3/2}/\delta)$ and $c_3 > 0$ is an absolute constant.*

*Proof of Lemma E.5.* See Appendix E.6 for a detailed proof. □

### E.3   Proof of Lemma E.2

*Proof of Lemma E.2.* We prove Lemma E.2 by induction. Suppose the conclusion (E.14) holds at step $h + 1$. For step $h$, let's first consider the robust $Q$ function part. Specifically, by using the robust Bellman optimal equation (Proposition 2.3) and (4.2), we have that

$$Q_{h, P^\star, \mathbf{\Phi}}^\star(s, a) - \overline{Q}_h^k(s, a)$$
$$\leq \max\left\{\mathbb{E}_{\mathcal{P}_\rho(s, a; P_h^\star)}\left[V_{h+1, P^\star, \mathbf{\Phi}}^\star\right] - \mathbb{E}_{\mathcal{P}_\rho(s, a; \widehat{P}_h^k)}\left[\overline{V}_{h+1}^k\right] - \texttt{bonus}_h^k(s, a),\right.$$
$$\left. Q_{h, P^\star, \mathbf{\Phi}}^\star(s, a) - \min\left\{H, \rho^{-1}\right\}\right\}$$
$$\leq \max\left\{\mathbb{E}_{\mathcal{P}_\rho(s, a; P_h^\star)}\left[V_{h+1, P^\star, \mathbf{\Phi}}^\star\right] - \mathbb{E}_{\mathcal{P}_\rho(s, a; \widehat{P}_h^k)}\left[V_{h+1, P^\star, \mathbf{\Phi}}^\star\right] - \texttt{bonus}_h^k(s, a), 0\right\}, \text{(E.15)}$$

where the second inequality follows from the induction of $V_{h+1, P^\star, \mathbf{\Phi}}^\star \leq \overline{V}_{h+1}^k$ at step $h + 1$ and the fact that $Q_{h, P^\star, \mathbf{\Phi}}^\star \leq \min\{H, \rho^{-1}\}$ (by Proposition 2.6 and Assumption 4.1). By Lemma E.7, we have that

$$\mathbb{E}_{\mathcal{P}_\rho(s, a; P_h^\star)}\left[V_{h+1, P^\star, \mathbf{\Phi}}^\star\right] - \mathbb{E}_{\mathcal{P}_\rho(s, a; \widehat{P}_h^k)}\left[V_{h+1, P^\star, \mathbf{\Phi}}^\star\right]$$
$$\leq \sqrt{\frac{\mathbb{V}_{\widehat{P}_h^k(\cdot|s, a)}\left[V_{h+1, P^\star, \mathbf{\Phi}}^\star\right] \cdot c_1 \iota}{N_h^k(s, a) \vee 1}} + \frac{c_2 H \iota}{N_h^k(s, a) \vee 1} + \frac{1}{\sqrt{K}},$$

Now by further applying Lemma E.11 to the variance term in the above inequality, we can obtain that

$$\mathbb{E}_{\mathcal{P}_\rho(s, a; P_h^\star)}\left[V_{h+1, P^\star, \mathbf{\Phi}}^\star\right] - \mathbb{E}_{\mathcal{P}_\rho(s, a; \widehat{P}_h^k)}\left[V_{h+1, P^\star, \mathbf{\Phi}}^\star\right]$$
$$\leq \sqrt{\frac{\left(\mathbb{V}_{\widehat{P}_h^k(\cdot|s, a)}\left[\left(\overline{V}_{h+1}^k + \underline{V}_{h+1}^k\right)/2\right] + 4H \cdot \mathbb{E}_{\widehat{P}_h^k(\cdot|s, a)}\left[\overline{V}_{h+1}^k - \underline{V}_{h+1}^k\right]\right) \cdot c_1 \iota}{N_h^k(s, a) \vee 1}}$$
$$\quad + \frac{c_2 H \iota}{N_h^k(s, a) \vee 1} + \frac{1}{\sqrt{K}}$$
$$\leq \sqrt{\frac{\mathbb{V}_{\widehat{P}_h^k(\cdot|s, a)}\left[\left(\overline{V}_{h+1}^k + \underline{V}_{h+1}^k\right)/2\right] \cdot c_1 \iota}{N_h^k(s, a) \vee 1}} + \sqrt{\frac{\mathbb{E}_{\widehat{P}_h^k(\cdot|s, a)}\left[\overline{V}_{h+1}^k - \underline{V}_{h+1}^k\right] \cdot 4H c_1 \iota}{N_h^k(s, a) \vee 1}}$$
$$\quad + \frac{c_2 H \iota}{N_h^k(s, a) \vee 1} + \frac{1}{\sqrt{K}}$$
$$\leq \sqrt{\frac{\mathbb{V}_{\widehat{P}_h^k(\cdot|s, a)}\left[\left(\overline{V}_{h+1}^k + \underline{V}_{h+1}^k\right)/2\right] \cdot c_1 \iota}{N_h^k(s, a) \vee 1}} + \frac{\mathbb{E}_{\widehat{P}_h^k(\cdot|s, a)}\left[\overline{V}_{h+1}^k - \underline{V}_{h+1}^k\right]}{H}$$
$$\quad + \frac{c_2' H^2 \iota}{N_h^k(s, a) \vee 1} + \frac{1}{\sqrt{K}}, \tag{E.16}$$

where the first inequality is due to Lemma E.11, the second inequality is due to $\sqrt{a+b} \leq \sqrt{a} + \sqrt{b}$, and the last inequality is from $\sqrt{ab} \leq a + b$ where $c_2' > 0$ is an absolute constant. Therefore, combining (E.15) and (E.16), and the choice of $\texttt{bonus}_h^k(s, a)$ in (4.5), we can conclude that

$$Q_{h, P^\star, \boldsymbol{\Phi}}^\star(s, a) \leq \overline{Q}_h^k(s, a).$$

Furthermore, it holds that $Q_{h, P^\star, \boldsymbol{\Phi}}^{\pi^k}(s, a) \leq Q_{h, P^\star, \boldsymbol{\Phi}}^\star(s, a)$. Thus it reduces to prove $\underline{Q}_h^k(s, a) \leq Q_{h, P^\star, \boldsymbol{\Phi}}^{\pi^k}(s, a)$. Again, by using the robust Bellman equation (Proposition 2.2) and (4.3), we have that

$$
\begin{aligned}
&\underline{Q}_h^k(s, a) - Q_{h, P^\star, \boldsymbol{\Phi}}^{\pi^k}(s, a) \\
&\leq \max \Bigg\{ \mathbb{E}_{\mathcal{P}_\rho(s, a; \widehat{P}_h^k)} \left[ \underline{V}_{h+1}^k \right] - \mathbb{E}_{\mathcal{P}_\rho(s, a; P_h^\star)} \left[ V_{h+1, P^\star, \boldsymbol{\Phi}}^{\pi^k} \right] - \texttt{bonus}_h^k(s, a), \\
&\qquad\quad 0 - Q_{h, P^\star, \boldsymbol{\Phi}}^{\pi^k}(s, a) \Bigg\} \\
&\leq \max \Bigg\{ \mathbb{E}_{\mathcal{P}_\rho(s, a; \widehat{P}_h^k)} \left[ V_{h+1, P^\star, \boldsymbol{\Phi}}^{\pi^k} \right] - \mathbb{E}_{\mathcal{P}_\rho(s, a; P_h^\star)} \left[ V_{h+1, P^\star, \boldsymbol{\Phi}}^{\pi^k} \right] - \texttt{bonus}_h^k(s, a), 0 \Bigg\} \quad\text{(E.17)}
\end{aligned}
$$

where the second inequality follows from the induction of $\underline{V}_{h+1}^k \leq V_{h+1, P^\star, \boldsymbol{\Phi}}^{\pi^k}$ at step $h+1$ and the fact that $Q_{h, P^\star, \boldsymbol{\Phi}}^{\pi^k} \geq 0$. By Lemma E.8, we have that

$$
\begin{aligned}
&\mathbb{E}_{\mathcal{P}_\rho(s, a; \widehat{P}_h^k)} \left[ V_{h+1, P^\star, \boldsymbol{\Phi}}^{\pi^k} \right] - \mathbb{E}_{\mathcal{P}_\rho(s, a; P_h^\star)} \left[ V_{h+1, P^\star, \boldsymbol{\Phi}}^{\pi^k} \right] \\
&\leq \sqrt{ \frac{ \mathbb{V}_{\widehat{P}_h^k(\cdot|s, a)} \left[ V_{h+1, P^\star, \boldsymbol{\Phi}}^\star \right] \cdot c_1 \iota }{ N_h^k(s, a) \vee 1 } } + \frac{ \mathbb{E}_{\widehat{P}_h^k(\cdot|s, a)} \left[ \overline{V}_{h+1}^k - \underline{V}_{h+1}^k \right] }{H} + \frac{ c_2' H^2 S \iota }{ N_h^k(s, a) \vee 1 } + \frac{1}{\sqrt{K}}.
\end{aligned}
$$

Now by applying Lemma E.11 to the variance term, with an argument similar to (E.16), we can obtain that

$$
\begin{aligned}
&\mathbb{E}_{\mathcal{P}_\rho(s, a; \widehat{P}_h^k)} \left[ V_{h+1, P^\star, \boldsymbol{\Phi}}^{\pi^k} \right] - \mathbb{E}_{\mathcal{P}_\rho(s, a; P_h^\star)} \left[ V_{h+1, P^\star, \boldsymbol{\Phi}}^{\pi^k} \right] \quad\text{(E.18)} \\
&\leq \sqrt{ \frac{ \mathbb{V}_{\widehat{P}_h^k(\cdot|s, a)} \left[ \left( \overline{V}_{h+1}^k + \underline{V}_{h+1}^k \right)/2 \right] \cdot c_1 \iota }{ N_h^k(s, a) \vee 1 } } + \frac{ 2 \mathbb{E}_{\widehat{P}_h^k(\cdot|s, a)} \left[ \overline{V}_{h+1}^k - \underline{V}_{h+1}^k \right] }{H} \\
&\quad + \frac{ c_2'' H^2 \iota }{ N_h^k(s, a) \vee 1 } + \frac{1}{\sqrt{K}},
\end{aligned}
$$

Thus by combining (E.17) and (E.18), and the choice of $\texttt{bonus}_h^k(s, a)$ in (4.5), we can conclude that

$$\underline{Q}_h^k(s, a) \leq Q_{h, P^\star, \boldsymbol{\Phi}}^{\pi^k}(s, a).$$

Therefore, we have proved that at step $h$, it holds that

$$\underline{Q}_h^k(s, a) \leq Q_{h, P^\star, \boldsymbol{\Phi}}^{\pi^k}(s, a) \leq Q_{h, P^\star, \boldsymbol{\Phi}}^\star(s, a) \leq \overline{Q}_h^k(s, a).$$

Finally for the robust $V$ function part, consider that by robust Bellman equation (Proposition 2.2) and (4.4),

$$\underline{V}_h^k(s) = \mathbb{E}_{\pi_h^k(\cdot|s)} \left[ \underline{Q}_h^k(s, \cdot) \right] \leq \mathbb{E}_{\pi_h^k(\cdot|s)} \left[ Q_{h, P^\star, \boldsymbol{\Phi}}^{\pi^k}(s, \cdot) \right] = V_{h, P^\star, \boldsymbol{\Phi}}^{\pi^k}(s),$$

and that by robust Bellman optimal equation (Proposition 2.3), the choice of $\pi^k$, and (4.4),

$$V_{h, P^\star, \boldsymbol{\Phi}}^\star(s) = \max_{a \in \mathcal{A}} Q_{h, P^\star, \boldsymbol{\Phi}}^\star(s, a) \leq \max_{a \in \mathcal{A}} \overline{Q}_h^k(s, a) = \overline{V}_h^k(s),$$

which proves that

$$\underline{V}_h^k(s) \leq V_{h, P^\star, \boldsymbol{\Phi}}^{\pi^k}(s) \leq V_{h, P^\star, \boldsymbol{\Phi}}^\star(s) \leq \overline{V}_h^k(s).$$

Since the conclusion (E.14) holds for the $V$ function part at step $H + 1$, an induction proves Lemma E.2. $\qquad\square$

## E.4 Proof of Lemma E.3

*Proof of Lemma E.3.* We upper bound the differences by a concentration inequality Lemma E.9,

$$\mathbb{E}_{\mathcal{P}_\rho(s,a;\widehat{P}_h^k)}\left[\overline{V}_{h+1}^k\right] - \mathbb{E}_{\mathcal{P}_\rho(s,a;P_h^\star)}\left[\overline{V}_{h+1}^k\right] + \mathbb{E}_{\mathcal{P}_\rho(s,a;\widehat{P}_h^k)}\left[\underline{V}_{h+1}^k\right] - \mathbb{E}_{\mathcal{P}_\rho(s,a;P_h^\star)}\left[\underline{V}_{h+1}^k\right]$$

$$\leq 2\sqrt{\frac{\mathbb{V}_{\widehat{P}_h^k(\cdot|s,a)}\left[V_{h+1,P^\star,\boldsymbol{\Phi}}^\star\right] \cdot c_1\iota}{N_h^k(s,a)\vee 1}} + \frac{2\cdot\mathbb{E}_{\widehat{P}_h^k(\cdot|s,a)}\left[\overline{V}_{h+1}^k - \underline{V}_{h+1}^k\right]}{H}$$

$$+ \frac{2c_2'H^2 S\iota}{N_h^k(s,a)\vee 1} + \frac{2}{\sqrt{K}}, \tag{E.19}$$

where $c_1, c_2' > 0$ are absolute constants. Then applying Lemma E.11 to the variance term in (E.19), with an argument the same as (E.16) in the proof of Lemma E.2, we can obtain that

$$\mathbb{E}_{\mathcal{P}_\rho(s,a;\widehat{P}_h^k)}\left[\overline{V}_{h+1}^k\right] - \mathbb{E}_{\mathcal{P}_\rho(s,a;P_h^\star)}\left[\overline{V}_{h+1}^k\right] + \mathbb{E}_{\mathcal{P}_\rho(s,a;\widehat{P}_h^k)}\left[\underline{V}_{h+1}^k\right] - \mathbb{E}_{\mathcal{P}_\rho(s,a;P_h^\star)}\left[\underline{V}_{h+1}^k\right]$$

$$\leq 2\sqrt{\frac{\mathbb{V}_{\widehat{P}_h^k(\cdot|s,a)}\left[\left(\overline{V}_{h+1}^k + \underline{V}_{h+1}^k\right)/2\right] \cdot c_1\iota}{N_h^k(s,a)\vee 1}} + \frac{4\cdot\mathbb{E}_{\widehat{P}_h^k(\cdot|s,a)}\left[\overline{V}_{h+1}^k - \underline{V}_{h+1}^k\right]}{H}$$

$$+ \frac{2c_2''H^2\iota}{N_h^k(s,a)\vee 1} + \frac{2}{\sqrt{K}}.$$

Therefore, by looking into the choice of $\texttt{bonus}_h^k(s,a)$ in (4.5), we can conclude that

$$\mathbb{E}_{\mathcal{P}_\rho(s,a;\widehat{P}_h^k)}\left[\overline{V}_{h+1}^k\right] - \mathbb{E}_{\mathcal{P}_\rho(s,a;P_h^\star)}\left[\overline{V}_{h+1}^k\right] + \mathbb{E}_{\mathcal{P}_\rho(s,a;\widehat{P}_h^k)}\left[\underline{V}_{h+1}^k\right] - \mathbb{E}_{\mathcal{P}_\rho(s,a;P_h^\star)}\left[\underline{V}_{h+1}^k\right]$$

$$\leq 2\cdot\texttt{bonus}_h^k(s,a),$$

This finishes the proof of Lemma E.3. □

## E.5 Proof of Lemma E.4

*Proof of Lemma E.4.* Recall that the $\texttt{bonus}_h^k(s,a)$ is defined as

$$\texttt{bonus}_h^k(s,a) = \sqrt{\frac{\mathbb{V}_{\widehat{P}_h^k(\cdot|s,a)}\left[\left(\overline{V}_{h+1}^k + \underline{V}_{h+1}^k\right)/2\right] \cdot c_1\iota}{N_h^k(s,a)\vee 1}}$$

$$+ \frac{2\mathbb{E}_{\widehat{P}_h^k(\cdot|s,a)}\left[\overline{V}_{h+1}^k - \underline{V}_{h+1}^k\right]}{H} + \frac{c_2 H^2 S\iota}{N_h^k(s,a)\vee 1} + \frac{1}{\sqrt{K}}.$$

The main thing we need to consider is to control the first term and the second term. We first deal with the second term of $\texttt{bonus}_h^k(s,a)$ by invoking Lemma E.10, which gives

$$\frac{2\mathbb{E}_{\widehat{P}_h^k(\cdot|s,a)}\left[\overline{V}_{h+1}^k - \underline{V}_{h+1}^k\right]}{H} \leq \left(\frac{2}{H} + \frac{2}{H^2}\right)\cdot\mathbb{E}_{P_h^\star(\cdot|s,a)}\left[\overline{V}_{h+1}^k - \underline{V}_{h+1}^k\right] + \frac{c_2'HS\iota}{N_h^k(s,a)\vee 1}$$

$$\leq \frac{3\mathbb{E}_{P_h^\star(\cdot|s,a)}\left[\overline{V}_{h+1}^k - \underline{V}_{h+1}^k\right]}{H} + \frac{c_2'HS\iota}{N_h^k(s,a)\vee 1}, \tag{E.20}$$

where the second inequality is from $H \geq 2$. Then we deal with the first term (variance term) of $\texttt{bonus}_h^k(s,a)$ by invoking Lemma E.12, which gives

$$\sqrt{\frac{\mathbb{V}_{\widehat{P}_h^k(\cdot|s,a)}\left[\left(\overline{V}_{h+1}^k + \underline{V}_{h+1}^k\right)/2\right] \cdot c_1 \iota}{N_h^k(s,a) \vee 1}} \tag{E.21}$$

$$\leq \sqrt{\frac{\left(\mathbb{V}_{P_h^\star(\cdot|s,a)}\left[V_{h+1,P^\star,\Phi}^{\pi^k}\right] + 4H \cdot \mathbb{E}_{P_h^\star(\cdot|s,a)}\left[\overline{V}_{h+1}^k - \underline{V}_{h+1}^k\right] + \frac{c_2'' H^4 S \iota}{N_h^k(s,a) \vee 1} + 1\right) \cdot c_1 \iota}{N_h^k(s,a)}}$$

$$\leq \sqrt{\frac{\mathbb{V}_{P_h^\star(\cdot|s,a)}\left[V_{h+1,P^\star,\Phi}^{\pi^k}\right] \cdot c_1 \iota}{N_h^k(s,a) \vee 1}} + \sqrt{\frac{4H \cdot \mathbb{E}_{P_h^\star(\cdot|s,a)}\left[\overline{V}_{h+1}^k - \underline{V}_{h+1}^k\right] \cdot c_1 \iota}{N_h^k(s,a) \vee 1}}$$

$$+ \frac{\sqrt{c_1 c_2'' S} H^2 \iota}{N_h^k(s,a) \vee 1} + \sqrt{\frac{c_1 \iota}{N_h^k(s,a) \vee 1}}$$

$$\leq \sqrt{\frac{\mathbb{V}_{P_h^\star(\cdot|s,a)}\left[V_{h+1,P^\star,\Phi}^{\pi^k}\right] \cdot c_1' \iota}{N_h^k(s,a) \vee 1}} + \frac{\mathbb{E}_{P_h^\star(\cdot|s,a)}\left[\overline{V}_{h+1}^k - \underline{V}_{h+1}^k\right]}{H} + \frac{\left(4c_1 + \sqrt{c_1 c_2'' S}\right) H^2 \iota}{N_h^k(s,a) \vee 1}$$

Thus by combining (E.20) and (E.21) with the choice of $\texttt{bonus}_h^k$, we can conclude the proof of Lemma E.4. $\qquad\square$

## E.6 Proof of Lemma E.5

*Proof of Lemma E.5.* The key idea is to relate the visitation distribution (w.r.t. $P^\star$) and the variance (w.r.t. $P^\star$) to the value function of $\pi^k$, after which we can derive an upper bound for the total variance. Throughout this proof, we use the shorthand that

$$\overline{H} = \min\left\{H, \rho^{-1}\right\}.$$

According to Proposition 2.6 and Assumption 4.1, for any policy $\pi$ and any step $h$, the robust value function of $\pi$ holds that

$$\max_{s \in \mathcal{S}} V_{h,P^\star,\Phi}^\pi(s) \leq \overline{H}, \tag{E.22}$$

which we usually apply in the sequel. Also, to facilitate our analysis, we define

$$\widetilde{T}_h^k(\cdot|s,a) = \operatorname*{argmin}_{P(\cdot) \in \mathcal{P}_h(s,a;P_h^\star)} \mathbb{E}_{P(\cdot)}\left[V_{h+1,P^\star,\Phi}^{\pi^k}\right], \quad \forall(s,a,h) \in \mathcal{S} \times \mathcal{A} \times [H],$$

and set $\widetilde{T}^k = \{\widetilde{T}_h^k\}_{h=1}^H$, which is the most adversarial transition for the true robust value function of $\pi^k$.

Now consider the following decomposition of our target,

$$\sum_{k=1}^K \sum_{h=1}^H \mathbb{V}_{P_h^\star(\cdot|s_h^k,a_h^k)}\left[V_{h+1,P^\star,\Phi}^{\pi^k}\right]$$

$$= \sum_{k=1}^K \sum_{h=1}^H \mathbb{V}_{P_h^\star(\cdot|s_h^k,a_h^k)}\left[V_{h+1,P^\star,\Phi}^{\pi^k}\right] - \mathbb{E}_{(s_h^k,a_h^k)\sim(P^\star,\pi^k)}\left[\sum_{h=1}^H \mathbb{V}_{P_h^\star(\cdot|s_h^k,a_h^k)}\left[V_{h+1,P^\star,\Phi}^{\pi^k}\right]\right]$$

$$+ \sum_{k=1}^K \mathbb{E}_{(s_h^k,a_h^k)\sim(P^\star,\pi^k)}\left[\sum_{h=1}^H \mathbb{V}_{P_h^\star(\cdot|s_h^k,a_h^k)}\left[V_{h+1,P^\star,\Phi}^{\pi^k}\right]\right]$$

$$= \underbrace{\sum_{k=1}^K \sum_{h=1}^H \mathbb{V}_{P_h^\star(\cdot|s_h^k,a_h^k)}\left[V_{h+1,P^\star,\Phi}^{\pi^k}\right] - \mathbb{E}_{(s_h^k,a_h^k)\sim(P^\star,\pi^k)}\left[\sum_{h=1}^H \mathbb{V}_{P_h^\star(\cdot|s_h^k,a_h^k)}\left[V_{h+1,P^\star,\Phi}^{\pi^k}\right]\right]}_{\text{Term (i): martingale difference term}}$$

$$+ \sum_{k=1}^{K} \mathbb{E}_{(s_h^k, a_h^k) \sim (\widetilde{T}^k, \pi^k)} \left[ \sum_{h=1}^{H} \mathbb{V}_{\widetilde{T}_h^k(\cdot | s_h^k, a_h^k)} \left[ V_{h+1, P^\star, \Phi}^{\pi^k} \right] \right]$$

Term (ii): total variance law

$$+ \sum_{k=1}^{K} \mathbb{E}_{(s_h^k, a_h^k) \sim (P^\star, \pi^k)} \left[ \sum_{h=1}^{H} \mathbb{V}_{P_h^\star(\cdot | s_h^k, a_h^k)} \left[ V_{h+1, P^\star, \Phi}^{\pi^k} \right] \right] - \mathbb{E}_{(s_h^k, a_h^k) \sim (\widetilde{T}^k, \pi^k)} \left[ \sum_{h=1}^{H} \mathbb{V}_{\widetilde{T}_h^k(\cdot | s_h^k, a_h^k)} \left[ V_{h+1, P^\star, \Phi}^{\pi^k} \right] \right].$$

Term (iii): error from $P^\star$ to $\widetilde{T}^k$

In the sequel, we upper bound each of the three terms respectively.

**Term (i): martingale difference term.** This is a summation of martingale difference term (with respect to filtration $\mathcal{G}_k = \sigma(\{(s_h^\tau, a_h^\tau)\}_{(h, \tau) \in [H] \times [k]})$). By Azuma-Hoeffding's inequality, with probability at least $1 - \delta$,

$$\text{Term (i)} \le c \cdot H \cdot \overline{H}^2 \cdot \sqrt{K \iota}, \tag{E.23}$$

where $c > 0$ is an absolute constant. We have utilized the fact of (E.22) to obtain the upper bound $H \overline{H}^2$ on each martingale difference term in the summation.

**Term (ii): total variance law.** The upper bound of this term is the core part of the analysis, for which we summarize it in the following lemma.

**Lemma E.6** (Total variance law). *Under the same setup as Theorem 4.3, given any deterministic policy $\pi$, define that*

$$\widetilde{T}_h(\cdot | s, a) = \operatorname*{argmin}_{P(\cdot) \in \mathcal{P}_\rho(s, a; P_h^\star)} \mathbb{E}_{P(\cdot)} \left[ V_{h+1, P^\star, \Phi}^\pi \right], \quad \forall (s, a, h) \in \mathcal{S} \times \mathcal{A} \times [H], \tag{E.24}$$

*and set $\widetilde{T} = \{\widetilde{T}_h\}_{h=1}^H$. Then we have*

$$\mathbb{E}_{(s_h, a_h) \sim (\widetilde{T}, \pi)} \left[ \sum_{h=1}^{H} \mathbb{V}_{\widetilde{T}_h(\cdot | s_h, a_h)} \left[ V_{h+1, P^\star, \Phi}^\pi \right] \right] \le 2H \cdot \overline{H}.$$

We defer the proof of Lemma E.6 to Appendix E.7. With Lemma E.6, we consider taking policy $\pi = \pi^k$ for $k \in [K]$ therein (which are deterministic policies), and obtain that the Term (ii) is upper bounded by

$$\text{Term (ii)} \le 2H \cdot \overline{H} \cdot K. \tag{E.25}$$

**Term (iii): error from $P^\star$ to $\widetilde{T}^k$.** We first relate the visitation distribution under $P^\star$ to that under $\widetilde{T}^k$. On the one hand, by the choice of the adversarial transition kernel $\widetilde{T}_h^k$, it holds that

$$D_{\mathrm{TV}} \left( P_h^\star(\cdot | s, a) \, \middle\| \, \widetilde{T}_h^k(\cdot | s, a) \right) \le \rho, \quad \forall (s, a, h) \in \mathcal{S} \times \mathcal{A} \times [H]. \tag{E.26}$$

On the other hand, by (E.22), we can upper bound the variance term by

$$\mathbb{V}_{P_h^\star(\cdot | s, a)} \left[ V_{h+1, P^\star, \Phi}^{\pi^k} \right] \le \overline{H}^2, \quad \forall (s, a, h) \in \mathcal{S} \times \mathcal{A} \times [H]. \tag{E.27}$$

Therefore, by combining (E.26) and (E.27), we can conclude that

$$\mathbb{E}_{(s_h^k, a_h^k) \sim (P_h^\star, \pi^k)} \left[ \sum_{h=1}^{H} \mathbb{V}_{P^\star(\cdot | s_h^k, a_h^k)} \left[ V_{h+1, P^\star, \Phi}^{\pi^k} \right] \right]$$

$$\le \mathbb{E}_{(s_h^k, a_h^k) \sim (\widetilde{T}^k, \pi^k)} \left[ \sum_{h=1}^{H} \mathbb{V}_{P_h^\star(\cdot | s_h^k, a_h^k)} \left[ V_{h+1, P^\star, \Phi}^{\pi^k} \right] \right]$$

$$+ H \cdot \left( \sup_{(s,a,h) \in \mathcal{S} \times \mathcal{A} \times [H]} D_{\mathrm{TV}} \left( P_h^\star(\cdot \mid s, a) \,\middle\|\, \widetilde{T}_h^k(\cdot \mid s, a) \right) \right.$$

$$\left. \cdot \sup_{(s,a,h) \in \mathcal{S} \times \mathcal{A} \times [H]} \mathbb{V}_{P_h^\star(\cdot|s,a)} \left[ V_{h+1,P^\star,\boldsymbol{\Phi}}^{\pi^k} \right] \right)$$

$$\leq \mathbb{E}_{(s_h^k, a_h^k) \sim (\widetilde{T}^k, \pi^k)} \left[ \sum_{h=1}^H \mathbb{V}_{P_h^\star(\cdot|s_h^k, a_h^k)} \left[ V_{h+1,P^\star,\boldsymbol{\Phi}}^{\pi^k} \right] \right] + \rho H \cdot \overline{H}^2. \qquad \text{(E.28)}$$

We then relate the variance term under $P^\star$ to that under $\widetilde{T}^k$. Specifically, we have

$$\mathbb{V}_{P_h^\star(\cdot|s,a)} \left[ V_{h+1,P^\star,\boldsymbol{\Phi}}^{\pi^k} \right] = \mathbb{E}_{P_h^\star(\cdot|s,a)} \left[ \left( V_{h+1,P^\star,\boldsymbol{\Phi}}^{\pi^k} \right)^2 \right] - \left( \mathbb{E}_{P_h^\star(\cdot|s,a)} \left[ V_{h+1,P^\star,\boldsymbol{\Phi}}^{\pi^k} \right] \right)^2$$

$$\leq \mathbb{E}_{\widetilde{T}_h^k(\cdot|s,a)} \left[ \left( V_{h+1,P^\star,\boldsymbol{\Phi}}^{\pi^k} \right)^2 \right] - \left( \mathbb{E}_{\widetilde{T}_h^k(\cdot|s,a)} \left[ V_{h+1}^{\pi^k}(s') \right] \right)^2$$

$$+ 2 \cdot \sup_{(s,a) \in \mathcal{S} \times \mathcal{A}} D_{\mathrm{TV}} \left( P_h^\star(\cdot|s,a) \,\middle\|\, \widetilde{T}_h^k(\cdot|s,a) \right) \cdot \left( \max_{s' \in \mathcal{S}} V_{h+1,P^\star,\boldsymbol{\Phi}}^{\pi}(s') \right)^2$$

$$\leq \mathbb{V}_{\widetilde{T}_h^k(\cdot|s,a)} \left[ V_{h+1,P^\star,\boldsymbol{\Phi}}^{\pi^k} \right] + 2\rho \cdot \overline{H}^2, \quad \forall (s,a,h) \in \mathcal{S} \times \mathcal{A} \times [H], \quad \text{(E.29)}$$

where the last inequality follows from the definition of $\widetilde{T}^k$ and (E.22). Combining (E.28) and (E.29), we can upper bound Term (iii) by the following,

$$\text{Term (iii)} = \sum_{k=1}^K \mathbb{E}_{(s_h^k, a_h^k) \sim (P^\star, \pi^k)} \left[ \sum_{h=1}^H \mathbb{V}_{P_h^\star(\cdot|s_h^k, a_h^k)} \left[ V_{h+1,P^\star,\boldsymbol{\Phi}}^{\pi^k} \right] \right] \qquad \text{(E.30)}$$

$$- \mathbb{E}_{(s_h^k, a_h^k) \sim (\widetilde{T}^k, \pi^k)} \left[ \sum_{h=1}^H \mathbb{V}_{P_h^\star(\cdot|s_h^k, a_h^k)} \left[ V_{h+1,P^\star,\boldsymbol{\Phi}}^{\pi^k} \right] \right]$$

$$+ \sum_{k=1}^K \mathbb{E}_{(s_h^k, a_h^k) \sim (\widetilde{T}^k, \pi^k)} \left[ \sum_{h=1}^H \mathbb{V}_{P_h^\star(\cdot|s_h^k, a_h^k)} \left[ V_{h+1,P^\star,\boldsymbol{\Phi}}^{\pi^k} \right] \right]$$

$$- \mathbb{E}_{(s_h^k, a_h^k) \sim (\widetilde{T}^k, \pi^k)} \left[ \sum_{h=1}^H \mathbb{V}_{\widetilde{T}_h^k(\cdot|s_h^k, a_h^k)} \left[ V_{h+1,P^\star,\boldsymbol{\Phi}}^{\pi^k} \right] \right]$$

$$\leq 3\rho H \cdot \overline{H}^2 \cdot K \leq 3H \cdot \overline{H} \cdot K,$$

where in the last inequality we use the fact that for any $\rho \in [0,1]$, it holds that

$$\rho \overline{H} = \rho \cdot \min \left\{ H, \rho^{-1} \right\} = \min \left\{ \rho H, 1 \right\} \leq 1.$$

**Finishing the proof.** Finally, combining the upper bounds for Terms (i), (ii), and (iii), i.e., (E.23), (E.25), and (E.30), we conclude that with probability at least $1 - \delta$, it holds that

$$\sum_{k=1}^K \sum_{h=1}^H \mathbb{V}_{P_h^\star(\cdot|s_h^k, a_h^k)} \left[ V_{h+1,P^\star,\boldsymbol{\Phi}}^{\pi^k} \right] \leq c \cdot H \cdot \overline{H}^2 \cdot \sqrt{K\iota} + 2H \cdot \overline{H} \cdot K + 3H \cdot \overline{H} \cdot K$$

$$\leq c' \cdot H \cdot \overline{H} \cdot K + c'' \cdot H \cdot \overline{H}^3 \cdot \iota,$$

where in the last inequality we use $\sqrt{ab} \leq a + b$ for any $a, b > 0$. Plug in the notation that $\overline{H} = \min\{H, \rho^{-1}\}$ and finish the proof of Lemma E.5. $\qquad \square$

### E.7 Proof of Lemma E.6

*Proof of Lemma E.6.* Using the property of variance, we have that for any $(s_h, a_h) \in \mathcal{S} \times \mathcal{A}$,

$$\mathbb{V}_{\widetilde{T}_h(\cdot|s_h, a_h)} \left[ V_{h+1,P^\star,\boldsymbol{\Phi}}^{\pi} \right]$$

$$= \mathbb{E}_{\widetilde{T}_h(\cdot|s_h, a_h)} \left[ \left( V_{h+1,P^\star,\boldsymbol{\Phi}}^{\pi} \right)^2 \right] - \left( \mathbb{E}_{\widetilde{T}_h(\cdot|s_h, a_h)} \left[ V_{h+1,P^\star,\boldsymbol{\Phi}}^{\pi} \right] \right)^2. \qquad \text{(E.31)}$$

By robust Bellman equation (Proposition 2.2) and the definition of $\widetilde{T}_h$ in (E.24), we have that

$$V_{h,P^\star,\Phi}^\pi(s_h) = R_h(s_h, \pi_h(s_h)) + \mathbb{E}_{\widetilde{T}_h(\cdot|s_h,\pi_h(s_h))}\left[V_{h+1,P^\star,\Phi}^\pi\right]. \tag{E.32}$$

Therefore, by (E.31) and (E.32), we have that

$$\mathbb{V}_{\widetilde{T}_h(\cdot|s_h,\pi_h(s_h))}\left[V_{h+1,P^\star,\Phi}^\pi\right]$$
$$= \mathbb{E}_{\widetilde{T}_h(\cdot|s_h,\pi_h(s_h))}\left[\left(V_{h+1,P^\star,\Phi}^\pi\right)^2\right] - \left(V_{h,P^\star,\Phi}^\pi(s_h) - R_h(s_h,\pi_h(s_h))\right)^2. \tag{E.33}$$

For the second term in (E.33), we can calculate it as

$$-\left(V_{h,P^\star,\Phi}^\pi(s_h) - R_h(s_h,\pi_h(s_h))\right)^2$$
$$= -\left(V_{h,P^\star,\Phi}^\pi\right)^2(s_h) + 2 \cdot V_{h,P^\star,\Phi}^\pi(s_h) \cdot R_h(s_h,\pi_h(s_h)) - R_h^2(s_h,\pi_h(s_h))$$
$$\leq -\left(V_{h,P^\star,\Phi}^\pi\right)^2(s_h) + 2\overline{H}, \tag{E.34}$$

where the last inequality utilizes the facts that $0 \leq R_h(s_h,\pi_h(s_h)) \leq 1$, $R_h^2(s_h,\pi_h(s_h)) \geq 0$, and (E.22) that $V_{h,P^\star,\Phi}^\pi(s_h) \leq \overline{H}$. Combining (E.33) and (E.34), we have that

$$\mathbb{V}_{\widetilde{T}_h(\cdot|s_h,\pi_h(s_h))}\left[V_{h+1,P^\star,\Phi}^\pi\right] \leq \mathbb{E}_{\widetilde{T}_h(\cdot|s_h,\pi_h(s_h))}\left[\left(V_{h+1,P^\star,\Phi}^\pi\right)^2\right] - \left(V_{h,P^\star,\Phi}^\pi\right)^2(s_h) + 2\overline{H},$$

which further implies that

$$\mathbb{E}_{(s_h,a_h)\sim(\widetilde{T},\pi)}\left[\mathbb{V}_{\widetilde{T}_h(\cdot|s_h,a_h)}\left[V_{h+1,P^\star,\Phi}^\pi\right]\right]$$

$$= \mathbb{E}_{s_h\sim(\widetilde{T},\pi)}\left[\mathbb{V}_{\widetilde{T}_h(\cdot|s_h,\pi_h(s_h))}\left[V_{h+1,P^\star,\Phi}^\pi\right]\right]$$

$$\leq \mathbb{E}_{s_h\sim(\widetilde{T},\pi)}\left[\mathbb{E}_{\widetilde{T}_h(\cdot|s_h,\pi_h(s_h))}\left[\left(V_{h+1,P^\star,\Phi}^\pi\right)^2\right] - \left(V_{h,P^\star,\Phi}^\pi\right)^2 + 2\overline{H}\right]$$

$$= \mathbb{E}_{s_{h+1}\sim(\widetilde{T},\pi)}\left[\left(V_{h+1,P^\star,\Phi}^\pi\right)^2\right] - \mathbb{E}_{s_h\sim(\widetilde{T},\pi)}\left[\left(V_{h,P^\star,\Phi}^\pi\right)^2\right] + 2\overline{H}.$$

Taking summation over $h \in [H]$ gives that

$$\mathbb{E}_{(s_h,a_h)\sim(\widetilde{T},\pi),h\in[H]}\left[\sum_{h=1}^H \mathbb{V}_{\widetilde{T}_h(\cdot|s_h,a_h)}\left[V_{h+1,P^\star,\Phi}^\pi\right]\right] \leq 2H \cdot \overline{H} = 2H \cdot \min\left\{H, \rho^{-1}\right\},$$

which concludes the proof of Lemma E.6. $\qquad\square$

## E.8 Other Technical Lemmas

Before presenting all lemmas, we recall that the typical event $\mathcal{E}$ is defined as

$$\mathcal{E} = \left\{ \left|\left(\mathbb{E}_{P_h^\star(\cdot|s,a)} - \mathbb{E}_{\widehat{P}_h^k(\cdot|s,a)}\right)\left[\left(\eta - V_{h+1,P^\star,\Phi}^\star\right)_+\right]\right| \right.$$

$$\leq \sqrt{\frac{\mathbb{V}_{\widehat{P}_h^k(\cdot|s,a)}\left[\left(\eta - V_{h+1,P^\star,\Phi}^\star\right)_+\right] \cdot c_1\iota}{N_h^k(s,a) \vee 1}} + \frac{c_2 H\iota}{N_h^k(s,a) \vee 1},$$

$$\left|P_h^\star(s'|s,a) - \widehat{P}_h(s'|s,a)\right| \leq \sqrt{\frac{\min\left\{P_h^\star(s'|s,a), \widehat{P}_h^k(s'|s,a)\right\} \cdot c_1\iota}{N_h^k(s,a) \vee 1}} + \frac{c_2\iota}{N_h^k(s,a) \vee 1},$$

$$\left. \forall (s,a,s',h,k) \in \mathcal{S} \times \mathcal{A} \times \mathcal{S} \times [H] \times [K], \ \forall \eta \in \mathcal{N}_{1/(S\sqrt{K})}([0,H]) \right\}, \tag{E.35}$$

where $\iota = \log(S^3AH^2K^{3/2}/\delta)$, $c_1, c_2 > 0$ are two absolute constants, $\mathcal{N}_{1/S\sqrt{K}}([0,H])$ denotes an $1/S\sqrt{K}$-cover of the interval $[0,H]$. where $c_1, c_2 > 0$ are two absolute constants, $\mathcal{N}_{1/S\sqrt{K}}([0,H])$ denotes an $1/S\sqrt{K}$-cover of the interval $[0,H]$.

### E.8.1 Concentration Inequalities

**Lemma E.7** (Bernstein bound for TV robust sets and the optimal robust value function). *Under event $\mathcal{E}$ in (E.35), it holds that*

$$\left| \mathbb{E}_{\mathcal{P}_\rho(s,a;\widehat{P}_h^k)}\left[ V_{h+1,P^\star,\mathbf{\Phi}}^\star \right] - \mathbb{E}_{\mathcal{P}_\rho(s,a;P_h^\star)}\left[ V_{h+1,P^\star,\mathbf{\Phi}}^\star \right] \right|$$

$$\leq \sqrt{\frac{\mathbb{V}_{\widehat{P}_h^k(\cdot|s,a)}\left[ V_{h+1,P^\star,\mathbf{\Phi}}^\star \right] \cdot c_1 \iota}{N_h^k(s,a) \vee 1}} + \frac{c_2 H \iota}{N_h^k(s,a) \vee 1} + \frac{1}{\sqrt{K}},$$

*where $\iota = \log(S^3 A H^2 K^{3/2}/\delta)$.*

*Proof of Lemma E.7.* By our definition of the operator $\mathbb{E}_{\mathcal{P}_\rho(s,a;\widehat{P}_h^k)}[V_{h+1,P^\star,\mathbf{\Phi}}^\star]$ in (4.1), we can arrive that

$$\left| \mathbb{E}_{\mathcal{P}_\rho(s,a;\widehat{P}_h^k)}\left[ V_{h+1,P^\star,\mathbf{\Phi}}^\star \right] - \mathbb{E}_{\mathcal{P}_\rho(s,a;P_h^\star)}\left[ V_{h+1,P^\star,\mathbf{\Phi}}^\star \right] \right|$$

$$= \left| \sup_{\eta \in [0,H]} \left\{ -\mathbb{E}_{\widehat{P}_h^k(\cdot|s,a)}\left[ \left( \eta - V_{h+1,P^\star,\mathbf{\Phi}}^\star \right)_+ \right] + \left( 1 - \frac{\rho}{2} \right) \cdot \eta \right\} \right.$$

$$\left. - \sup_{\eta \in [0,H]} \left\{ -\mathbb{E}_{P_h^\star(\cdot|s,a)}\left[ \left( \eta - V_{h+1,P^\star,\mathbf{\Phi}}^\star \right)_+ \right] + \left( 1 - \frac{\rho}{2} \right) \cdot \eta \right\} \right|$$

$$\leq \sup_{\eta \in [0,H]} \left\{ \left| \left( \mathbb{E}_{\widehat{P}_h^k(\cdot|s,a)} - \mathbb{E}_{P_h^\star(\cdot|s,a)} \right) \left[ \left( \eta - V_{h+1,P^\star,\mathbf{\Phi}}^\star \right)_+ \right] \right| \right\}, \qquad (\text{E.36})$$

Now according to the first inequality of event $\mathcal{E}$, we have that

$$\left| \left( \mathbb{E}_{P_h^\star(\cdot|s,a)} - \mathbb{E}_{\widehat{P}_h^k(\cdot|s,a)} \right) \left[ \left( \eta - V_{h+1,P^\star,\mathbf{\Phi}}^\star \right)_+ \right] \right|$$

$$\leq \sqrt{\frac{\mathbb{V}_{\widehat{P}_h^k(\cdot|s,a)}\left[ \left( \eta - V_{h+1,P^\star,\mathbf{\Phi}}^\star \right)_+ \right] \cdot c_1 \iota}{N_h^k(s,a) \vee 1}} + \frac{c_2 H \iota}{N_h^k(s,a) \vee 1}$$

$$\leq \sqrt{\frac{\mathbb{V}_{\widehat{P}_h^k(\cdot|s,a)}\left[ V_{h+1,P^\star,\mathbf{\Phi}}^\star \right] \cdot c_1 \iota}{N_h^k(s,a) \vee 1}} + \frac{c_2 H \iota}{N_h^k(s,a) \vee 1},$$

for any $\eta \in \mathcal{N}_{1/(S\sqrt{K})}([0,H])$. Here the second inequality is because $\text{Var}[(a - X)_+] \leq \text{Var}[X]$. Therefore, by a covering argument, for any $\eta \in [0,H]$, it holds that

$$\left| \left( \mathbb{E}_{P_h^\star(\cdot|s,a)} - \mathbb{E}_{\widehat{P}_h^k(\cdot|s,a)} \right) \left[ \left( \eta - V_{h+1,P^\star,\mathbf{\Phi}}^\star \right)_+ \right] \right|$$

$$\leq \sqrt{\frac{\mathbb{V}_{\widehat{P}_h^k(\cdot|s,a)}\left[ V_{h+1,P^\star,\mathbf{\Phi}}^\star \right] \cdot c_1 \iota}{N_h^k(s,a) \vee 1}} + \frac{c_2 H \iota}{N_h^k(s,a) \vee 1} + \frac{1}{\sqrt{K}}.$$

This finishes the proof of Lemma E.7. $\qquad\qquad\square$

**Lemma E.8** (Bernstein bound for TV robust sets and the robust value function of $\pi^k$). *Under event $\mathcal{E}$ in (E.35), suppose that the optimism and pessimism (E.14) holds at $(h+1,k)$, then it holds that*

$$\left| \mathbb{E}_{\mathcal{P}_\rho(s,a;\widehat{P}_h^k)}\left[ V_{h+1,P^\star,\mathbf{\Phi}}^{\pi^k} \right] - \mathbb{E}_{\mathcal{P}_\rho(s,a;P_h^\star)}\left[ V_{h+1,P^\star,\mathbf{\Phi}}^{\pi^k} \right] \right|$$

$$\leq \sqrt{\frac{\mathbb{V}_{\widehat{P}_h^k(\cdot|s,a)}\left[ V_{h+1,P^\star,\mathbf{\Phi}}^\star \right] \cdot c_1 \iota}{N_h^k(s,a) \vee 1}} + \frac{\mathbb{E}_{\widehat{P}_h^k(\cdot|s,a)}\left[ \overline{V}_{h+1}^k - \underline{V}_{h+1}^k \right]}{H} + \frac{c_2' H^2 S \iota}{N_h^k(s,a) \vee 1} + \frac{1}{\sqrt{K}},$$

*where $\iota = \log(S^3 A H^2 K^{3/2}/\delta)$ and $c_1, c_2'$ are absolute constants.*

*Proof of Lemma E.8.* By our definition of the operator $\mathbb{E}_{\mathcal{P}_\rho(s,a;P)}[V^{\pi^k}_{h+1,P^\star,\Phi}]$ in (4.1), we can arrive that,

$$
\left| \mathbb{E}_{\mathcal{P}_\rho(s,a;\widehat{P}^k_h)}\left[V^{\pi^k}_{h+1,P^\star,\Phi}\right] - \mathbb{E}_{\mathcal{P}_\rho(s,a;P^\star_h)}\left[V^{\pi^k}_{h+1,P^\star,\Phi}\right] \right|
$$

$$
= \left| \sup_{\eta\in[0,H]} \left\{ -\mathbb{E}_{\widehat{P}^k_h(\cdot|s,a)}\left[\left(\eta - V^{\pi^k}_{h+1,P^\star,\Phi}\right)_+\right] + \left(1 - \frac{\rho}{2}\right)\cdot\eta \right\} \right.
$$

$$
\left. - \sup_{\eta\in[0,H]} \left\{ -\mathbb{E}_{P^\star_h(\cdot|s,a)}\left[\left(\eta - V^{\pi^k}_{h+1,P^\star,\Phi}\right)_+\right] + \left(1 - \frac{\rho}{2}\right)\cdot\eta \right\} \right|
$$

$$
\leq \sup_{\eta\in[0,H]} \left\{ \left| \left(\mathbb{E}_{\widehat{P}^k_h(\cdot|s,a)} - \mathbb{E}_{P^\star_h(\cdot|s,a)}\right)\left[\left(\eta - V^{\pi^k}_{h+1,P^\star,\Phi}\right)_+\right] \right| \right\}
$$

$$
\leq \underbrace{\sup_{\eta\in[0,H]} \left\{ \left| \left(\mathbb{E}_{\widehat{P}^k_h(\cdot|s,a)} - \mathbb{E}_{P^\star_h(\cdot|s,a)}\right)\left[\left(\eta - V^\star_{h+1,P^\star,\Phi}\right)_+\right] \right| \right\}}_{\text{Term (i)}}
$$

$$
+ \underbrace{\sup_{\eta\in[0,H]} \left\{ \left| \left(\mathbb{E}_{\widehat{P}^k_h(\cdot|s,a)} - \mathbb{E}_{P^\star_h(\cdot|s,a)}\right)\left[\left(\eta - V^{\pi^k}_{h+1,P^\star,\Phi}\right)_+ - \left(\eta - V^\star_{h+1,P^\star,\Phi}\right)_+\right] \right| \right\}}_{\text{Term (ii)}}.
$$

We deal with Term (i) and Term (ii) respectively. For Term (i), this is exactly the same as the right hand side of (E.36). Therefore, applying the same argument as Lemma E.7 gives the following upper bound,

$$
\text{Term (i)} \leq \sqrt{\frac{\mathbb{V}_{\widehat{P}^k_h(\cdot|s,a)}\left[V^\star_{h+1,P^\star,\Phi}\right]\cdot c_1\iota}{N^k_h(s,a)\vee 1}} + \frac{c_2 H\iota}{N^k_h(s,a)\vee 1} + \frac{1}{\sqrt{K}}. \tag{E.37}
$$

For Term (ii), we first apply the second inequality of event $\mathcal{E}$ to obtain that,

$$
\text{Term (ii)} \tag{E.38}
$$

$$
\leq \sup_{\eta\in[0,H]} \left\{ \sum_{s'\in\mathcal{S}} \left( \sqrt{\frac{\widehat{P}^k_h(s'|s,a)\cdot c_1\iota}{N^k_h(s,a)\vee 1}} + \frac{c_2\iota}{N^k_h(s,a)\vee 1} \right) \right.
$$

$$
\left. \cdot \left| \left(\eta - V^{\pi^k}_{h+1,P^\star,\Phi}(s')\right)_+ - \left(\eta - V^\star_{h+1,P^\star,\Phi}(s')\right)_+ \right| \right\}.
$$

By the assumption that (E.14) holds at $(h+1,k)$, we can upper bound the absolute value above by

$$
\left| \left(\eta - V^{\pi^k}_{h+1,P^\star,\Phi}(s')\right)_+ - \left(\eta - V^\star_{h+1,P^\star,\Phi}(s')\right)_+ \right| \leq \left| V^{\pi^k}_{h+1,P^\star,\Phi}(s') - V^\star_{h+1,P^\star,\Phi}(s') \right|
$$

$$
\leq \overline{V}^k_{h+1}(s') - \underline{V}^k_{h+1}(s'). \tag{E.39}
$$

where the first inequality is due to the 1-Lipschitz continuity of $\psi_\eta(x) = (\eta - x)_+$, and the second inequality is due to (E.14). Thus combining (E.38) and (E.39), we know that

$$
\text{Term (ii)} \leq \sum_{s'\in\mathcal{S}} \left( \sqrt{\frac{\widehat{P}^k_h(s'|s,a)\cdot c_1\iota}{N^k_h(s,a)\vee 1}} + \frac{c_2\iota}{N^k_h(s,a)\vee 1} \right) \cdot \left( \overline{V}^k_{h+1}(s') - \underline{V}^k_{h+1}(s') \right). \tag{E.40}
$$

Now following the argument first identified by Azar et al. (2017), we proceed to upper bound (E.40) as

$$
\text{Term (ii)} \leq \sum_{s'\in\mathcal{S}} \left( \frac{\widehat{P}^k_h(s'|s,a)}{H} + \frac{c_1 H\iota}{N^k_h(s,a)\vee 1} + \frac{c_2\iota}{N^k_h(s,a)\vee 1} \right) \cdot \left( \overline{V}^k_{h+1}(s') - \underline{V}^k_{h+1}(s') \right)
$$

$$\leq \frac{\mathbb{E}_{\widehat{P}_h^k(\cdot|s,a)}\left[\overline{V}_{h+1}^k - \underline{V}_{h+1}^k\right]}{H} + \frac{c_2' H^2 S \iota}{N_h^k(s,a) \vee 1}, \tag{E.41}$$

where $c_2' > 0$ is another absolute constant. The first inequality is by $\sqrt{ab} \leq a + b$ and the second inequality is due to $\overline{V}_{h+1}^k, \underline{V}_{h+1}^k \in [0, H]$. Finally, combining (E.37) and (E.41), we prove Lemma E.8. $\qquad\square$

**Lemma E.9** (Bernstein bounds for TV robust sets and optimistic and pessimistic robust value estimators). *Under event $\mathcal{E}$ in (E.35), suppose that the optimism and pessimism (E.14) holds at $(h+1, k)$, it holds that*

$$\max\left\{\left|\mathbb{E}_{\mathcal{P}_\rho(s,a;\widehat{P}_h^k)}\left[\overline{V}_{h+1}^k\right] - \mathbb{E}_{\mathcal{P}_\rho(s,a;P_h^\star)}\left[\overline{V}_{h+1}^k\right]\right|, \left|\mathbb{E}_{\mathcal{P}_\rho(s,a;\widehat{P}_h^k)}\left[\underline{V}_{h+1}^k\right] - \mathbb{E}_{\mathcal{P}_\rho(s,a;P_h^\star)}\left[\underline{V}_{h+1}^k\right]\right|\right\}$$

$$\leq \sqrt{\frac{\mathbb{V}_{\widehat{P}_h^k(\cdot|s,a)}\left[V_{h+1,P^\star,\Phi}^\star\right] \cdot c_1 \iota}{N_h^k(s,a) \vee 1}} + \frac{\mathbb{E}_{\widehat{P}_h^k(\cdot|s,a)}\left[\overline{V}_{h+1}^k - \underline{V}_{h+1}^k\right]}{H} + \frac{c_2' H^2 S \iota}{N_h^k(s,a) \vee 1} + \frac{1}{\sqrt{K}},$$

*where $\iota = \log(S^3 A H^2 K^{3/2}/\delta)$ and $c_1, c_2'$ are absolute constants.*

*Proof of Lemma E.9.* This follows from the same proof as Lemma E.8 and is thus omitted. $\qquad\square$

**Lemma E.10** (Non-robust concentration). *Under event $\mathcal{E}$ in (E.35), suppose that the optimism and pessimism (E.14) holds at $(h+1, k)$, then it holds that*

$$\left|\left(\mathbb{E}_{\widehat{P}_h^k(\cdot|s,a)} - \mathbb{E}_{P_h^\star(\cdot|s,a)}\right)\left[\overline{V}_{h+1}^k - \underline{V}_{h+1}^k\right]\right| \leq \frac{1}{H} \cdot \mathbb{E}_{P_h^\star(\cdot|s,a)}\left[\overline{V}_{h+1}^k - \underline{V}_{h+1}^k\right] + \frac{c_2' H^2 S \iota}{N_h^k(s,a) \vee 1}.$$

*where $\iota = \log(S^2 A H^2 K^{3/2}/\delta)$ and $c_2'$ is an absolute constant.*

*Proof of Lemma E.10.* According to the second inequality of event $\mathcal{E}$, we have that

$$\left|\left(\mathbb{E}_{\widehat{P}_h^k(\cdot|s,a)} - \mathbb{E}_{P_h^\star(\cdot|s,a)}\right)\left[\overline{V}_{h+1}^k - \underline{V}_{h+1}^k\right]\right|$$

$$\leq \sum_{s' \in \mathcal{S}}\left(\sqrt{\frac{P_h^\star(s'|s,a) \cdot c_1 \iota}{N_h^k(s,a) \vee 1}} + \frac{c_2 \iota}{N_h^k(s,a) \vee 1}\right) \cdot \left(\overline{V}_{h+1}^k(s') - \underline{V}_{h+1}^k(s')\right),$$

where we also apply (E.14) that $\overline{V}_{h+1}^k(s') \geq \underline{V}_{h+1}^k(s')$. Now using the same argument as (E.41) in the proof of Lemma E.8, we can arrive at

$$\left|\left(\mathbb{E}_{\widehat{P}_h^k(\cdot|s,a)} - \mathbb{E}_{P_h^\star(\cdot|s,a)}\right)\left[\overline{V}_{h+1}^k - \underline{V}_{h+1}^k\right]\right|$$

$$\leq \frac{\mathbb{E}_{P_h^\star(\cdot|s,a)}\left[\overline{V}_{h+1}^k(s') - \underline{V}_{h+1}^k(s')\right]}{H} + \frac{c_2' H^2 S \iota}{N_h^k(s,a) \vee 1},$$

which finishes the proof of Lemma E.10. $\qquad\square$

### E.8.2 Variance Analysis

**Lemma E.11** (Variance analysis 1). *Suppose that the optimism and pessimism (E.14) holds at $(h+1, k)$, then the following inequality holds,*

$$\left|\mathbb{V}_{\widehat{P}_h^k(\cdot|s,a)}\left[\left(\overline{V}_{h+1}^k + \underline{V}_{h+1}^k\right)/2\right] - \mathbb{V}_{\widehat{P}_h^k(\cdot|s,a)}\left[V_{h+1,P^\star,\Phi}^\star\right]\right| \leq 4H \cdot \mathbb{E}_{\widehat{P}_h^k(\cdot|s,a)}\left[\overline{V}_{h+1}^k - \underline{V}_{h+1}^k\right].$$

*Proof of Lemma E.11.* Directly consider that the left hand side can be upper bounded by the following,

$$\left| \mathbb{V}_{\widehat{P}_h^k(\cdot|s,a)}\left[ \left( \overline{V}_{h+1}^k + \underline{V}_{h+1}^k \right)/2 \right] - \mathbb{V}_{\widehat{P}_h^k(\cdot|s,a)}\left[ V_{h+1,P^\star,\Phi}^\star \right] \right|$$

$$\leq \left| \mathbb{E}_{\widehat{P}_h^k(\cdot|s,a)}\left[ \left( \overline{V}_{h+1}^k + \underline{V}_{h+1}^k \right)^2/4 \right] - \mathbb{E}_{\widehat{P}_h^k(\cdot|s,a)}\left[ \left( V_{h+1,P^\star,\Phi}^\star \right)^2 \right] \right|$$

$$+ \left| \left( \mathbb{E}_{\widehat{P}_h^k(\cdot|s,a)}\left[ \left( \overline{V}_{h+1}^k + \underline{V}_{h+1}^k \right)/2 \right] \right)^2 - \left( \mathbb{E}_{\widehat{P}_h^k(\cdot|s,a)}\left[ V_{h+1,P^\star,\Phi}^\star \right] \right)^2 \right|. \quad \text{(E.42)}$$

Since all of $\overline{V}_{h+1}^k, \underline{V}_{h+1}^k, V_{h+1,P^\star,\Phi}^\star \in [0, H]$ (by the correctness of (E.14) and the definitions of $\overline{V}_{h+1}^k, \underline{V}_{h+1}^k$), we can further upper bound the right hand side of (E.42) as

$$\left| \mathbb{V}_{\widehat{P}_h^k(\cdot|s,a)}\left[ \left( \overline{V}_{h+1}^k + \underline{V}_{h+1}^k \right)/2 \right] - \mathbb{V}_{\widehat{P}_h^k(\cdot|s,a)}\left[ V_{h+1,P^\star,\Phi}^\star \right] \right|$$

$$\leq 4H \cdot \mathbb{E}_{\widehat{P}_h^k(\cdot|s,a)}\left[ \left| \left( \overline{V}_{h+1}^k + \underline{V}_{h+1}^k \right)/2 - V_{h+1,P^\star,\Phi}^\star \right| \right]$$

$$\leq 4H \cdot \mathbb{E}_{\widehat{P}_h^k(\cdot|s,a)}\left[ \overline{V}_{h+1}^k - \underline{V}_{h+1}^k \right],$$

where the last inequality is due to the correctness of (E.14) at $(h+1, k)$. This proves Lemma E.11. $\square$

**Lemma E.12** (Variance analysis 2). *Under event $\mathcal{E}$ in (E.35), suppose that optimism and pessimism (E.14) holds at $(h+1, k)$, then it holds that*

$$\left| \mathbb{V}_{\widehat{P}_h^k(\cdot|s,a)}\left[ \left( \overline{V}_{h+1}^k + \underline{V}_{h+1}^k \right)/2 \right] - \mathbb{V}_{P_h^\star(\cdot|s,a)}\left[ V_{h+1,P^\star,\Phi}^{\pi^k} \right] \right|$$

$$\leq 4H \cdot \mathbb{E}_{P_h^\star(\cdot|s,a)}\left[ \overline{V}_{h+1}^k - \underline{V}_{h+1}^k \right] + \frac{c_2' H^4 S \iota}{N_h^k(s,a)} + 1.$$

*Proof of Lemma E.12.* We first relate the variance on $\widehat{P}_h^k$ to the variance on $P_h^\star$. Specifically, we have

$$\left| \mathbb{V}_{\widehat{P}_h^k(\cdot|s,a)}\left[ \left( \overline{V}_{h+1}^k + \underline{V}_{h+1}^k \right)/2 \right] - \mathbb{V}_{P_h^\star(\cdot|s,a)}\left[ \left( \overline{V}_{h+1}^k + \underline{V}_{h+1}^k \right)/2 \right] \right|$$

$$= \left| \mathbb{E}_{\widehat{P}_h^k(\cdot|s,a)}\left[ \left( \left( \overline{V}_{h+1}^k + \underline{V}_{h+1}^k \right)/2 - \mathbb{E}_{\widehat{P}_h^k(\cdot|s,a)}\left[ \left( \overline{V}_{h+1}^k + \underline{V}_{h+1}^k \right)/2 \right] \right)^2 \right] \right|$$

$$+ \left| \mathbb{E}_{P_h^\star(\cdot|s,a)}\left[ \left( \left( \overline{V}_{h+1}^k + \underline{V}_{h+1}^k \right)/2 - \mathbb{E}_{P_h^\star(\cdot|s,a)}\left[ \left( \overline{V}_{h+1}^k + \underline{V}_{h+1}^k \right)/2 \right] \right)^2 \right] \right| \text{(E.43)}$$

Since $(\overline{V}_{h+1}^k + \underline{V}_{h+1}^k)/2 \in [0, H]$, we can further upper bound (E.43) by

$$\left| \mathbb{V}_{\widehat{P}_h^k(\cdot|s,a)}\left[ \left( \overline{V}_{h+1}^k + \underline{V}_{h+1}^k \right)/2 \right] - \mathbb{V}_{P_h^\star(\cdot|s,a)}\left[ \left( \overline{V}_{h+1}^k + \underline{V}_{h+1}^k \right)/2 \right] \right|$$

$$\leq H^2 \cdot \sum_{s' \in \mathcal{S}} \left| P_h^\star(s'|s,a) - \widehat{P}_h(s'|s,a) \right|$$

$$\leq H^2 \cdot \sum_{s' \in \mathcal{S}} \left( \sqrt{\frac{P_h^\star(\cdot|s,a) \cdot c_1 \iota}{N_h^k(s,a) \vee 1}} + \frac{c_2 \iota}{N_h^k(s,a) \vee 1} \right)$$

$$\leq H^2 \cdot \left( \sqrt{\frac{c_1 S \iota}{N_h^k(s,a) \vee 1}} + \frac{c_2 S \iota}{N_h^k(s,a) \vee 1} \right)$$

$$\leq 1 + \frac{c_2' H^4 S \iota}{N_h^k(s,a) \vee 1}, \quad \text{(E.44)}$$

where the second inequality is by the second inequality in event $\mathcal{E}$, the third inequality is by Cauchy-Schwartz inequality and the probability distribution sums up to 1, and the last inequality is from $\sqrt{ab} \le a + b$. Thus by (E.44), we can bound our target as

$$\left| \mathbb{V}_{\widehat{P}_h^k(\cdot|s,a)}\left[\left(\overline{V}_{h+1}^k + \underline{V}_{h+1}^k\right)/2\right] - \mathbb{V}_{P_h^\star(\cdot|s,a)}\left[V_{h+1,P^\star,\mathbf{\Phi}}^{\pi^k}\right]\right|$$

$$\le \left| \mathbb{V}_{P_h^\star(\cdot|s,a)}\left[\left(\overline{V}_{h+1}^k + \underline{V}_{h+1}^k\right)/2\right] - \mathbb{V}_{P_h^\star(\cdot|s,a)}\left[V_{h+1,P^\star,\mathbf{\Phi}}^{\pi^k}\right]\right|$$

$$+ \frac{c_2' H^4 S \iota}{N_h^k(s,a) \vee 1} + 1. \tag{E.45}$$

Now by the same proof of Lemma E.11, using the correctness of (E.14) at $(h+1, k)$, we can show that

$$\left| \mathbb{V}_{P_h^\star(\cdot|s,a)}\left[\left(\overline{V}_{h+1}^k + \underline{V}_{h+1}^k\right)/2\right] - \mathbb{V}_{P_h^\star(\cdot|s,a)}\left[V_{h+1,P^\star,\mathbf{\Phi}}^{\pi^k}\right]\right|$$

$$\le 4H \cdot \mathbb{E}_{P_h^\star(\cdot|s,a)}\left[\overline{V}_{h+1}^k - \underline{V}_{h+1}^k\right]. \tag{E.46}$$

Combining (E.45) and (E.46), we can finish the proof of Lemma E.12. $\qquad\square$

### E.8.3 Other Auxiliary Lemmas

**Lemma E.13** (Lemma 7.5 in Agarwal et al. (2019)). *For the sequences of $\{s_h^k, a_h^k\}_{h,k=1}^{H,K}$, it holds that*

$$\sum_{k=1}^{K}\sum_{h=1}^{H} \frac{1}{N_h^k(s_h^k, a_h^k) \vee 1} \le c \cdot HSA \log(K).$$

*where $c > 0$ is an absolute constant.*

*Proof of Lemma E.13.* See Lemma 7.5 in Agarwal et al. (2019) for a detailed proof. $\qquad\square$

## F Proofs for Extensions in Section B.4.2

In this section, we prove the theoretical results in Section B.4.2.

### F.1 Proof of Corollary B.5

*Proof of Corollary B.5.* We consider applying Algorithm 1 on the auxiliary $\mathcal{S} \times \mathcal{A}$-rectangular RMDP with a TV robust set $\widetilde{\mathcal{M}}$ (see Section B.4.2) which satisfies the vanishing minimal value assumption (Assumption 4.1). Suppose the algorithm outputs $\widetilde{\pi}^1, \cdots, \widetilde{\pi}^K$ for the $K$ episodes. Then Theorem 4.3 shows that by a proper choice of the hyperparameters, with probability at least $1 - \delta$

$$\text{Regret}_{\widetilde{\mathbf{\Phi}}}(K) = \sum_{k=1}^{K} \max_{\widetilde{\pi}} V_{1,\widetilde{P}^\star,\widetilde{\mathbf{\Phi}}}^{\widetilde{\pi}}(s_1) - V_{1,\widetilde{P}^\star,\widetilde{\mathbf{\Phi}}}^{\widetilde{\pi}^k}(s_1)$$

$$\le \mathcal{O}\left(\sqrt{\min\{H, \rho^{-1}\} H^2 (S+1) A K \iota'}\right),. \tag{F.1}$$

where $\iota' = \log^2(SAHK/\delta)$ and $\rho = 2 - 2\rho' \in [0, 1)$. In the sequel, we prove that for any policy $\widetilde{\pi}$ of $\widetilde{\mathcal{M}}$ and its induced policy $\widetilde{\pi}_{\mathcal{S}}$ of $\mathcal{M}_\gamma$, their robust value functions coincide at the initial state $s_1 \in \mathcal{S}$, that is,

$$V_{1,\widetilde{P}^\star,\widetilde{\mathbf{\Phi}}}^{\widetilde{\pi}}(s_1) = V_{1,P^\star,\mathbf{\Phi}'}^{\widetilde{\pi}_{\mathcal{S}}}(s_1),$$

where $V_{1,\widetilde{P}^\star,\widetilde{\mathbf{\Phi}}}^{\widetilde{\pi}}$ is the robust value function of $\widetilde{\pi}$ in $\widetilde{\mathcal{M}} = (\widetilde{\mathcal{S}}, \mathcal{A}, H, \widetilde{P^\star}, \widetilde{R}, \widetilde{\mathbf{\Phi}})$, and $V_{1,P^\star,\mathbf{\Phi}'}^{\widetilde{\pi}_{\mathcal{S}}}$ is the robust value function of $\widetilde{\pi}_{\mathcal{S}}$ in $\mathcal{M}_\gamma = (\mathcal{S}, \mathcal{A}, H, P^\star, R_\gamma, \mathbf{\Phi}')$. To this end, we actually prove a stronger result that for any step $h \in [H]$, it holds that

$$(\rho')^{h-1} \cdot V_{h,\widetilde{P}^\star,\widetilde{\mathbf{\Phi}}}^{\widetilde{\pi}}(s) = V_{h,P^\star,\mathbf{\Phi}'}^{\widetilde{\pi}_{\mathcal{S}}}(s), \quad \forall s \in \mathcal{S}. \tag{F.2}$$

We prove (F.2) by induction. For step $H$, by robust Bellman equation, we have that, for any $(s, a) \in \mathcal{S} \times \mathcal{A}$,

$$(\rho')^{H-1} \cdot Q^{\widetilde{\pi}}_{H, \widetilde{P}^\star, \widetilde{\boldsymbol{\Phi}}}(s, a) = (\rho')^{H-1} \cdot \left(\frac{\gamma}{\rho'}\right)^{H-1} \cdot R_H(s, a) = R_{\gamma, H}(s, a) = Q^{\widetilde{\pi}_\mathcal{S}}_{H, P^\star, \boldsymbol{\Phi}'}(s, a),$$

and thus for any $s \in \mathcal{S}$,

$$\begin{aligned} (\rho')^{H-1} \cdot V^{\widetilde{\pi}}_{h, \widetilde{P}^\star, \widetilde{\boldsymbol{\Phi}}}(s) &= \mathbb{E}_{\widetilde{\pi}(\cdot|s)}\left[(\rho')^{H-1} \cdot Q^{\widetilde{\pi}}_{H, \widetilde{P}^\star, \widetilde{\boldsymbol{\Phi}}}(s, \cdot)\right] \\ &= \mathbb{E}_{\widetilde{\pi}_\mathcal{S}(\cdot|s)}\left[Q^{\widetilde{\pi}_\mathcal{S}}_{H, P^\star, \boldsymbol{\Phi}}(s, \cdot)\right] \\ &= V^{\widetilde{\pi}_\mathcal{S}}_{H, P^\star, \boldsymbol{\Phi}'}(s). \end{aligned}$$

This proves (F.2) for step $H$. Suppose that (F.2) holds at some step $h+1$, that is,

$$(\rho')^h \cdot V^{\widetilde{\pi}}_{h+1, \widetilde{P}^\star, \widetilde{\boldsymbol{\Phi}}}(s) = V^{\widetilde{\pi}_\mathcal{S}}_{h+1, P^\star, \boldsymbol{\Phi}'}(s), \quad \forall s \in \mathcal{S}. \tag{F.3}$$

Then for step $h$, by robust Bellman equation and Proposition 4.2, we have that

$$\begin{aligned} (\rho')^{h-1} \cdot Q^{\widetilde{\pi}}_{h, \widetilde{P}^\star, \widetilde{\boldsymbol{\Phi}}}(s, a) &= (\rho')^{h-1} \cdot \left(\frac{\gamma}{\rho'}\right)^{H-1} \cdot R_h(s, a) + (\rho')^{h-1} \cdot \mathbb{E}_{\widetilde{\mathcal{P}}_\rho(s, a; \widetilde{P}^\star_h)}\left[V^{\widetilde{\pi}}_{h+1, \widetilde{P}^\star, \widetilde{\boldsymbol{\Phi}}}\right] \\ &= R_{\gamma, h}(s, a) + (\rho')^{h-1} \cdot \rho' \cdot \mathbb{E}_{\widetilde{\mathcal{B}}_\rho(s, a; \widetilde{P}^\star_h)}\left[V^{\widetilde{\pi}}_{h+1, \widetilde{P}^\star, \widetilde{\boldsymbol{\Phi}}}\right], \tag{F.4} \end{aligned}$$

where the last equality utilizes Proposition 4.2 since $\min_{s \in \widetilde{\mathcal{S}}} V^{\widetilde{\pi}}_{h+1, \widetilde{P}^\star, \widetilde{\boldsymbol{\Phi}}}(s) = 0$, and we adopt the notation

$$\widetilde{\mathcal{B}}_\rho(s, a; \widetilde{P}^\star_h) = \left\{\widetilde{P}(\cdot) \in \Delta(\widetilde{\mathcal{S}}) : \sup_{s' \in \widetilde{\mathcal{S}}} \frac{\widetilde{P}(s')}{\widetilde{P}^\star_h(s'|s, a)} \le \frac{1}{\rho'}\right\}.$$

Notice that by the definition (B.3), we know for $(s, a) \in \mathcal{S} \times \mathcal{A}$ it holds that $\widetilde{P}^\star_h(\cdot|s, a) = P^\star_h(\cdot|s, a)$ which is supported on $\mathcal{S}$. Therefore, we can equivalently write

$$\begin{aligned} \widetilde{\mathcal{B}}_\rho(s, a; \widetilde{P}^\star_h) &= \left\{\widetilde{P}(\cdot) \in \Delta(\widetilde{\mathcal{S}}) : \sup_{s' \in \mathcal{S}} \frac{\widetilde{P}(s')}{\widetilde{P}^\star_h(s'|s, a)} \le \frac{1}{\rho'}\right\} \\ &= \left\{\widetilde{P}(\cdot) \in \Delta(\mathcal{S}) : \sup_{s' \in \mathcal{S}} \frac{\widetilde{P}(s')}{P^\star_h(s'|s, a)} \le \frac{1}{\rho'}\right\} \\ &= \mathcal{B}_\rho(s, a; P^\star_h). \tag{F.5} \end{aligned}$$

Thus by (F.4) and (F.5) and the induction hypothesis (F.3), we obtain that for any $(s, a) \in \mathcal{S} \times \mathcal{A}$,

$$\begin{aligned} (\rho')^{h-1} \cdot Q^{\widetilde{\pi}}_{h, \widetilde{P}^\star, \widetilde{\boldsymbol{\Phi}}}(s, a) &= R_{\gamma, h}(s, a) + (\rho')^h \cdot \mathbb{E}_{\mathcal{B}_\rho(s, a; P^\star_h)}\left[V^{\widetilde{\pi}}_{h+1, \widetilde{P}^\star, \widetilde{\boldsymbol{\Phi}}}\right] \\ &= R_{\gamma, h}(s, a) + \mathbb{E}_{\mathcal{B}_\rho(s, a; P^\star_h)}\left[V^{\widetilde{\pi}_\mathcal{S}}_{h+1, P^\star, \boldsymbol{\Phi}}\right] = Q^{\widetilde{\pi}_\mathcal{S}}_{h, P^\star, \boldsymbol{\Phi}}(s, a), \end{aligned}$$

where the second equality applies (F.3) and the last equality is from robust Bellman equation. Consequently, for any $s \in \mathcal{S}$, we have that

$$\begin{aligned} (\rho')^{h-1} \cdot V^{\widetilde{\pi}}_{h, \widetilde{P}^\star, \widetilde{\boldsymbol{\Phi}}}(s) &= \mathbb{E}_{\widetilde{\pi}(\cdot|s)}\left[(\rho')^{h-1} \cdot Q^{\widetilde{\pi}}_{h, \widetilde{P}^\star, \widetilde{\boldsymbol{\Phi}}}(s, \cdot)\right] \\ &= \mathbb{E}_{\widetilde{\pi}_\mathcal{S}(\cdot|s)}\left[Q^{\widetilde{\pi}_\mathcal{S}}_{h, P^\star, \boldsymbol{\Phi}}(s, \cdot)\right] \\ &= V^{\widetilde{\pi}_\mathcal{S}}_{h, P^\star, \boldsymbol{\Phi}'}(s), \end{aligned}$$

which finishes the induction argument, proving our claim (F.2). By taking $h = 1$, we can derive that for any initial state $s_1 \in \mathcal{S}$, it holds that for any policy $\widetilde{\pi}$ of $\widetilde{\mathcal{M}}$ and its induced policy $\widetilde{\pi}_\mathcal{S}$ of $\mathcal{M}_\gamma$,

$$V^{\widetilde{\pi}}_{1, \widetilde{P}^\star, \widetilde{\boldsymbol{\Phi}}}(s_1) = V^{\widetilde{\pi}_\mathcal{S}}_{1, P^\star, \boldsymbol{\Phi}'}(s_1).$$

This indicates two facts: the first is that

$$\max_{\widetilde{\pi}} V^{\widetilde{\pi}}_{1,\widetilde{P}^\star,\widetilde{\Phi}}(s_1) = \max_{\pi} V^{\pi}_{1,P^\star,\Phi'}(s_1), \tag{F.6}$$

where on the right hand side the maximization is with respect to all the policies for $\mathcal{M}_\gamma$; the second is that

$$V^{\widetilde{\pi}^k}_{1,\widetilde{P}^\star,\widetilde{\Phi}}(s_1) = V^{\widetilde{\pi}^k_{\mathcal{S}}}_{1,P^\star,\Phi'}(s_1), \tag{F.7}$$

for each $k \in [K]$, where recall that $\widetilde{\pi}^k$ is the policy output by Algorithm 1 for episode $k$. As a result, the $k$ policies $\{\widetilde{\pi}^k_{\mathcal{S}}\}^K_{k=1}$ of $\mathcal{M}_\gamma$ during interactive data collection satisfies with probability at least $1-\delta$,

$$
\begin{aligned}
\mathrm{Regret}_{\Phi'}(K) &= \sum_{k=1}^{K} \max_{\pi} V^{\pi}_{1,P^\star,\Phi'}(s_1) - V^{\widetilde{\pi}^k_{\mathcal{S}}}_{1,P^\star,\Phi'}(s_1) \\
&= \sum_{k=1}^{K} \max_{\widetilde{\pi}} V^{\widetilde{\pi}}_{1,\widetilde{P}^\star,\widetilde{\Phi}}(s_1) - V^{\widetilde{\pi}^k}_{1,\widetilde{P}^\star,\widetilde{\Phi}}(s_1) \\
&\leq \mathcal{O}\left( \sqrt{\min\left\{H, (2-2\rho')^{-1}\right\} H^2 SAK\iota'} \right),
\end{aligned}
$$

where in the second equality we apply the facts (F.6) and (F.7), and the last inequality follows from (F.1) and that $\rho = 2 - 2\rho'$. This completes the proof of Corollary B.5. $\qquad\square$

