# OpenReview forum: "Distributionally Robust Reinforcement Learning with Interactive Data Collection: Fundamental Hardness and Near-Optimal Algorithms"
_NeurIPS.cc/2024/Conference — NeurIPS 2024 poster_

### Official Review · Reviewer_bEix · 2024-07-07

**Soundness:** 3
**Presentation:** 3
**Contribution:** 3
**Rating:** 6
**Confidence:** 4

**Summary:**

This work deal with sample complexity of Robust MDPs. The major improvement is that, contrary to previous research that relies on generative models or pre-collected datasets, this paper focuses on RMDPs learning through interactive data collection, addressing two key challenges: distributional robustness and balancing exploration and exploitation.

Three main contributions are :

1)  Explaining that sample-efficient learning is unachievable without additional assumptions due to the curse of support shift, where training and testing environments may have non-overlapping distributions.
2) Introducing of the vanishing minimal value assumption for Robust Markov Decision Processes (RMDPs) with a total-variation distance robust set, assuming the minimal value of the optimal robust value function is zero, leading to a tractable case.
3)  Proposing an algorithm with a provable sample complexity guarantee under this framework.

**Strengths:**

The strengths of this paper are :

1) The paper is clearly written and ideas are well exposed.
2) Authors derive a nice lower bound with counterexample on the sample complexity of RMDPs in an online setting, and tight upper bound using extra assumptions such that vanishing minimal value assumption where TV uncertainty set can be rewritten using Radon-Nikodym derivatives.
3) The proof seems correct for me.
4) Algorithm is quite classic but make sense to derive robust policy while balancing exploration and exploitation.

**Weaknesses:**

5) It would also be interesting to extend results for $s$- rectangular case.
6) There is no lower bound with vanishing minimal assumption to ensure that the upper bound under these assumptions is tight.
7)  It would be nice to add the range of $\epsilon$ for the upper bound where sample complexity is valid or give  a condition on K rather than saying "lower order term in K"
8) I think it would be interesting to gives more intuition on vanishing minimal assumption.

**Questions:**

9) Do you think that vanishing minimal assumption is restrictive to derive Deep Robust RL algorithm ?
10) Do you think it would be possible to adapt Proposition 4.2 to other norms such as $L_p$ ?
11) To clarify thing, the main difference between generative model setting and online setting in RMDPs is to deal with support shift of kernel ?
12) I understand that KL and $\chi^2$ is a slightly different problem, does vanishing minimal assumption for KL or $\chi^2$ divergences RMDPs make sense as the definition of these divergences already impose bounded support shift of transition kernel ?
13) From a practical point of view, would the idea that sample complexity of RMDPS is smaller than MDPs (both in online and generative model setting) could lead to more sample efficient algorithms ?

**Limitations:**

No limitations.

---

> ### Author Rebuttal · Authors · 2024-08-05
>
> **Q1: It is interesting to extend to $\mathcal{S}$-rectangular case.**
>
> **A1:** Thanks, we appreciate your suggestions! But still, we would like to emphasize that our work is the first one on robust RL with interactive data collection that proves the hardness result and provides the algorithm with sharp sample complexity under suitable assumptions. We leave possible extensions to a broader range of cases to future work.
>
> **Q2: About the lower bound under the vanishing minimal value assumption.**
>
> **A2:** Thank you for this question! The prior work [1] derived a lower bound of $\Omega(\min\{H_\gamma,1/\rho\}H_\gamma^2SA/\varepsilon^2)$ for discounted RMDPs with generative model, where $H_\gamma = 1/(1-\gamma)$ is the effective horizon. After examining the proof, we find this lower bound can be extended to finite-horizon RMDPs, giving an $\Omega(\min\{H,1/\rho\}H^2SA/\varepsilon^2)$ lower bound. This applies to both the generative model setting and the online interactive learning setting, as the latter is considered more challenging.
>
> However, we acknowledge that the hardness instance given by [1] does not satisfy the vanishing minimal value assumption and requires some modifications. We conjecture that this lower bound still holds with the vanishing minimal value assumption and we will attempt to provide a rigorous proof in the revision.
>
> **Q3: On lower order terms in $K$ and the range of $\varepsilon$ s.t. the sample complexity is valid.**
>
> **A3:** Thanks for pointing this out! For the lower order terms in $K$ of the online regret, we actually included them clearly in the proof (see Appendix E.1, Line 999). Correspondingly, the sample complexity of Algorithm 1 is $\widetilde{O}(\min\{H,1/\rho\}H^2SA/\varepsilon^2 + H^3SA/\varepsilon)$ for any $\varepsilon>0$. Thus Corollary 4.4 holds for $\varepsilon\in(0,c\cdot\min\{1, 1/(H\rho)\}]$ with $c$ being an absolute constant. We will make this clear in the revision.
>
> **Q4: About intuitions on the vanishing minimal value assumption.**
>
> **A4:** As we concluded in the paper, the main difficulty of robust RL with interactive data collection is from the curse of support shift. By looking into the duality representation of the robust value functions, once the minimal values vanishes, the robust set is then equivalent to a new type of robust set without explicit support shift, thus combatting the original difficulty (Proposition 4.2). We also refer to the discussions after Proposition 4.2 for more explanations (Lines 259-265).
>
> **Q5: Is the vanishing minimal value assumption restrictive for deep robust RL?**
>
> **A5:** Thank you for asking! The vanishing minimal value assumption is a general, not restrictive, assumption for deep reinforcement learning in real-world applications. One sufficient condition for this assumption is the fail-state assumption, which is common in practice. For example, in robotics scenarios where a "destroyed robotics" state is absorbing and yields a minimum zero reward, it satisfies both the fail-state assumption and the more general vanishing minimal value assumption.
>
> **Q6: Is it possible to adapt Proposition 4.2 to other norms such as $L_p$?**
>
> **A6:** Thanks for the interesting direction! We do notice that previous works e.g., [2] (and references therein) considered the general $L_p$-norm, but in a different learning setup than ours. However, the general $L_p$-norm robust set gives a complicated duality representation for the robust Bellman operator (Lemma B.5 in [2]), and the vanishing minimal value assumption alone does not provide similar equivalence results like Proposition 4.2 to combat the curse of support shift. This assumption does rely on $p=1$. Therefore, a direct extension of Proposition 4.2 to $L_p$-norm is hard. However, we appreciate this question. It definitely serves as an exciting direction to study when robust RL with interactive data collection is possible under general $L_p$-norm robust set.
>
> **Q7: Is the difference between generative model and the online setting to deal with support shift?**
>
> **A7:** Yes! More specifically: For generative model setting, since the learner can query each state-action pair for the next state, there is no curse of support shift in estimating the nominal transition. In contrast, in the online setting, the agent collects data by interacting with the training environment, and there could exist hard-to-reach states which are important for generalizing to testing environments (support shift happens).
>
> **Q8: Does vanishing minimal value assumption for KL or $\chi^2$ divergences make sense?**
>
> **A8:** As we have already mentioned in **Q4** and **Q6**, this assumption is tailored for the TV robust set. It is not a suit for KL or $\chi^2$ divergences RMDPs.
>
> However, we note that even though there is no explicit support shift for KL and $\chi^2$, we an still build hard instances where the probability of reaching certain states that appear in the test environment is extremely low in the training environment (a broader understanding of support shift, see Appendix B.3). Thus it is unknown whether robust RL is possible with interactive data collection for these types of robust sets even with suitable assumptions. We leave this question as future work.
>
> **Q9: Would that the sample complexity of RMDPs is smaller than MDPs leads to more sample efficient algorithms?**
>
> **A9:** We remark that RMDPs and MDPs use different performance metrics. The former seeks the optimal robust value function while the latter finds the optimal standard value. Therefore, it is reasonable that deep robust RL algorithms find *robust* optimal policy using less samples than standard RL algorithms to find a standard optimal policy.
>
> **References:**
>
> [1] Shi, Laixi, et al. "The curious price of distributional robustness in reinforcement learning with a generative model." NeurIPS 2023.
>
> [2] Clavier, Pierre, et al. "Towards minimax optimality of model-based robust reinforcement learning." arXiv preprint 2023.

---

> > ### Comment · Reviewer_bEix · 2024-08-09
> >
> > Thank you for your detailed answers. I maintain my score and advocate for acceptance.

---

> > > ### Author Response · Authors · 2024-08-09
> > >
> > > Thank you for your efforts reviewing our paper and your support! We will further improve our paper following your suggestions during revision.

---

### Official Review · Reviewer_sH9q · 2024-07-12

**Soundness:** 3
**Presentation:** 3
**Contribution:** 3
**Rating:** 7
**Confidence:** 3

**Summary:**

This paper studies the learnability of the optimal policy for robust Markov decision process (RMDP) under the interactive data setting. The paper first show a fundamental hardness results which necessitates identifying a subclass of RMDPs which is actually solvable. The authors propose an algorithm whose sample complexity and regret are analyzed.

**Strengths:**

1. The paper is well-organized and well-written.
2. The setting is meaningful and motivated.
3. The analysis is thorough.
4. The comparisons with existing works are very detailed, making the paper easy to follow.

**Weaknesses:**

I am not aware of notable weaknesses.

**Questions:**

1. In my understanding, this is the first **value-based** online/interactive data collection DRRL algorithm. Can authors confirm this is the case?
2.  I am aware of Dong et al., 2022, and I am also aware that their proof has fatal flaws. I would like to mention that there is another actor-critic method for DRRL, Zhou et al., 2023 [1]. I believe this can also handle online interaction with the nominal model and learn a robust policy. Can the authors compare it with your work, especially about their robust critic component.
3. Assumption 4.1 is a sufficient condition for any RMDP to be solvable. Can the authors share some insights about the "gap" between this sufficient condition and a could-be necessary condition.

Reference
[1] ZHOU, R., LIU, T., CHENG, M., KALATHIL, D., KUMAR, P. and TIAN, C. (2023). Natural actor-560
critic for robust reinforcement learning with function approximation. In Thirty-seventh Conference561
on Neural Information Processing Systems.

**Limitations:**

No other limitations I would like to bring up.

---

> ### Author Rebuttal · Authors · 2024-08-05
>
> **Q1: Whether our work is the first value-based interactive data collection DRRL algorithm?**
>
> **A1:** Thanks for pointing this out! But we want to clarify that our algorithm is *model-based* since it necessitates an explicit estimation of the training environment transition kernel, denoted by $\widehat{P}$. We will make this conceptual point clearer in the revision.
>
> **Q2: Comparison with the previous work [1].**
>
> **A2:** Thanks for pointing out this interesting work [1]! We have also cited and mentioned this work in the paper. [1] proposes a novel robust natural actor-critic algorithm under function approximation to solve robust RL with interactive data collection, also with certain theoretical guarantees.
>
> The key difference between the theory in [1] and our work is in the data assumptions and thus the subsequent techniques required to design and analysis the respective algorithms. Regarding the data assumptions, [1] relies on interacting with a nominal environment that satisfies several concentrability and mixing assumptions (Assumptions 1, 3, 6 in [1]), and thus [1] does not explicitly address the problem of *exploration* where the agent needs to adaptively use interactive data collection to explore and robustify its policy. In contrast, our work does not make any assumption regarding the concentrability or mixing properties of the underlying nominal MDP. Instead, we use algorithmic design to incentivize the agent to explore automatically.
>
> Besides, regarding the comparison to the robust critic component, since our algorithm is not in the actor-critic style, a direct comparison is hard. But still, we maintain a robust value function estimation during training, which is similar to a robust critic. The key difference is that, in order to address the fundamental challenge of exploration during interactive data collection, our robust value estimator features a carefully designed optimistic bonus that encourages the agent to explore the nominal environment sample-efficiently. This is different from the design idea of the robust critic in [1].
>
> **Q3: About the "gap" between the sufficient condition (Assumption 4.1) and other potentially necessary conditions.**
>
> **A3:** While we have established a *worst-case* hardness result without the vanishing minimal value assumption, we acknowledge that more general sufficient or even necessary conditions for sample-efficient learning with interactive data collection may exist. We agree with the reviewer that this is an interesting and challenging direction to explore, as our vanishing minimal value assumption is currently the most general sufficient condition enabling sample-efficient learning to the best of our knowledge. We will include further discussions on this topic in our revision and will consider it for future research.
>
> **References:**
>
> [1] Zhou, Ruida, et al. "Natural actor-critic for robust reinforcement learning with function approximation." Advances in neural information processing systems 36 (2023).

---

> > ### Comment · Reviewer_sH9q · 2024-08-12
> >
> > Dear authors,
> >
> > Thank you for responding to my questions. I will maintain my rating. Good luck.

---

> > > ### Author Response · Authors · 2024-08-13
> > >
> > > Thank you very much for your efforts reviewing our paper and we appreciate your support! We will further improve our paper following your suggestions during revision.

---

### Official Review · Reviewer_BadN · 2024-07-12

**Soundness:** 3
**Presentation:** 3
**Contribution:** 3
**Rating:** 5
**Confidence:** 3

**Summary:**

This paper studies robust RL in a finite-horizon RMDP through interactive data collection. They give both a fundamental hardness result in the general case and a sample-efficient algorithm within tractable settings.

**Strengths:**

1. Unlike previous work, which relies on a generative model or a pre-collected offline dataset enjoying good coverage of the deployment environment, this paper tackles robust RL via interactive data collection.
2. This paper introduces the vanishing minimal value assumption to RMDPs with a TV-distance robust set, postulating that the minimal value of the optimal robust value function is zero, which eliminates the support shift issue for RMDPs.
3. This paper proposes an algorithm with sharp sample complexity.

**Weaknesses:**

1. This paper is purely theoretical. Although I understand the focus of this paper, but I still want to see some empirical results or even some simulations to get more insight. Moreover, an algorithm is given in this article, thus some numerical studies are required.
2. I am not sure if the Interactive Data Collection design is feasible in practice especially in some real-world problems.
3. Some more detailed discussion about the comparison between the proposed method and the baseline methods are required especially when they are designed for different cases. So maybe some numerical results are helpful.

**Questions:**

Please see the weakness part.

---

> ### Author Rebuttal · Authors · 2024-08-05
>
> **Q1: This paper is purely theoretical. Although the reviewer understands the focus of this paper, but still want to see some empirical results to get more insight. Moreover, since an algorithm is given in this article, some numerical studies and comparisons are required. (Weakness 1&3)**
>
> **A1:** Thank you for recognizing that our work focuses on theoretical study. We appreciate your suggestion to add experimental results. We will consider including them in the revision.
>
> **Q2: About the feasibility of the interactive data collection setup. (Weakness 2)**
>
> **A2:** Thanks for the question! Yes, this setup is indeed possible and is even a common practice in certain real-world problems. For examples, in robust RL for robotics, e.g., [1, 2, 3, 4], the agent is usually trained in a simulated environment (corresponding to the training environment in our theoretical setup) through trial and error, which is interactive data collection since the algorithm does not use pre-collected offline data or a generative model (we remark that in the theoretical literature a generative model means that the algorithm has the access to directly query the next state given *any* state-action pair). The goal is to ensure that the robot still functions well even if the environment changes (corresponding to the testing environment in the theoretical setup), e.g., when the coefficient of friction differs or the weight of the robot changes. Thus the interactive data collection setup is indeed feasible in real-world problems.
>
> **References:**
>
> [1] Pinto, Lerrel, James Davidson, Rahul Sukthankar, and Abhinav Gupta. "Robust adversarial reinforcement learning." In International conference on machine learning, pp. 2817-2826. PMLR, 2017.
>
> [2] Tessler, Chen, Yonathan Efroni, and Shie Mannor. "Action robust reinforcement learning and applications in continuous control." In International Conference on Machine Learning, pp. 6215-6224. PMLR, 2019.
>
> [3] Zhao, Wenshuai, Jorge Peña Queralta, and Tomi Westerlund. "Sim-to-real transfer in deep reinforcement learning for robotics: a survey." 2020 IEEE symposium series on computational intelligence (SSCI). IEEE, 2020.
>
> [4] Brunke L, Greeff M, Hall A W, et al. Safe learning in robotics: From learning-based control to safe reinforcement learning[J]. Annual Review of Control, Robotics, and Autonomous Systems, 2022, 5(1): 411-444.

---

> > ### Comment · Area_Chair_Gd61 · 2024-08-12
> > **Engage with authors**
> >
> > Dear Reviewer,
> >
> > Please engage with the authors. This is the last day a back and forth discussion is possible.
> >
> > The AC

---

> > ### Comment · Reviewer_BadN · 2024-08-13
> >
> > Hope you can include some numerical studies in the final version, as you mentioned that the setup in this paper 'is indeed possible and is even a common practice in certain real-world problems.'

---

> > > ### Author Response · Authors · 2024-08-13
> > >
> > > Thank you for your valuable time reviewing our work and thanks for your support! We appreciate your suggestions on adding numerical experiments in the revision. We will include that and further improve our paper following your suggestion during revision.

---

### Official Review · Reviewer_3BYp · 2024-07-14

**Soundness:** 3
**Presentation:** 3
**Contribution:** 3
**Rating:** 6
**Confidence:** 4

**Summary:**

The paper addresses the challenges in distributionally robust reinforcement learning (DRRL), particularly focusing on robust Markov decision processes (RMDPs) under the framework of interactive data collection. Unlike previous work that depends on generative models or pre-collected datasets, this study emphasizes interactive data collection where the learner refines policies through interaction with the training environment. The main contributions include identifying fundamental challenges that for total variation (TV) distance, there exists a hard RMDP that all algorithms at least need $O(KH)$ regret. In addition, by introducing a vanishing minimal value assumption to mitigate these challenges, this work proposes a sample-efficient algorithm (OPROVI-TV) with regret $O(\sqrt{\min\{H, 1/\rho\}H^2 SAK } after $K$ trajectories, which matches the results from the state-of-the-art non-robust MDP online learning and also robust MDP with generative model.

**Strengths:**

1. This work focuses on interactive data collection for robust RL, addressing practical challenges and moving beyond reliance on generative models or pre-collected datasets, which is an underdeveloped open direction.
2. The proposed OPROVI-TV algorithm balances exploration and exploitation, providing strong guarantees for online regret and sample complexity for such robust RL problems with online settings.

**Weaknesses:**

1. The main assumption (Assumption 4.1) that this work mainly depends on is interesting, but will such assumptions make the problems (robust MDPs with TV uncertainty set ) not a suitable and meaningful robust RL problems for those tasks?
Specifically, no matter Assumption 4.1 or the fail-state assumption from [1] indeed implies that for all policy, $ \min_{s\inS} V_{h}^{\pi}(s) = 0 $ for all $h=1,2,\cdots, H$ if the reviewer does not miss something. The main concern from the reviewer is that (actually the author already discuss about this in Appendix B.4.1): For those tasks under such assumptions, if we consider robust MDPs using TV distance, will those robust MDPs directly equivalent to some non-robust MDPs but with a discounted reward (non-robust MDP with reward function $r<1-\rho/2)$.
* So in such cases, will the robust RL problem using TV to construct the uncertainty set be meaningful? I think those robust formulations will reduce back to non-robust MDPs. So maybe some uncertainty set with support control (the adversarial can't make the transition kernel to be out of the support of the nominal transition kernel) will be more suitable to use in such tasks.
* In addition, the reviewer believe that under Assumption 4.1, the regret of this problem can be improved to $O(\sqrt{\min\\{H, 1/\rho\\}^3 SAK}?$ Since the maximum value of the robust value function will be $\min\\{ H, 1/\rho\\}$ but not $H$ in the non-robust value function anymore.



[1]Panaganti, Kishan, and Dileep Kalathil. "Sample complexity of robust reinforcement learning with a generative model." International Conference on Artificial Intelligence and Statistics. PMLR, 2022.

**Questions:**

1. In the introduction "Given that all the existing52
literature on robust RL theory relies on a generative model or pre-collected data"
Not all the existing works are using generative model or offline dataset? Such as [1]. If so, please check


[1] Dong, Jing, et al. "Online policy optimization for robust mdp." arXiv preprint arXiv:2209.13841 (2022).

**Limitations:**

Well-answered

---

> ### Author Rebuttal · Authors · 2024-08-05
>
> **Q1: About whether Assumption 4.1 reduces our problem setup to a non-robust RL problem. (Weakness 1&2)**
>
> **A1:** Here we clarify that Assumption 4.1 does **not** reduce the problem to its non-robust counterpart.
>
> It is a *misinterpretation* of the discussions in Appendix B.4.1 that Assumption 4.1 reduces the problem to non-robust RL. In fact, Appendix B.4.1 clearly states that (Lines 706-709) under Assumption 4.1, the TV-robust MDP is equivalent to a discounted *robust* MDP with another formulation of robust set (bounded transition probability ratio w.r.t. the nominal model, see Proposition 4.2). Thus, under Assumption 4.1, our problem setup is still in the regime of robust RL and is fundamentally different from doing non-robust RL.
>
> Regarding robust sets where one explicitly assumes that the adversarial can't change the support, again thanks to the equivalent result in Proposition 4.2, we can actually solve robust RL with interactive data collection for discounted robust MDPs with robust set containing transitions that have bounded ratio to the nominal model (and thus the support does not change) by using a variant of our Algorithm 1 (see discussions in Appendix B.4.3).
> It is an interesting future work to consider other (perhaps more general) types of robust set where the adversarial can't change the support.
>
>
> **Q2: Whether the regret is further improvable to $\widetilde{O}(\sqrt{\min\{H, 1/\rho\}^3SAK})$ given that the maximum value of the robust value function is $\min\{H, 1/\rho\}$? (Weakness 3)**
>
> **A2:** We first want to point out that achieving the regret bound $\widetilde{O}(\sqrt{\min\{H, 1/\rho\}^3SAK})$ seems impossible. Specifically, even for MDPs with $V_{\text{MAX}} = R$, the lower bound still scales with $\Omega(\sqrt{R^2HSAK})$. The intuition is that when we consider the inhomogeneous setting where the transition kernel $\{P_h^{\star}\}_{h\in H}$ varies in the $H$ time steps (as we considered in this paper), the factor $\sqrt{H}$ is inevitable. We believe that the dependence on $H$ instead of $\min\{H, 1/\rho\}$ is inherent to the problem and cannot be further improved.
>
> In addition, we need to point out that, actually the proof of the current result has already utilized the fact that the upper bound and the variance of the robust value functions are controlled by terms related to $\min\{H, 1/\rho\}$, as is observed by the reviewer. More specifically, in the proofs of Lemma C.2 (showing that the robust value estimators are optimistic/pessimistic, see Eq (C.15)) and Lemma C.5 (controlling the summation of robust value functions over time horizons and episodes, see Eq (C.22)), we have utilized the fact that the robust value function is bounded by $\min\{H, 1/\rho\}$ to make the sharpest of our result.
>
> Given all that, we are still the first work that provides the algorithm with sharp sample complexity bound in the interactive data collection regime. We leave it as a future work to further figure out the sample complexity lower bound under assumptions that can make interactive data collecting robust RL feasible.
>
> **Q3: About the previous work [1] on robust RL with interactive data collection.**
>
> **A3:** We do agree that this work also considers robust RL that relies on interactive data collection, and we actually cited and discussed this work in our paper (see Appendix B.1, Lines 649 to 653).
>
> However, as we pointed out in Appendix B.1, this work exhibits an essential flaw (misuse of Lemma 12 therein) in the proof of their main result (online regret). This error invalidates their theoretical results, as Reviewer sH9q has also recognized. Thus, even though [1] works on the setup of interactive data collection, they actually did not answer the question that *"Can we design a provably sample-efficient robust RL algorithm that relies on interactive data collection in the training environment?"*. In contrast, we are the first work that proves the hardness result in the general case and provides the algorithm with sharp sample complexity bound in the interactive data collection regime under suitable assumptions. We would make this comparison clearer in the main part of our paper during revision.
>
>
> **References:**
>
> [1] Dong, Jing, et al. "Online Policy Optimization for Robust MDP." arXiv preprint arXiv:2209.13841 (2022).

---

> > ### Comment · Area_Chair_Gd61 · 2024-08-12
> > **Engagement with authors**
> >
> > Dear Reviewer,
> >
> > Please engage with the authors. This is the last day a back and forth discussion is possible.
> >
> > The AC

---

> > > ### Comment · Reviewer_3BYp · 2024-08-13
> > > **Response to the authors**
> > >
> > > Thanks for the detailed explaination. The reviewer's concerns are mainly addressed. I raised my score. One more question: Could you explain more about the difference between Assumption 1 and fail state assumption. I know there is a detailed discussion, while still want more intuitions.

---

> > > > ### Author Response · Authors · 2024-08-13
> > > >
> > > > Thank you for your valuable time reviewing our paper! We are glad that we have mainly addressed your concerns and thank you very much for raising your score!
> > > >
> > > > **Q: Could you explain more about the difference between Assumption 1 and fail state assumption?**
> > > >
> > > > **A:** Sure we are happy to explain more on these assumptions. As noticed by the reviewer, we have discussed in the paper that the vanishing minimal value assumption (Assumption 4.1) is *strictly a more general assumption* than the fail-state condition (Condition B.2 in the Appendix). That being said, the any RMDP satisfying the fail-state assumption also satisfies the vanishing minimal value assumption. But there exist instances that satisfy the vanishing minimal value assumption but do not have a fail state (Remark B.3). To get more intuition on why such instance exists, we notice that the robust value function is the worst case total return over the robust set of transitions. Thus, even if there is no fail state (zero-reward and absorbing), there could exist a worst case transition such that the total return is still zero. That is the intuition behind the example of Remark 4.3.
> > > >
> > > > We hope this can answer your questions, and we are happy to explain more if there are further confusions or concerns.

---

### Decision · Program_Chairs · 2024-09-25

**Decision:**

Accept (poster)

**Comment:**

This work introduces algorithms for robust MDPs. It is the first work to prove bounds in the setting of interactive data collection from the base model in this scenario. The authors also prove impossibility results that justify the structural assumptions under which they analyze their algorithms. The reviewers agreed this work meet the criteria for publication at Neurips.